# Correcting Data Distribution Mismatch in Offline Meta-Reinforcement Learning with Few-Shot Online Adaptation

## Abstract

Offline meta-reinforcement learning (offline meta-RL) extracts knowledge from a given dataset of multiple tasks and achieves fast adaptation to new tasks. Recent offline meta-RL methods typically use task-dependent behavior policies (e.g., training RL agents on each individual task) to collect a multi-task dataset and learn an offline meta-policy. However, these methods always require extra information for fast adaptation, such as offline context for testing tasks or oracle reward functions. Offline meta-RL with few-shot online adaptation remains an open problem. In this paper, we first formally characterize a unique challenge under this setting: reward and transition distribution mismatch between offline training and online adaptation. This distribution mismatch may lead to unreliable offline policy evaluation and the regular adaptation methods of online meta-RL will suffer. To address this challenge, we introduce a novel mechanism of data distribution correction, which ensures the consistency between offline and online evaluation by filtering out out-of-distribution episodes in online adaptation. As few-shot out-of-distribution episodes usually have lower returns, we propose a *Greedy Context-based data distribution Correction* approach, called GCC, which greedily infers how to solve new tasks. GCC diversely samples "task hypotheses" from the current posterior belief and selects greedy hypotheses with higher return to update the task belief. Our method is the first to provide an effective online adaptation without additional information, and can be combined with off-the-shelf context-based offline meta-training algorithms. Empirical experiments show that GCC achieves state-of-the-art performance on the Meta-World ML1 benchmark compared to baselines with/without offline adaptation.

## 1 Introduction

Human intelligence is capable of learning a wide variety of skills from past history and can adapt to new environments by transferring skills with limited experience. Current reinforcement learning (RL) has surpassed human-level performance (Mnih et al., 2015; Silver et al., 2017; Hafner et al., 2019) but requires a vast amount of experience. However, in many real-world applications, RL encounters two major challenges: multi-task efficiency and costly online interactions. In multi-task settings, such as robotic manipulation or locomotion (Yu et al., 2020b), agents are expected to solve new tasks with few-shot adaptation with previously learned knowledge. Moreover, collecting sufficient exploratory interactions is usually expensive or dangerous in robotics (Rafailov et al., 2021), autonomous driving (Yu et al., 2018), and healthcare (Gottesman et al., 2019). One popular paradigm for breaking this practical barrier is *offline meta reinforcement learning* (offline meta-RL; Li et al., 2020; Mitchell et al., 2021), which trains a meta-RL agent with pre-collected offline multi-task datasets and enables fast policy adaptation to unseen tasks.

Recent offline meta-RL methods have been proposed to utilize a multi-task dataset collected by task-dependent behavior policies (Li et al., 2020; Dorfman et al., 2021). They show promise by solving new tasks with few-shot adaptation. However, existing offline meta-RL approaches require additional information or assumptions for fast adaptation. For example, FOCAL (Li et al., 2020) and MACAW (Mitchell et al., 2021) conduct offline adaptation using extra offline contexts for meta-testing tasks. BOReL (Dorfman et al., 2021) and SMAC (Pong et al., 2022) employ few-shot online adaptation, in which the former assumes known reward functions, and the latter focuses on offline meta-training

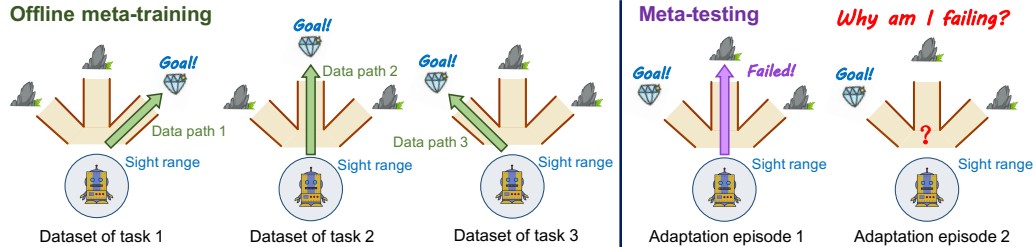

Figure 1: Illustration of data distribution mismatch between offline training and online adaptation.

with unsupervised online samples (without rewards). Therefore, achieving effective few-shot online adaptation purely relying on online experience remains an open problem for offline meta-RL.

One particular challenge of meta-RL compared to meta-supervised learning is the need to learn how to explore in the testing environments (Finn & Levine, 2019). In offline meta-RL, the gap of reward and transition distribution between offline training and online adaptation presents a unique conundrum for meta-policy learning, namely ***data distribution mismatch***. Note that this data distribution mismatch differs *distributional shift* in offline single-task RL (Levine et al., 2020), which refers to the shifts between offline policy learning and a behavior policy, not the gap between reward and transition distribution. As illustrated in Figure 1, when we collect an offline dataset using the expert policy for each task, the robot will be meta-trained on all **successful trajectories** (→). The robot aims to fast adapt to new tasks during meta-testing. However, when it tries the middle path and hits a stone, this **failed adaptation trajectory** (→) does not match the offline training data distribution, which can lead to false task inference or adaptation. To formally characterize this phenomenon, we utilize the perspective of Bayesian RL (BRL; Duff, 2002; Zintgraf et al., 2019; Dorfman et al., 2021) that maintains a task belief given the context history and learns a meta-policy on belief states. Our theoretical analysis shows that task-dependent data collection may lead the agent to out-of-distribution belief states during meta-testing and results in an inconsistency between offline meta-policy evaluation and online adaptation evaluation. This contrasts with policy evaluation consistency in offline single-task RL (Fujimoto et al., 2019). To deal with this inconsistency, we can choose either to trust the offline dataset or to trust new experience and continue online exploration. The latter may not be able to collect sufficient data in few-shot adaptation to learn a good policy only on online data. Therefore, we adopt the former strategy and introduce a mechanism of online data distribution correction that meta-policies with *Thompson sampling* (Strens, 2000) can filter out out-of-distribution episodes and provide a theoretical consistency guarantee in policy evaluations.

To realize our theoretical implications in practical settings, we propose a context-based offline meta-RL algorithm with a novel online adaptation mechanism, called ***Greedy Context-based data distribution Correction*** (GCC). To align adaptation context with the meta-training distribution, GCC utilizes greedy task inference, which diversely samples "task hypotheses" and selects hypotheses with higher return to update task belief. In Figure 1, the robot can sample other "task hypotheses" (i.e., try other paths) and the failed adaptation trajectory (middle) will not be used for task inference due to out-of-distribution with a lower return. To our best knowledge, our method is the first to design a delicate context mechanism to achieve effective online adaptation for offline meta-RL and has the advantage of directly combining with off-the-shelf context-based offline meta-training algorithms.

Our main contribution is to formalize a specific challenge (i.e., data distribution mismatch) in offline meta-RL with online adaptation and propose a greedy context mechanism with theoretical motivation. We extensively evaluate the performance of GCC in didactic problems proposed by prior work (Rakelly et al., 2019; Zhang et al., 2021) and Meta-World ML1 benchmark with 50 tasks (Yu et al., 2020b). In these didactic problems, GCC demonstrates that our context mechanism can accurately infer task identification, whereas the original online adaptation methods will suffer due to out-of-distribution data. Empirical results on the more challenging Meta-World ML1 benchmark show that GCC significantly outperforms baselines with few-shot online adaptation, and achieves better or comparable performance than offline adaptation baselines with expert context.

## 2    NOTATIONS AND PRELIMINARIES

### 2.1    STANDARD META-RL

The goal of meta-RL (Finn et al., 2017; Rakelly et al., 2019) is to train a meta-policy that can quickly adapt to new tasks using $N$ adaptation episodes. The standard meta-rl setting deals with a distribution

$p(\kappa)$ over MDPs, in which each task $\kappa_i$ sampled from $p(\kappa)$ presents a finite-horizon MDP (Zintgraf et al., 2019; Yin & Wang, 2021). $\kappa_i$ is defined by a tuple $\left(\mathcal{S}, \mathcal{A}, \mathcal{R}, H, P^{\kappa_i}, R^{\kappa_i}\right)$, including state space $\mathcal{S}$, action space $\mathcal{A}$, reward space $\mathcal{R}$, planning horizon $H$, transition function $P^{\kappa_i}(s'|s, a)$, and reward function $R^{\kappa_i}(r|s, a)$. Denote $\mathcal{K}$ is the space of task $\kappa_i$. In this paper, we assume dynamics function $P$ and reward function $R$ may vary across tasks and share a common structure. The meta-RL algorithms repeatedly sample batches of tasks to train a meta-policy. In the meta-testing, agents aim to rapidly adapt a good policy for new tasks drawn from $p(\kappa)$.

From a perspective of Bayesian RL (BRL; Ghavamzadeh et al., 2015), recent meta-RL methods (Zintgraf et al., 2019) utilize a Bayes-adaptive Markov decision process (BAMDP; Duff, 2002) to formalize standard meta-RL. A BAMDP is a belief MDP (Kaelbling et al., 1998) of a special Partially Observable MDP (POMDP; Astrom, 1965) whose unobserved state information presents unknown task identification in $N$ adaptation episodes. BAMDPs are defined as a tuple $M^+ = \left(\mathcal{S}^+, \mathcal{A}, \mathcal{R}, H^+, P_0^+, P^+, R^+\right)$ (Zintgraf et al., 2019), in which $\mathcal{S}^+ = \mathcal{S} \times \mathcal{B}$ is the hyper-state space, $\mathcal{B}$ is the space of task beliefs over meta-RL MDPs, $\mathcal{A}$ is the action space, $\mathcal{R}$ is the reward space, $H^+ = N \times H$ is the planning horizon across adaptation episodes, $P_0^+\left(s_0^+\right)$ is the initial hyper-state distribution representing task distribution $p(\kappa)$, $P^+\left(s_{t+1}^+ \middle| s_t^+, a_t, r_t\right)$ is the transition function, and $R^+\left(r_t \middle| s_t^+, a_t\right)$ is the reward function. A meta-policy $\pi^+\left(a_t \middle| s_t^+\right)$ on BAMDPs prescribes a distribution over actions for each hyper-state $s_t^+ = (s_t, b_t)$. The objective of meta-RL agents is to find a meta-policy $\pi^+$ that maximizes expected return, i.e., online policy evaluation denoted by $\mathcal{J}_{M^+}\left(\pi^+\right)$. Formal descriptions are deferred to Appendix A.1.3.

## 2.2 OFFLINE META-RL

In the offline meta-RL setting, a meta-learner only has access to an offline multi-task dataset $\mathcal{D}^+$ and is not allowed to interact with the environment during meta-training (Li et al., 2020). Recent offline meta-RL methods (Dorfman et al., 2021) always utilize task-dependent behavior policies $p(\mu|\kappa)$, which represents the random variable of the behavior policy $\mu(a|s)$ conditioned on the random variable of the task $\kappa$. For brevity, we overload $[\mu] = p(\mu|\kappa)$. Similar to related work on offline RL (Yin & Wang, 2021), we assume that $\mathcal{D}^+$ consists of multiple i.i.d. trajectories that are collected by executing task-dependent policies $[\mu]$ in $M^+$. We define the reward and transition distribution of the task-dependent data collection by $\mathbb{P}_{M^+,[\mu]}$ (Jin et al., 2021), i.e., for each step $t$ in a trajectory,

$$\mathbb{P}_{M^+,[\mu]}\left(r_t, s_{t+1} \middle| s_t^+, a_t\right) \propto \mathbb{E}_{\kappa_i \sim p(\kappa), \mu_i \sim p(\mu|\kappa_i)}\left[\mathbb{P}_{\kappa_i}\left(r_t, s_{t+1} \middle| s_t, a_t\right) \cdot p_{M^+}\left(s_t^+ \middle| \kappa_i, \mu_i\right)\right], \quad (1)$$

where the reward and transition distribution of data collection with $\mu_i$ in a task $\kappa_i$ is defined as

$$\mathbb{P}_{\kappa_i}\left(r_t, s_{t+1} \middle| s_t, a_t\right) = R^{\kappa_i}\left(r_t \middle| s_t, a_t\right) \cdot P^{\kappa_i}\left(s_{t+1} \middle| s_t, a_t\right), \quad (2)$$

and $p_{M^+}\left(s_t^+ \middle| \kappa_i, \mu_i\right)$ denotes the probability of $s_t^+$ when executing $\mu_i$ in a task $\kappa_i$. Note that the offline dataset $\mathcal{D}^+$ can be narrow and a large amount of state-action pairs are not covered. These unseen state-action pairs will be erroneously estimated to have unrealistic values, called a phenomenon of *extrapolation error* (Fujimoto et al., 2019). To overcome extrapolation error in offline RL, related works (Fujimoto et al., 2019) introduce batch-constrained RL, which restricts the action space in order to force policy selection of an agent with respect to a given dataset. A policy $\pi^+$ is defined to be batch-constrained by $\mathcal{D}^+$ if $\pi^+\left(a \middle| s^+\right) = 0$ for all $(s^+, a)$ tuples that are not contained in $\mathcal{D}^+$. Offline RL (Liu et al., 2020; Chen & Jiang, 2019) approximates policy evaluation for a batch-constrained policy $\pi^+$ by sampling from an offline dataset $\mathcal{D}^+$, which is denoted by $\mathcal{J}_{\mathcal{D}^+}\left(\pi^+\right)$ and called *Approximate Dynamic Programming* (ADP; Bertsekas & Tsitsiklis, 1995). During meta-testing, RL agents perform online adaptation using a meta-policy $\pi^+$ in new tasks drawn from meta-RL task distribution. The reward and transition distribution of data collection with $\pi^+$ in $M^+$ during adaptation is defined by

$$\mathbb{P}_{M^+}\left(r_t, s_{t+1} \middle| s_t^+, a_t\right) = R^+\left(r_t \middle| s_t^+, a_t\right) \cdot P^+\left(s_{t+1} \middle| s_t^+, a_t\right), \quad (3)$$

where $R^+, P^+$ are defined in $M^+$. Detailed formulations are deferred to Appendix A.1.4.

## 3 THEORY: DATA DISTRIBUTION MISMATCH CORRECTION

Consistency of training and testing conditions is an important principle for machine learning (Finn & Levine, 2019). Recently, offline meta-RL with task-dependent behavior policies (Dorfman et al., 2021) faces a special challenge: *the reward and transition distribution in offline training and online adaptation may not match*. We first build a theory about data distribution mismatch to understand

this phenomenon. Our theoretical analysis is based on Bayesian RL (BRL; Zintgraf et al., 2019) and demonstrates that data distribution mismatch may lead to out-of-distribution belief states during meta-testing and results in a large gap between online and offline policy evaluation. To address this challenge, we propose a new mechanism to correct data distribution, which transforms the BAMDPs (Duff, 2002) to enrich information about task-dependent behavior policies into overall beliefs. This transformed BAMDPs provide reliable policy evaluation by filtering out out-of-distribution episodes.

### 3.1 DATA DISTRIBUTION MISMATCH IN OFFLINE META-RL

We define data distribution mismatch in offline meta-RL as follows.

**Definition 1** (Data Distribution Mismatch). *In a BAMDP $M^+$, for each task-dependent behavior policies $[\mu]$ and batch-constrained meta-policy $\pi^+$, the data distribution mismatch between $\pi^+$ and $[\mu]$ is defined by that $\exists s_t^+, a_t$, s.t.,*

$$p_{M^+}^{\pi^+}\left(s_t^+, a_t\right) > 0 \quad and \quad \mathbb{P}_{M^+}\left(r_t, s_{t+1} \,\middle|\, s_t^+, a_t\right) \neq \mathbb{P}_{M^+,[\mu]}\left(r_t, s_{t+1} \,\middle|\, s_t^+, a_t\right), \tag{4}$$

*where $p_{M^+}^{\pi^+}\left(s_t^+, a_t\right)$ is the probability of reaching a tuple $\left(s_t^+, a_t\right)$ while executing $\pi^+$ in $M^+$ (formal definition deferred to Appendix A.2), and $\mathbb{P}_{M^+}, \mathbb{P}_{M^+,[\mu]}$ are the reward and transition distribution of data collection with $\pi^+$ and $[\mu]$ defined in Eq. (1) and (3), respectively.*

The data distribution induced by $\pi^+$ and $[\mu]$ mismatches when the reward and transition distribution of $\pi^+$ and $[\mu]$ differs in a tuple $\left(s_t^+, a_t\right)$, in which the agent can reach this tuple by executing $\pi^+$ in $M^+$, i.e., $p_{M^+}^{\pi^+}\left(s_t^+, a_t\right) > 0$. Note that if $\pi^+$ can reach a tuple $\left(s_t^+, a_t\right)$, this tuple is guaranteed to be collected in the offline dataset, i.e., $p_{M^+}^{[\mu]}\left(s_t^+, a_t\right) > 0$, because a batch-constrained policy $\pi^+$ will not select an action outside of the dataset collected by $[\mu]$, as introduced in Section 2.2.

**Theorem 1.** *There exists a BAMDP $M^+$ with task-dependent behavior policies $[\mu]$ such that, for any batch-constrained meta-policy $\pi^+$, the data distribution induced by $\pi^+$ and $[\mu]$ does not match.*

To serve a concrete example, we construct an offline meta-RL setting shown in Figure 2. In this example, there are $v$ meta-RL tasks $\mathcal{K} = \{\kappa_1, \ldots, \kappa_v\}$ and $v$ behavior policies $\{\mu_1, \ldots, \mu_v\}$, where $v \geq 3$. Each task $\kappa_i$ has one state $s_0$, $v$ actions, and horizon $H = 1$. For each task $\kappa_i$, RL agents can receive reward 1 performing action $a_i$. During adaptation, the RL agent can interact with the environment within $v$ episodes. The task distribution is uniform, the behavior policy of task $\kappa_i$ is $\mu_i$, and each behavior policy $\mu_i$ will perform $a_i$. When any

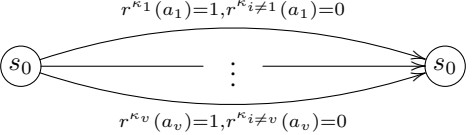

$$p(\kappa_i) = \tfrac{1}{v}, p(\mu_i|\kappa_i) = 1, \text{ and } \mu_i(a_i|s_0) = 1$$

Figure 2: A concrete example, which has $v$ meta-RL tasks, one state, $v$ actions, $v$ behavior policies, horizon $H = 1$ in an episode, and $v$ adaptation episodes, where $v \geq 3$.

batch-constrained meta-policy $\pi^+$ selects an action $\tilde{a}$ on the initial state $s_0^+$, we find that

$$\mathbb{P}_{M^+}\left(r = 1 \,\middle|\, s_0^+, \tilde{a}\right) = \frac{1}{v} \neq \mathbb{P}_{M^+,[\mu]}\left(r = 1 \,\middle|\, s_0^+, \tilde{a}\right) = 1, \tag{5}$$

in which there is the probability of $\frac{1}{v}$ to sample a corresponding testing task, whose reward function of $\tilde{a}$ is 1, whereas the reward in the offline dataset collected by $[\mu]$ is all 1.

**Fact 1.** *With any probability $1 - \delta \in [0, 1)$, there exists a BAMDP $M^+$ with task-dependent behavior policies $[\mu]$ such that, for any batch-constrained meta-policy $\pi^+$, the agent will visit out-of-distribution belief states during meta-testing due to data distribution mismatch.*

In the example shown in Figure 2, an offline multi-task dataset $D^+$ is drawn from the task-dependent data collection $\mathbb{P}_{M^+,[\mu]}$. Since the reward of $D^+$ is all 1, the belief states in $D^+$ has two types: (i) all task possible $s_0^+$ and (ii) determining task $i$ with receiving reward 1 in action $a_i$. For any batch-constrained meta-policy $\pi^+$ selecting an action $\tilde{a}_j$ on $s_0^+$ during meta-testing, there has probability $1 - \frac{1}{v}$ to receive reward 0 and the belief state will become "excluding task $j$", which is not contained in $D^+$ with $v \geq 3$. For any $\delta \in (0, 1]$, let $v > \frac{1}{\delta}$, with probability $1 - \delta$, the agent will visit out-of-distribution belief states during adaptation. Note that, this phenomenon will directly lead to the following proposition since RL agents will visit out-of-distribution belief states during meta-testing in BAMDPs and induce unreliable offline policy evaluation.

**Proposition 1.** *There exists a BAMDP $M^+$ with task-dependent behavior policies $[\mu]$ such that, for any batch-constrained meta-policy $\pi^+$, the gap of policy evaluation of $\pi^+$ between offline meta-training and online adaptation is at least $\frac{H^+ - 1}{2}$, where $H^+$ is the planning horizon of $M^+$.*

In Figure 2, an offline dataset $D^+$ only contains reward 1, thus for each batch-constrained meta-policy $\pi^+$, the offline evaluation of $\pi^+$ in $D^+$ is $\mathcal{J}_{D^+}(\pi^+) = H^+ = vH$, as defined in Section 2.2. The optimal meta-policy $\pi^{+,*}$ in this example is to enumerate $a_1, \ldots, a_v$ until the task identification is inferred from the action with a reward of 1. A meta-policy $\pi^{+,*}$ needs to explore in the testing environments and its online policy evaluation is $\mathcal{J}_{M^+}(\pi^{+,*}) = \frac{H^+ + 1}{2}$. The detailed proof is deferred to Appendix A.2. Thus, the evaluation gap between offline meta-training and online adaptation is

$$\left| \mathcal{J}_{M^+}(\pi^+) - \mathcal{J}_{D^+}(\pi^+) \right| \geq \mathcal{J}_{D^+}(\pi^+) - \mathcal{J}_{M^+}(\pi^{+,*}) = \frac{H^+ - 1}{2}. \tag{6}$$

Proposition 1 suggests that offline policy evaluation can fail by visiting out-of-distribution belief states. As few-shot online adaptation, we choose to trust the offline dataset and it is imperative to build a connection between online and offline evaluation for effective offline meta-policy training.

### 3.2 Data Distribution Correction Mechanism

To correct data distribution mismatch, we define the transformed BAMDPs as follows.

**Definition 2** (Transformed BAMDPs, informal)**.** *A transformed BAMDP is defined by a new BAMDP $\overline{M}^+$, whose belief $\bar{b}_t$ is about meta-RL MDPs $\kappa$ and task-dependent behavior policies $[\mu]$. The goal of meta-RL agents is to find a meta-policy $\bar{\pi}^+\left(a_t \,\middle|\, \bar{s}_t^+\right)$ that maximizes online policy evaluation $\mathcal{J}_{\overline{M}^+}(\bar{\pi}^+)$, where $\bar{s}_t^+ = (s_t, \bar{b}_t)$ is a hyper-state of $\overline{M}^+$. Denote the offline dataset on $\overline{M}^+$ by $\overline{\mathcal{D}}^+$.*

Transformed BAMDPs maintain an overall belief about the task and behavior policies given the current context history. Compared to the original BAMDPs stated in Section 2.2, the transformed BAMDPs incorporate additional information about offline data collection into beliefs. This transformation will introduce an extra condition to the online adaptation process, as indicated by the following fact.

**Fact 2.** *For feasible Bayesian belief updating, transformed BAMDPs confine the agent in the in-distribution belief states during meta-testing.*

During online adaptation, RL agents construct a hyper-state $\bar{s}_t^+ = (s_t, \bar{b}_t)$ from the context history and perform a meta-policy $\bar{\pi}^+\left(a_t \,\middle|\, \bar{s}_t^+\right)$. The new belief $\bar{b}_t$ accounts for the uncertainty of task MDPs and task-dependent behavior policies. In contrast with Fact 1, transformed BAMDPs do not allow the agent to visit out-of-distribution belief states with $\overline{\mathcal{D}}^+$. Otherwise, the context history will conflict with the belief of behavior policies, i.e., RL agents cannot update beliefs $\bar{b}_t$ when they have observed an event that they believe to have probability zero. For example in Figure 2, RL agents can take an action $a_1$ and receive a reward of 0 during meta-testing. After observing an action $a_1$, the posterior belief will be $p\left(\kappa_1, \mu_1 \,\middle|\, \bar{b}_t\right) = 1$, which is contradictory because the reward function of $a_1$ is 1 in $\kappa_1$. To support feasible Bayesian belief updating, transformed BAMDPs require the RL agents to filter out out-of-distribution episodes and induce the following theorem.

**Theorem 2.** *In a transformed BAMDP $\overline{M}^+$, for each task-dependent behavior policy $[\mu]$ and batch-constrained meta-policy $\bar{\pi}^+$, the data distribution induced by $\bar{\pi}^+$ and $[\mu]$ matches after filtering out out-of-distribution episodes in online adaptation. Besides, the policy evaluation of $\bar{\pi}^+$ in offline meta-training and online adaptation will be asymptotically consistent, as the offline dataset grows.*

In Theorem 2, the proof of consistent policy evaluation is similar to that in offline single-task RL (Fujimoto et al., 2019), whose proof is deferred to Appendix A.3. This theorem indicates that we can design a delicate context mechanism to correct data distribution and guarantee the final performance of online adaptation by maximizing the expected future return during offline meta-training. Moreover, we prove that meta-policies with *Thompson sampling* (Strens, 2000) can filter out out-of-distribution episodes in $\overline{M}^+$ with high probability, as the offline dataset $\overline{\mathcal{D}}^+$ grows (see Appendix A.5).

## 4 GCC: Greedy Context-Based Data Distribution Correction

In this section, we investigate a scheme to address data distribution mismatch in offline meta-RL with few-shot online adaptation. Inspired by the theoretical results discussed in Section 3, we aim to distinguish whether an adaptation episode is in the distribution of the given meta-training dataset,

then meta-policies with Thompson sampling (Strens, 2000) can utilize in-distribution context to infer how to solve meta-testing tasks. However, in practice, accurate out-of-distribution quantification is challenging (Yu et al., 2021; Wang et al., 2021). For example, we conduct a popular empirical measure of offline RL, uncertainty estimation via an ensemble (Yu et al., 2020c; Kidambi et al., 2020), and find that such quantification works poorly in the offline meta-RL (see Section 5.1). To address this challenge, we introduce a novel *Greedy Context-based data distribution Correction* (GCC), which elaborates a greedy online adaptation mechanism and can directly combine with off-the-shelf offline meta-RL algorithms. GCC consists of two main components: (i) an off-the-shelf context-based offline meta-training method which extracts meta-knowledge from a given multi-task dataset, and (ii) a greedy context-based online adaptation that realizes a selective context mechanism to infer how to solve meta-testing tasks. The whole algorithm is illustrated in Algorithm 1.

## 4.1 CONTEXT-BASED OFFLINE META-TRAINING

To support effective offline meta-training, we employ a state-of-the-art off-the-shelf context-based algorithm, i.e., FOCAL (Li et al., 2020), which follows the algorithmic framework of a popular meta-RL approach, PEARL (Rakelly et al., 2019). In this meta-training paradigm, task identification $\kappa$ is modeled by a latent task embedding $z$. GCC meta-trains a context encoder $q(z|\boldsymbol{c})$, a policy $\pi(a|s,z)$, and a value function $Q(s,a,z)$, where $\boldsymbol{c}$ is the context information including states, actions, rewards, and next states. The encoder $q$ infers a task belief about the latent task variable $z$ based on the received context. We use $q(z)$ to denote the prior distribution for $\boldsymbol{c} = \emptyset$. The policy $\pi$ and value function $Q$ are conditioned on the latent task variable $z$, in which the representation of $z$ can be end-to-end trained on the RL losses of $\pi$ or $Q$. In addition to the gradient from $\pi$ or $Q$, recent offline meta-RL (Li et al., 2020; Yuan & Lu, 2022) also uses a contrastive loss to help the representation of $z$ distinguish different tasks. We argue that meta-training the inference network $q$ within a given dataset implicitly incorporates the information about offline data collection into the belief, as discussed in Definition 3. Formally, in the offline setting, the prior distribution $q(z)$ can approximately present a distribution of the latent task variable $z$ over meta-training tasks, i.e., a sample $\tilde{z}$ drawn from $q(z)$ is equivalent to sampling a task during meta-training and inferring $\tilde{z}$ using the offline context from the offline dataset. During meta-testing, few-shot out-of-distribution episodes may confuse the context encoder $q(z|\boldsymbol{c})$ and RL agents need to use in-distribution context to provide reliable task beliefs.

## 4.2 GREEDY CONTEXT-BASED META-TESTING

GCC is a context-based meta-RL algorithm (Rakelly et al., 2019), whose adaptation protocol follows the framework of *Thompson sampling* (Strens, 2000). The RL agent will iteratively update task belief by interacting with the environment and improve the meta-policy based on the "task hypothesis" sampled from the current beliefs. This adaptation paradigm is generalized from Bayesian inference (Thompson, 1933) and has a solid background in RL theory (Agrawal & Goyal, 2012). To ensure feasible Bayesian belief updating, we adopt a heuristic of offline RL (Fujimoto et al., 2019), i.e., few-shot out-of-distribution episodes generated by an offline-learned meta-policy usually have lower returns since offline training paradigm does not well-optimize meta-policies on out-of-distribution states. Its contrapositive statement is that episodes with higher returns have a higher probability of being in-distribution and the true return of episodes can be a good out-of-distribution quantification. In this way, GCC contains two steps to perform a greedy posterior belief update at each iteration: (i) a diverse sampling of latent task variables, and (ii) a greedy context mechanism that selects a task embedding with higher return to update the belief. For each iteration $t$, denote the current task belief by $b_t$, the meta-testing task by $\kappa_{test}$, and the number of belief updating iterations by $n_{it}$.

**A diverse sampling of latent task variables** will generate $n_t^z$ candidates of the task embedding $z_t$, denoting by $\mathcal{Z}_t = \left\{ z_t^i \right\}_{i=0}^{n_t^z - 1}$, to provide various policies $\pi\left( a \left| s, z_t^i \right. \right)$ for the subsequent context selection mechanism. Due to the contrastive loss applied to the representation of $z$ (Li et al., 2020), a closer task embedding $z$ may yield a more similar policy. Hence, to encourage the diversity of policies, each candidate $z_t^i \in \mathcal{Z}_t$ is designed by

$$z_t^i = \underset{\tilde{z} \in \widetilde{\mathcal{Z}}_t^i}{\arg\max} \left( \min \left( \min_{j < i} \left\| \tilde{z} - z_t^j \right\|_2, \min_{t' < t, k < n_{t'}^z} \left\| \tilde{z} - z_{t'}^k \right\|_2 \right) \right) \text{ and } \widetilde{\mathcal{Z}}_t^i = \left\{ \tilde{z}_t^{i,u} \sim b_t \right\}_{u=0}^{n_{\tilde{z}} - 1}, \quad (7)$$

in which $z_t^i$ aims to maximize the minimum distance to the previous embeddings among $n_{\tilde{z}}$ samples. This greedy method is similar to *Farthest-Point Clustering* (Gonzalez, 1985), a two-approximation algorithm for an NP-hard problem *Minimax Facility Location* (Fowler et al., 1981), which seeks a set of locations to minimize the maximum distance from other facilities to the set.

**A greedy selective context mechanism** will select a latent task variable $z_t \in \mathcal{Z}_t$ with higher return to update the task belief $b_t$. To evaluate the return of each $z_t^i$, GCC utilizes the policy $\pi\left(a \,|\, s, z_t^i\right)$ to draw $n_e$ episodes in $\kappa_{test}$, denoting by $\mathcal{E}_t^i = \left\{e_t^{i,j}\right\}_{j=0}^{n_e-1}$. The policy evaluation of $z_t^i$ can be approximated by sampling:

$$\mathcal{J}_{\kappa_{test}}\left(z_t^i\right) = \mathcal{J}_{\kappa_{test}}\left(\pi\left(a \,|\, s, z_t^i\right)\right) \approx \widetilde{\mathcal{J}}_{\kappa_{test}}\left(\mathcal{E}_t^i\right) = \frac{1}{n_e}\sum_{j=0}^{n_e-1}\left(\sum_{k=0}^{H-1}r_k\left(e_t^{i,j}\right)\right), \qquad (8)$$

where $\widetilde{\mathcal{J}}_{\kappa_{test}}\left(\mathcal{E}_t^i\right)$ is the average return of episodes $\mathcal{E}_t^i$ and $r_k\left(e_t^{i,j}\right)$ is the reward of $k$-th step in an episode $e_t^{i,j}$. The task belief update in GCC consists of two phases: (i) an initial stage and (ii) an iterative optimization process. In the initial stage, GCC aims to find a reliable initial task inference $z_0$ using a large amount of $n_0^z$ diverse candidates, i.e., the initial context is $\boldsymbol{c}_0 = \arg\max_{\mathcal{E}_0 \in \{\mathcal{E}_0^i\}}\widetilde{\mathcal{J}}_{\kappa_{test}}\left(\mathcal{E}_0\right)$, maintaining the corresponding task embedding $z_0$, and deriving the posterior belief $b_1 = q(z|\boldsymbol{c}_0)$. In the following iterations, GCC utilizes an iterative optimization method to maximize final performance during few-shot online adaptation, i.e., when $t > 1$, let $n_t^z = 1$ and if $\widetilde{\mathcal{J}}_{\kappa_{test}}\left(\mathcal{E}_t^0\right) > \widetilde{\mathcal{J}}_{\kappa_{test}}\left(\boldsymbol{c}_{t-1}\right)$, we have $\boldsymbol{c}_t = \mathcal{E}_t^0$ and update the posterior belief $b_{t+1} = q(z| \cup_{t' \leq t}\boldsymbol{c}_{t'})$, otherwise $\boldsymbol{c}_t = \boldsymbol{c}_{t-1}$ and keep the belief $b_{t+1} = b_t$. The final policy $\pi\left(a\,|\,s, z_t\right)$ will depend on the optimal task embedding $z_t$ with the highest return.

---

**Algorithm 1** GCC: Greedy Context-based data distribution Correction

---

1: **Require:** An offline multi-task dataset $\mathcal{D}^+$, a meta-testing task $\kappa_{test} \sim p(\kappa)$, the number of iterations $n_{it}, n_0^z$, and a context-based offline meta-training algorithm $\mathbb{A}$ (i.e., FOCAL)
2: Randomly initialize a context encoder $q(z|\boldsymbol{c})$, a policy $\pi(a|s, z)$, and a value function $Q(s, a, z)$
3: Offline meta-train $q$, $\pi$, and $Q$ given an algorithm $\mathbb{A}$ and a dataset $\mathcal{D}^+$     ▷ *Offline meta-training*
4: Generate a prior task distribution $q(z)$ using the dataset $\mathcal{D}^+$     ▷ *Start online meta-testing*
5: Collect diverse adaptation episodes $\left\{\mathcal{E}_0^i\right\}_{i=0}^{n_0^z-1}$ using $q(z)$ and $\pi$ in $\kappa_{test}$
6: Compute the greedy context $\boldsymbol{c}_0$, task embedding $z_0$, and posterior belief $b_1$     ▷ *An initial stage*
7: **for** $t = 1 \ldots n_{it} - 1$ **do**     ▷ *An iterative optimization process*
8:      Collect a diverse episode $\mathcal{E}_t^0$ using $b_t$ and $\pi(a|s, z_t)$ in $\kappa_{test}$
9:      Compute the greedy context $\boldsymbol{c}_t$ and posterior belief $b_{t+1}$
10: Derive the final policy $\pi_{\text{out}}\left(a\,|\,s, z_t\right)$ with the optimal task embedding $z_t$
11: **Return:** $\pi_{\text{out}}$

---

## 5 EXPERIMENTS

In this section, we first study a didactic example to analyze the out-of-distribution problem, and show how GCC alleviates this problem by its greedy selective mechanism. Then we conduct large-scale experiments on Meta-World ML1(Yu et al., 2020a), a popular meta-RL benchmark that consists of 50 robot arm manipulation task sets. Finally, we perform ablation studies to analyze GCC's sensitivity to hyper-parameter settings and dataset qualities. Following FOCAL (Li et al., 2020), we use expert-level datasets sampled by policies trained with SAC on the corresponding tasks. We compare against FOCAL (Li et al., 2020) and MACAW (Mitchell et al., 2021), as well as their online adaptation variants. We also compare against BOReL (Dorfman et al., 2021), an offline Meta-RL algorithm build upon VariBAD (Zintgraf et al., 2019). For fair comparison, we evaluate a variant of BOReL that does not utilize oracle reward functions, as introduced in the original paper(Dorfman et al., 2021). FOCAL is built upon PEARL (Rakelly et al., 2019) and uses contrastive losses to learn context embeddings, while MACAW is a MAML-based (Finn et al., 2017) algorithm and incorporates AWR (Peng et al., 2019). Both FOCAL and MACAW are originally proposed for the offline adaptation settings (i.e., with expert context). For online adaptation, we use online experiences instead of expert contexts, and adopt the adaptation protocol of PEARL and MAML, respectively. Evaluation results are averaged over six random seeds, and variance is measured by 95% bootstrapped confidence interval. Detailed hyper-parameter settings are deferred to Appendix B.

### 5.1 DIDACTIC EXAMPLE

We introduce Point-Robot, a simple 2D navigation task set commonly used in meta-RL (Rakelly et al., 2019; Zhang et al., 2021). Figure 3(a) illustrates the distribution mismatch between offline

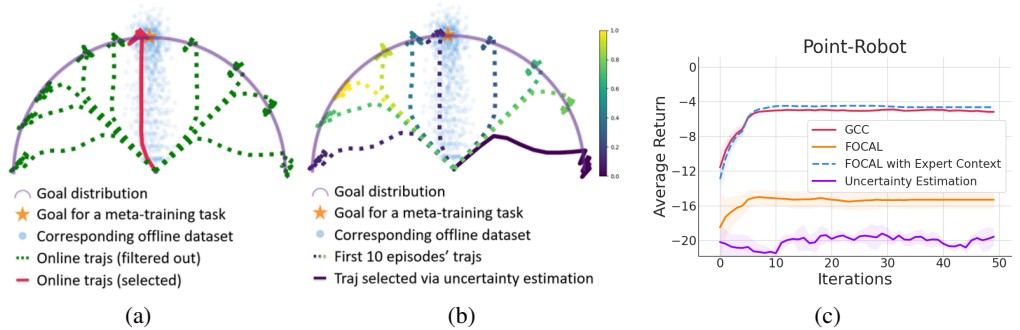

(a)      (b)      (c)

Figure 3: (a) Illustration of data distribution mismatch between offline meta-training (blue) and online adaptation (green and red trajectories). (b) The agent fails to identify in-distribution trajectories via uncertainty estimation. Trajectories are colored with corresponding normalized uncertainty estimation. (c) Adaptation performance of GCC, FOCAL, FOCAL with expert context, and selecting trajectories with uncertainty estimation.

meta-training and online adaptation, as the dataset is collected by task-dependent behavior policies. As a consequence, directly performing adaptation with the online collected trajectories lead to poor adaptation performance (see FOCAL in Figure 3(c)). Figure 3(b) shows that existing methods that estimate uncertainty with ensembles (Kidambi et al., 2020) cannot correctly detect OOD data, as there is a large error on uncertainty estimation. GCC fixes this problem by greedily selecting trajectories. As shown in Figure 3(a), at the end of the initial stage, GCC only updates its belief with the red trajectory as it has the highest return. After the initial stage, GCC iteratively optimizes the posterior belief to get the final policy. As shown in Figure 3(c), GCC achieves comparable performance to FOCAL with expert context and significantly outperforms FOCAL with online adaptation. Visualization of GCC's and FOCAL's adaptation trajectories after the initial stage are deferred to Appendix D. Further analysis and visualization about the uncertainty estimation methods are deferred to Appendix C.

## 5.2 MAIN RESULTS

We evaluate on Meta-World ML1(Yu et al., 2020a), a popular meta-RL benchmark that consists of 50 robot arm manipulation task sets. Each task set consists of 50 tasks with different goals. For each task set, we use 40 tasks as meta-training tasks, and remain the other 10 tasks as meta-testing tasks. As shown in Table 1, GCC significantly outperforms baselines under the online context setting. With expert contexts, FOCAL and MACAW both achieve reasonable performance. GCC achieves better or comparable performance to baselines with expert contexts, which implies that expert contexts may not be necessary for offline meta-RL. Under online contexts, FOCAL fails due to the data distribution mismatch between offline training and online adaptation. MACAW has the ability of online fine-tuning as it is based on MAML, but it also suffers from the distribution mismatch problem, and online fine-tuning can hardly improve its performance within a few adaptation episodes. BOReL fails on most of the tasks, as BOReL without oracle reward functions will also suffer from the distribution mismatch problem, which is consistent with the results in the original paper. We also test the uncertainty estimation method on several representative tasks, and it performs similarly to FOCAL. Detailed results are deferred to Table 5 in Appendix E.1.

Table 1: Algorithms' normalized scores averaged over 50 Meta-World ML1 task sets. Scores are normalized by expert-level policy return.

| GCC | FOCAL | MACAW | FOCAL with Expert Context | MACAW with Expert Context | BOReL |
|---|---|---|---|---|---|
| **0.73** ± 0.07 | 0.53 ± 0.1 | 0.18 ± 0.1 | 0.67 ± 0.07 | 0.68 ± 0.07 | 0.04 ± 0.01 |

Table 2 shows algorithms' performance on 20 representative Meta-World ML1 task sets, as well a sparse-reward version of Point-Robot and Cheetah-Vel, which are popular meta-RL tasks (Li et al., 2020). GCC achieves remarkable performance in most tasks and may fail in some hard tasks as offline meta-training is difficult. We also find that GCC achieves better or comparable performance to baselines with expert contexts on 33 out of the 50 task sets. Detailed algorithm performance on all 50 tasks as well as comparison to baselines with expert contexts are deferred to Appendix E.2. We further perform ablation studies on hyper-parameter settings and dataset qualities, and results are deferred to Appendix F. Results demonstrate that GCC is generally robust to the choice of hyper-parameters, and performs well on medium-quality datasets.

Table 2: Performance on example tasks, a bunch of Meta-World ML1 tasks with normalized scores. For Meta-World tasks, "-V2" is omitted for brevity.

| Example Env | GCC | FOCAL | MACAW | BOReL |
|---|---|---|---|---|
| Coffee-Push | **1.26** ± 0.13 | 0.66 ± 0.07 | 0.01 ± 0.01 | 0.00 ± 0.00 |
| Faucet-Close | **1.12** ± 0.01 | 1.06 ± 0.02 | 0.07 ± 0.01 | 0.13 ± 0.03 |
| Faucet-Open | **1.05** ± 0.02 | 1.01 ± 0.02 | 0.08 ± 0.04 | 0.12 ± 0.05 |
| Door-Close | **0.99** ± 0.00 | 0.97 ± 0.01 | 0.00 ± 0.00 | 0.37 ± 0.19 |
| Drawer-Close | **0.99** ± 0.02 | **0.96** ± 0.04 | 0.53 ± 0.50 | 0.00 ± 0.00 |
| Door-Lock | **0.97** ± 0.01 | 0.90 ± 0.02 | 0.25 ± 0.11 | 0.14 ± 0.00 |
| Plate-Slide-Back | **0.96** ± 0.02 | 0.58 ± 0.06 | 0.21 ± 0.17 | 0.01 ± 0.00 |
| Dial-Turn | **0.91** ± 0.05 | 0.84 ± 0.09 | 0.00 ± 0.00 | 0.00 ± 0.00 |
| Handle-Press | **0.88** ± 0.05 | **0.87** ± 0.02 | 0.28 ± 0.10 | 0.01 ± 0.00 |
| Hammer | **0.84** ± 0.06 | 0.59 ± 0.07 | 0.10 ± 0.01 | 0.09 ± 0.01 |
| Button-Press | **0.74** ± 0.08 | **0.68** ± 0.14 | 0.02 ± 0.01 | 0.01 ± 0.01 |
| Push-Wall | **0.71** ± 0.15 | 0.43 ± 0.06 | 0.23 ± 0.18 | 0.00 ± 0.00 |
| Hand-Insert | **0.63** ± 0.04 | 0.29 ± 0.07 | 0.02 ± 0.01 | 0.00 ± 0.00 |
| Peg-Unplug-Side | **0.56** ± 0.07 | 0.19 ± 0.09 | 0.00 ± 0.00 | 0.00 ± 0.00 |
| Bin-Picking | 0.53 ± 0.16 | 0.31 ± 0.21 | **0.66** ± 0.11 | 0.00 ± 0.00 |
| Soccer | **0.44** ± 0.04 | 0.11 ± 0.03 | **0.38** ± 0.31 | 0.04 ± 0.02 |
| Coffee-Pull | **0.40** ± 0.05 | 0.23 ± 0.04 | 0.19 ± 0.12 | 0.00 ± 0.00 |
| Pick-Place-Wall | 0.28 ± 0.12 | 0.09 ± 0.04 | **0.39** ± 0.25 | 0.00 ± 0.00 |
| Pick-Out-Of-Hole | 0.26 ± 0.25 | 0.16 ± 0.16 | **0.59** ± 0.06 | 0.00 ± 0.00 |
| Handle-Pull-Side | **0.14** ± 0.04 | **0.13** ± 0.09 | 0.00 ± 0.00 | 0.00 ± 0.00 |
| Cheetah-Vel | **-171.5** ± 22.00 | -287.7 ± 30.6 | -234.0 ± 23.5 | -301.4 ± 36.8 |
| Point-Robot | **-5.10** ± 0.26 | -15.38 ± 0.95 | -14.61 ± 0.98 | -17.28 ± 1.16 |
| Point-Robot-Sparse | **7.78** ± 0.64 | 0.83 ± 0.37 | 0.00 ± 0.00 | 0.00 ± 0.00 |

## 6 RELATED WORK

In the literature, offline meta-RL methods utilize a context-based (Rakelly et al., 2019) or gradient-based (Finn et al., 2017) meta-RL framework to solve new tasks with few-shot adaptation. They utilize the techniques of contrastive learning (Li et al., 2020; Yuan & Lu, 2022), more expressive power (Mitchell et al., 2021), or reward relabeling (Dorfman et al., 2021; Pong et al., 2022) with various popular offline single-task RL tricks, i.e., using KL divergence (Wu et al., 2019; Peng et al., 2019; Nair et al., 2020) or explicitly constraining the policy to be close to the dataset (Fujimoto et al., 2019; Zhou et al., 2020). However, these methods always require extra information for fast adaptation, such as offline context for testing tasks (Li et al., 2020; Mitchell et al., 2021; Yuan & Lu, 2022), oracle reward functions (Dorfman et al., 2021), or free interactions without reward supervision (Pong et al., 2022). To address the challenge, we propose GCC, a greedy context mechanism with theoretical motivation, to perform effective online adaptation without requiring additional information.

The concepts of *distribution shift in z-space* in Pong et al. (2022) and *MDP ambiguity* in Dorfman et al. (2021) are related to the data distribution mismatch proposed in this paper. We reveal that the task-dependent behavior policies will induce different reward and transition distribution between offline meta-training and online adaptation, in which this mismatch combined with "policy" distribution shift in offline single-task RL (Levine et al., 2020) are two essential factors in the phenomenon of distribution shift in z-space (Pong et al., 2022). After filtering out these out-of-distribution data, GCC can maintain an overall belief about the task with behavior policies to address MDP ambiguity.

## 7 CONCLUSION

This paper formalizes data distribution mismatch in offline meta-RL with online adaptation and introduces GCC, a novel context-based online adaptation approach. Inspired by theoretical implications, GCC adopts a greedy context mechanism to filter out out-of-distribution with lower return for online data correction. We demonstrate that GCC can perform accurate task inference and achieve state-of-the-art performance on Meta-World ML1 benchmark with 50 tasks. Compared to offline adaptation baselines with expert context, GCC also performs better or comparably, suggesting that offline context may not be necessary for the testing environments. One potential future direction is to extend GCC to gradient-based online adaptation methods with data distribution correction and to deal with task distributional shift problem in offline meta-RL (which is not considered in GCC).

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

# A  THEORY

## A.1  BACKGROUND

Throughout this paper, for a given non-negative integer $N \in \mathbb{Z}_+$, we use $[N]$ to denote the set $\{0, 1, \ldots, N-1\}$. For any object that is a function of/distribution over $\mathcal{S}, \mathcal{S} \times \mathcal{A}, \mathcal{S} \times \mathcal{A} \times \mathcal{S}$, or $\mathcal{S} \times \mathcal{A} \times \mathcal{R}$, we will treat it as a vector whenever convenient.

### A.1.1  FINITE-HORIZON SINGLE-TASK RL

In single-task RL, an agent interacts with a Markov Decision Process (MDP) to maximize its cumulative reward (Sutton & Barto, 2018). A finite-horizon MDP is defined as a tuple $M = (\mathcal{S}, \mathcal{A}, \mathcal{R}, H, P, R)$ (Zintgraf et al., 2019; Du et al., 2019), where $\mathcal{S}$ is the state space, $\mathcal{A}$ is the action space, $\mathcal{R}$ is the reward space, $H \in \mathbb{Z}_+$ is the planning horizon, $P : \mathcal{S} \times \mathcal{A} \to \Delta(\mathcal{S})$ is the transition function which takes a state-action pair and returns a distribution over states, and $R : \mathcal{S} \times \mathcal{A} \to \Delta(\mathcal{R})$ is the reward distribution. In particular, we consider finite state, action, and reward spaces in the theoretical analysis, i.e., $|\mathcal{S}| < \infty, |\mathcal{A}| < \infty, |\mathcal{R}| < \infty$. Without loss of generality, we assume a fixed initial state $s_0$[1]. A policy $\pi : \mathcal{S} \to \Delta(\mathcal{A})$ prescribes a distribution over actions for each state. The policy $\pi$ induces a (random) $H$-horizon trajectory $\tau_H^\pi = (s_0, a_0, r_0, s_1, a_1, \ldots, s_{H-1}, a_{H-1}, r_{H-1})$, where $a_0 \sim \pi(s_0), r_0 \sim R(s_0, a_0), s_1 \sim P(s_0, a_0), a_1 \sim \pi(s_1)$, etc. To streamline our analysis, for each $h \in [H]$, we use $\mathcal{S}_h \subseteq \mathcal{S}$ to denote the set of states at $h$-th timestep, and we assume $\mathcal{S}_h$ do not intersect with each other. To simplify notation, we assume the transition from any state in $S_{H-1}$ and any action to the initial state $s_0$, i.e., $\forall s \in S_{H-1}, a \in \mathcal{A}$, we have $P(s_0|s, a) = 1$[2]. We also assume $r_t \in [0, 1], \forall t \in [H]$ almost surely. Denote the probability of $\tau_H$:

$$p(\tau_H^\pi) = \left( \prod_{t \in [H]} \pi(a_t|s_t) \cdot R(r_t|s_t, a_t) \right) \prod_{t \in [H-1]} P(s_{t+1}|s_t, a_t). \tag{9}$$

For any policy $\pi$, we define a value function $V_\pi : \mathcal{S} \to \mathbb{R}$ as: $\forall h \in [H], \forall s \in \mathcal{S}_h$,

$$V_\pi(s) = \mathbb{E}_{s_h = s, a_t \sim \pi(\cdot|s_t), r_t \sim R(\cdot|s_t, a_t), s_{t+1} \sim P(\cdot|s_t, a_t)} \left[ \sum_{t=h}^{H-1} r_t \right] \tag{10}$$

$$= \begin{cases} \sum_{a \in \mathcal{A}} \pi(a|s) \cdot \mathbb{E}_{r \sim R(\cdot|s,a)}[r], & \text{if } h = H-1, \\[2ex] \sum_{a \in \mathcal{A}} \pi(a|s) \left( \mathbb{E}_{r \sim R(\cdot|s,a)}[r] + \sum_{s' \in \mathcal{S}_{h+1}} P(s'|s,a) V_\pi(s') \right), & \text{otherwise,} \end{cases}$$

and a visitation distribution of $\pi$ is defined by $\rho_\pi(\cdot) : \Delta(\mathcal{S})$ which is $\forall h \in [H], \forall s \in \mathcal{S}_h$,

$$\rho_\pi(s) = \begin{cases} \dfrac{1}{H}, & \text{if } h = 0 \text{ and } s = s_0, \\[2ex] \sum_{\tilde{s} \in \mathcal{S}_{h-1}, \tilde{a} \in \mathcal{A}} \rho_\pi(\tilde{s}) \cdot \pi(\tilde{a}|\tilde{s}) \cdot P(s|\tilde{s}, \tilde{a}), & \text{if } h > 0, \\[2ex] 0, & \text{otherwise,} \end{cases} \tag{11}$$

and $\forall s \in \mathcal{S}, a \in \mathcal{A}, r \in \mathcal{R}$,

$$\rho_\pi(s, a) = \rho_\pi(s) \cdot \pi(a|s) \quad \text{and} \quad \rho_\pi(s, a, r) = \rho_\pi(s) \cdot \pi(a|s) \cdot R(r|s, a). \tag{12}$$

The expected total reward induced by policy $\pi$, i.e., the policy evaluation of $\pi$, is defined by

$$\mathcal{J}_M(\pi) = V_\pi(s_0) = H \sum_{s \in \mathcal{S}, a \in \mathcal{A}} \rho_\pi(s, a) \cdot \mathbb{E}_{r \sim R(\cdot|s,a)}[r]. \tag{13}$$

The goal of RL is to find a policy $\pi$ that maximizes its expected return $\mathcal{J}(\pi)$.

---

[1]Some papers assume the initial state is sampled from a distribution $P_1$. Note this is equivalent to assuming a fixed initial state $s_0$, by setting $P(s_0, a) = P_1$ for all $a \in \mathcal{A}$ and now our state $s_1$ is equivalent to the initial state in their assumption.

[2]The transition from the state in $S_{H-1}$ does not affect learning in the finite-horizon MDP $M$.

A.1.2 OFFLINE FINITE-HORIZON SINGLE-TASK RL

We consider the offline finite-horizon single-task RL setting, that is, a learner only has access to a dataset $\mathcal{D}$ consisting of $K$ trajectories $\left\{\left(s_t^k, a_t^k, r_t^k\right)\right\}_{t \in [H]}^{k \in [K]}$ (i.e., $|\mathcal{D}| = KH$ tuples) and is not allowed to interact with the environment for additional online explorations. The data can be collected through multi-source logging policies and denote the unknown behavior policy $\mu$. Similar with related work (Ren et al., 2021; Yin et al., 2020; Yin & Wang, 2021; Yin et al., 2021; Shi et al., 2022), we assume that $\mathcal{D}$ is collected through interacting $K$ i.i.d. episodes using policy $\mu$ in $M$. Define the reward and transition distribution of data collection with $\mu$ in $M$ by $\mathbb{P}_M$ (Jin et al., 2021), i.e., $\forall t \in [H]$ in each episode,

$$\mathbb{P}_M\left(r_t, s_{t+1} \mid s_t, a_t\right) = R^M\left(r_t \mid s_t, a_t\right) \cdot P^M\left(s_{t+1} \mid s_t, a_t\right), \tag{14}$$

where the action $a_t$ is drawn from a behavior policy $\mu$. Denote a dataset collected following the i.i.d. data collecting process, i.e., $\mathcal{D} \sim (\mathbb{P}_M, \mu)$ is an i.i.d. dataset. Note that the offline dataset $\mathcal{D}$ can be narrow collected by some behavior policy $\mu$ and a large amount of state-action pairs are not contained in $\mathcal{D}$. These unseen state-action pairs will be erroneously estimated to have unrealistic values, called a phenomenon *extrapolation error* (Fujimoto et al., 2019). To overcome extrapolation error in policy learning of finite MDPs, Fujimoto et al. (Fujimoto et al., 2019) introduces batch-constrained RL, which restricts the action space in order to force policy selection of an agent with respect to a subset of the given data. Thus, define a batch-constrained policy set is

$$\Pi^{\mathcal{D}} = \left\{\pi \mid \pi(a|s) = 0 \text{ whenever } (s, a) \notin \mathcal{D}\right\}, \tag{15}$$

where denoting $(s, a) \in \mathcal{D}$ if there exists a trajectory containing $(s, a)$ in the dataset $\mathcal{D}$, and similarly for $s \in \mathcal{D}$, $(s, a, r) \in \mathcal{D}$, or $(s, a, r, s') \in \mathcal{D}$. The batch-constrained policy set $\Pi^{\mathcal{D}}$ consists of the policies that for any state $s$ observed in the dataset $\mathcal{D}$, the agent will not select an action outside of the dataset. Thus, for any batch-constrained policy $\pi \in \Pi^{\mathcal{D}}$, define the approximate value function $V_\pi^{\mathcal{D}} : \mathcal{S} \to \mathbb{R}$ estimated from $\mathcal{D}$ (Fujimoto et al., 2019; Liu et al., 2020) as: $\forall h \in [H], \forall s \in \mathcal{S}_h$,

$$V_\pi^{\mathcal{D}}(s) = \mathbb{E}_{s_h=s, a_t \sim \pi(\cdot|s_t), (s_t, a_t, r_t, s_{t+1}) \sim \mathcal{D}} \left[\sum_{t=h}^{H-1} r_t\right] \tag{16}$$

$$= \begin{cases} \displaystyle\sum_{a \in \mathcal{A}} \pi(a|s) \mathbb{E}_{(s,a,r) \in \mathcal{D}}[r], & \text{if } h = H - 1, \\ \displaystyle\sum_{a \in \mathcal{A}} \pi(a|s) \mathbb{E}_{(s,a,r,s') \in \mathcal{D}}\left[r + V_\pi^{\mathcal{D}}(s')\right], & \text{otherwise,} \end{cases} \tag{17}$$

which is called Approximate Dynamic Programming (ADP) (Bertsekas & Tsitsiklis, 1995) and such methods take sampling data as input and approximate the value-function (Liu et al., 2020; Chen & Jiang, 2019). In addition, define the approximate policy evaluation of $\pi$ estimated from $\mathcal{D}$ as

$$\mathcal{J}_{\mathcal{D}}(\pi) = V_\pi^{\mathcal{D}}(s_0). \tag{18}$$

The offline RL literature (Fujimoto et al., 2019; Liu et al., 2020; Chen & Jiang, 2019; Kumar et al., 2019; 2020) aims to utilize approximate expected total reward $\mathcal{J}_{\mathcal{D}}(\pi)$ with various conservatism regularizations (i.e., policy constraints, policy penalty, uncertainty penalty, etc.) (Levine et al., 2020) to find a good policy within a batch-constrained policy set $\Pi^{\mathcal{D}}$.

Similar to offline finite-horizon single-task RL theory (Ren et al., 2021; Yin et al., 2020; Yin & Wang, 2021; Yin et al., 2021; Shi et al., 2022), define

$$d_\mu^M = \min\left\{\rho_\mu(s, a) \mid \rho_\mu(s, a) > 0, \forall s \in \mathcal{S}, a \in \mathcal{A}\right\}, \tag{19}$$

which is the minimal visitation state-action distribution induced by the behavior policy $\mu$ in $M$ and is an intrinsic quantity required by theoretical offline learning (Yin et al., 2020). Note that, different from recent offline episodic RL theory (Ren et al., 2021; Yin et al., 2020; Yin & Wang, 2021; Yin et al., 2021; Shi et al., 2022), we do not assume any weak or uniform converage assumption in the dataset because we focus on the policy evaluation of all batch-constrained policies in $\Pi^{\mathcal{D}}$ rather than the optimal policy in the MDP $M$.

### A.1.3 STANDARD META-RL

The goal of meta-RL (Finn et al., 2017; Rakelly et al., 2019) is to train a meta-policy that can quickly adapt to new tasks using $N$ adaptation episodes. The standard meta-RL setting deals with a distribution $p(\kappa)$ over MDPs, in which each task $\kappa_i$ sampled from $p(\kappa)$ presents a finite-horizon MDP (Zintgraf et al., 2019; Du et al., 2019). $\kappa_i$ is defined by a tuple $(\mathcal{S}, \mathcal{A}, \mathcal{R}, H, P^{\kappa_i}, R^{\kappa_i})$, including state space $\mathcal{S}$, action space $\mathcal{A}$, reward space $\mathcal{R}$, planning horizon $H$, transition function $P^{\kappa_i}(s'|s, a)$, and reward function $R^{\kappa_i}(r|s, a)$. Denote $\mathcal{K}$ is the space of task $\kappa_i$. In this paper, we assume dynamics function $P$ and reward function $R$ may vary across tasks and share a common structure. The meta-RL algorithms repeatedly sample batches of tasks to train a meta-policy. In the meta-testing, agents aim to rapidly adapt a good policy for new tasks drawn from $p(\kappa)$.

**POMDPs.** We can formalize the meta-RL with few-shot adaptation as a specific finite-horizon Partially Observable Markov Decision Process (POMDP), which is defined by a tuple $\widehat{M} = \left(\widehat{\mathcal{S}}, \mathcal{A}, \mathcal{R}, \Omega, \widehat{H}, \widehat{P}, \widehat{P}_0, O, \widehat{R}\right)$, where $\widehat{\mathcal{S}} = \mathcal{S} \times \mathcal{K}$ is the state space, $\mathcal{A}$ and $\mathcal{R}$ are the same action and reward spaces as the finite-horizon MDP $M$ defined in Appendix A.1.1, respectively, $\Omega = \mathcal{S}$ is the observation space, $\widehat{H} = N \times H$ is the planning horizon which represents $N$ adaptation episodes for a single meta-RL MDP $\kappa_i$, as discussed in Zintgraf et al. (Zintgraf et al., 2019), $\widehat{P} : \widehat{\mathcal{S}} \times \mathcal{A} \to \Delta\left(\widehat{\mathcal{S}}\right)$ is the transition function: $\forall \hat{s}, \hat{s}' \in \widehat{\mathcal{S}}, a \in \mathcal{A}$, where denoting $\hat{s} = (s, \kappa_i)$ and $\hat{s}' = (s', \kappa_j)$,

$$\widehat{P}\left(\hat{s}'|\hat{s}, a\right) = \begin{cases} P^{\kappa_i}(s'|s, a), & \text{if } \kappa_i = \kappa_j, \\ 0, & \text{otherwise,} \end{cases} \tag{20}$$

$\widehat{P}_0 : \Delta\left(\widehat{\mathcal{S}}\right)$ is the initial state distribution: $\forall \hat{s} = (s, \kappa_i) \in \widehat{\mathcal{S}}$,

$$\widehat{P}_0\left(\hat{s}\right) = \begin{cases} p(\kappa_i), & \text{if } s = s_0, \\ 0, & \text{otherwise,} \end{cases} \tag{21}$$

$O : \widehat{\mathcal{S}} \to \Delta\left(\Omega\right)$ is the observation probability distribution conditioned on a state: $\forall \hat{s} = (s, \kappa_i) \in \widehat{\mathcal{S}}, o \in \Omega$,

$$O\left(o|\hat{s}\right) = \begin{cases} 1, & \text{if } o = s, \\ 0, & \text{otherwise,} \end{cases} \tag{22}$$

and $\widehat{R} : \widehat{\mathcal{S}} \times \mathcal{A} \to \Delta\left(\mathcal{R}\right)$ is the reward distribution: $\forall \hat{s} = (s, \kappa_i) \in \widehat{\mathcal{S}}, a \in \mathcal{A}, r \in \mathcal{R}$,

$$\widehat{R}\left(r|\hat{s}, a\right) = R^{\kappa_i}(r|s, a). \tag{23}$$

Denote context $c_t = (a_t, r_t, s_{t+1})$ as an experience collected at timestep $t$, and $c_{:t} = \langle s_0, c_0, \dots, c_{t-1}\rangle^3 \in \mathcal{C}_t \equiv \Omega \times (\mathcal{A} \times \mathcal{R} \times \Omega)^t$ indicates all experiences collected during $t$ timesteps. Note that $t$ may be larger than $H$, and when it is the case, $c_{:t}$ represents experiences collected across episodes in the single meta-RL MDP $\kappa_i$. Denote the entire context space $\mathcal{C} = \bigcup_{t=0}^{\widehat{H}-1} \mathcal{C}_t$ and a meta-policy $\hat{\pi} : \mathcal{C} \to \Delta\left(\mathcal{A}\right)$ (Wang et al., 2016; Duan et al., 2016) prescribes a distribution over actions for each context. The goal of meta-RL is to find a meta-policy on history contexts $\hat{\pi}$ that maximizes the expected return within $N$ adaptation episodes:

$$\mathcal{J}_{\widehat{M}}(\hat{\pi}) = \mathbb{E}_{\hat{s}_0 \sim P_0, o_t \sim O(\cdot|s_t), a_t \sim \hat{\pi}(\cdot|c_{:t}), r_t \sim \widehat{R}(\cdot|s_t, a_t), \hat{s}_{t+1} \sim \widehat{P}(\cdot|\hat{s}_t, a_t)} \left[\sum_{t=0}^{\widehat{H}-1} r_t\right] \tag{24}$$

$$= \mathbb{E}_{\kappa_i \sim p(\kappa)} \left[\sum_{j=0}^{N-1} \mathbb{E}_{a_t \sim \hat{\pi}\left(\cdot|c_{:(jH+t)}\right), r_t \sim R^{\kappa_i}(\cdot|s_t, a_t), s_{t+1} \sim P^{\kappa_i}(\cdot|s_t, a_t)} \left[\sum_{t=0}^{H-1} r_t\right]\right]. \tag{25}$$

---

[3] For clarity, we denote $c_{:0}^{\kappa_i} = s_0$.

**BAMDPs.** A Markovian belief state allows a POMDP to be formulated as a Markov decision process where every belief is a state (Cassandra et al., 1994). We can transform the finite-horizon POMDP $\widehat{M}$ to a finite-horizon belief MDP, which is called Bayes-Adaptive MDP (BAMDP) in the literature (Zintgraf et al., 2019; Ghavamzadeh et al., 2015; Dorfman et al., 2021) and is defined by a tuple $M^+ = \left(\mathcal{S}^+, \mathcal{A}, \mathcal{R}, H^+, P^+, P_0^+, R^+\right)$, $\mathcal{S}^+ = \mathcal{S} \times \mathcal{B}$ is the hyper-state space, where $\mathcal{B} = \{p(\kappa|c) \,|c \in \mathcal{C}\}$ is the set of belief states over the meta-RL MDPs, the prior

$$b_0^\kappa = p\left(\kappa|c_{:0}\right) = p(\kappa) \tag{26}$$

is the meta-RL MDP distribution, and $\forall t \in \left[\widehat{H}-1\right], \forall c_{:(t+1)} \in \mathcal{C}$, denoting $b_t^\kappa = p\left(\kappa|c_{:t}\right)$ and

$$b_{t+1}^\kappa = p\left(\kappa|c_{:(t+1)}\right) = p\left(p\left(\kappa|c_{:t}\right)|c_{:(t+1)}\right) = p\left(p\left(\kappa|c_{:t}\right)|s_t, c_t\right) = p\left(b_t^\kappa|s_t, c_t\right) \tag{27}$$

$$\propto p\left(b_t^\kappa, c_t|s_t\right) = p\left(c_t|s_t, b_t^\kappa\right) p\left(b_t^\kappa|s_t\right) = p\left(c_t|s_t, b_t^\kappa\right) b_t^\kappa \tag{28}$$

$$= \mathbb{E}_{\kappa_i \sim b_t^\kappa}\left[R^{\kappa_i}(r_t|s_t, a_t) \cdot P^{\kappa_i}(s_{t+1}|s_t, a_t)\right] \cdot b_t^\kappa \tag{29}$$

is the posterior over the MDPs given the context $c_{:(t+1)}$, $\mathcal{A}, \mathcal{R}$ are the same action space and reward space as the finite-horizon POMDP $\widehat{M}$, respectively, $H^+ = N \times H$ is the planning horizon across adaptation episodes, $P^+ : \mathcal{S}^+ \times \mathcal{A} \times \mathcal{R} \to \Delta\left(\mathcal{S}^+\right)$ is the transition function: $\forall s_t^+, s_{t+1}^+ \in \mathcal{S}^+, a_t \in \mathcal{A}, r_t \in \mathcal{R}$, where denoting $s_t^+ = (s_t, b_t^\kappa)$ and $s_{t+1}^+ = \left(s_{t+1}, \tilde{b}_{t+1}^\kappa\right)$,

$$P^+\left(s_{t+1}^+ \,|s_t^+, a_t, r_t\right) = P^+\left(s_{t+1}, \tilde{b}_{t+1}^\kappa \,|s_t, b_t^\kappa, a_t, r_t\right) \tag{30}$$

$$= P^+\left(s_{t+1} \,|s_t, b_t^\kappa, a_t, r_t\right) P^+\left(\tilde{b}_{t+1}^\kappa \,|s_t, b_t^\kappa, c_t\right) \tag{31}$$

$$= \mathbb{E}_{\kappa_i \sim p(b_t^\kappa|s_t, a_t, r_t)}\left[P^{\kappa_i}(s_{t+1}|s_t, a_t)\right] \cdot \mathbb{1}\left[\tilde{b}_{t+1}^\kappa = p(b_t^\kappa|s_t, c_t)\right], \tag{32}$$

$P_0^+ : \Delta\left(\mathcal{S}^+\right)$ is the initial hyper-state distribution, i.e., a deterministic initial hyper-state is

$$s_0^+ = (s_0, b_0^\kappa) = (s_0, p(\kappa)) \in \mathcal{S}^+, \tag{33}$$

and $R^+ : \mathcal{S}^+ \times \mathcal{A} \to \Delta\left(\mathcal{R}\right)$ is the reward distribution: $\forall s^+ = (s, b^\kappa) \in \mathcal{S}^+, a \in \mathcal{A}, r \in \mathcal{R}$,

$$R^+\left(r|s^+, a\right) = R^+\left(r|s, b^\kappa, a\right) = \mathbb{E}_{\kappa_i \sim b^\kappa}\left[R^{\kappa_i}(r|s, a)\right]. \tag{34}$$

In a BAMDP, the belief is over the transition and reward functions, which are constant for a given task. A meta-policy on BAMDP $\pi^+ : \mathcal{S}^+ \to \Delta\left(\mathcal{A}\right)$ prescribes a distribution over actions for each hyper-state. The agent's objective is now to find a meta-policy on hyper-states $\pi^+$ that maximizes the expected return in the BAMDP,

$$\mathcal{J}_{M^+}\left(\pi^+\right) = \mathbb{E}_{a_t \sim \pi^+\left(\cdot|s_t^+\right), r_t \sim R^+\left(\cdot|s_t^+, a_t\right), s_{t+1}^+ \sim P^+\left(\cdot|s_t^+, a_t\right)}\left[\sum_{t=0}^{\widehat{H}-1} r_t\right] \tag{35}$$

$$= \mathbb{E}_{\kappa_i \sim p(\kappa)}\left[\sum_{j=0}^{N-1} \mathbb{E}_{a_t \sim \pi^+\left(\cdot|s_{jH+t}^+\right), r_t \sim R^{\kappa_i}(\cdot|s_t, a_t), s_{t+1} \sim P^{\kappa_i}(\cdot|s_t, a_t)}\left[\sum_{t=0}^{H-1} r_t\right]\right]. \tag{36}$$

For any meta-policy on hyper-states $\pi^+$, denote the corresponding meta-policy on history contexts $\hat{f}_{\pi^+} : \mathcal{C} \to \Delta\left(\mathcal{A}\right)$, i.e., $\forall t \in \left[\widehat{H}-1\right], \forall c_{:t} \in \mathcal{C}_t$, s.t., $\hat{f}_{\pi^+}\left(\cdot|c_{:t}\right) = \pi^+\left(\cdot|s_t^+\right)$, where $s_t^+ = (s_t, b_t^\kappa) = (s_t, p(\kappa|c_{:t}))$, and we have

$$\mathcal{J}_{\widehat{M}}\left(\hat{f}_{\pi^+}\right) = \mathbb{E}_{\kappa_i \sim p(\kappa)}\left[\sum_{j=0}^{N-1} \mathbb{E}_{a_t \sim \hat{f}_{\pi^+}\left(\cdot|c_{:(jH+t)}\right), r_t \sim R^{\kappa_i}(\cdot|s_t, a_t), s_{t+1} \sim P^{\kappa_i}(\cdot|s_t, a_t)}\left[\sum_{t=0}^{H-1} r_t\right]\right] \tag{37}$$

$$= \mathcal{J}_{M^+}\left(\pi^+\right). \tag{38}$$

The belief MDP is such that an optimal policy for it, coupled with the correct state estimator, will give rise to optimal behavior for the original POMDP (Astrom, 1965; Smallwood & Sondik, 1973; Kaelbling et al., 1998), which indicates that

$$\mathcal{J}_{M^+}\left(\pi^{+,*}\right) = \mathcal{J}_{\widehat{M}}\left(\hat{f}_{\pi^{+,*}}\right) = \mathcal{J}_{\widehat{M}}\left(\hat{\pi}^*\right), \tag{39}$$

where $\pi^{+,*}$ and $\hat{\pi}^*$ are the optimal policies for BAMDP $M^+$ and POMDP $\widehat{M}$, respectively. Thus, the agent can find a policy $\pi^+$ to maximize the expected return in the BAMDP $M^+$ to address the POMDP $\widehat{M}$ by the transformed policy $\hat{f}_{\pi^+}$.

A.1.4   OFFLINE META-RL

In the offline meta-RL setting, a meta-learner only has access to an offline multi-task dataset $\mathcal{D}^+$ and is not allowed to interact with the environment during meta-training (Li et al., 2020). Recent offline meta-RL methods (Dorfman et al., 2021) always utilize task-dependent behavior policies $p(\mu|\kappa)$, which represents the random variable of the behavior policy $\mu(a|s)$ conditioned on the random variable of the task $\kappa$. For brevity, we overload $[\mu] = p(\mu|\kappa)$. Similar to related work on offline RL (Shi et al., 2022), we assume that $\mathcal{D}^+$ is collected through interacting multiple i.i.d. trajectories using task-dependent policies $[\mu]$ in $M^+$. Define the reward and transition distribution of the task-dependent data collection by $\mathbb{P}_{M^+,[\mu]}$ (Jin et al., 2021), i.e., for each step $t$ in a trajectory,

$$\mathbb{P}_{M^+,[\mu]}\left(r_t, s_{t+1} \left| s_t^+, a_t\right.\right) \propto \mathbb{E}_{\kappa_i \sim p(\kappa), \mu_i \sim p(\mu|\kappa_i)}\left[\mathbb{P}_{\kappa_i}\left(r_t, s_{t+1} \left| s_t, a_t\right.\right) \cdot p_{M^+}\left(s_t^+ \left| \kappa_i, \mu_i\right.\right)\right], \quad (40)$$

where $\mathbb{P}_{\kappa_i}$ is the reward and transition distribution of $\kappa_i$ defined in Eq. (14), and $p_{M^+}\left(s_t^+ \left| \kappa_i, \mu_i\right.\right)$ denotes the probability of $s_t^+$ when executing $\mu_i$ in a task $\kappa_i$, i.e.,

$$p_{M^+}\left(s_t^+ \left| \kappa_i, \mu_i\right.\right) = \sum_{c_{:t} \in \mathcal{C}_t} p_{\kappa_i}^{\mu_i}(c_{:t}) \cdot \mathbb{1}\left[b_t^\kappa = p(\kappa|c_{:t})\right], \quad (41)$$

where the state in $c_{:t}$ is $s_t$ and $p_{\kappa_i}^{\mu_i}(c_{:t})$ is defined in Eq. (9). Similar to offline single-task RL (see Appendix A.1.2), offline dataset $\mathcal{D}^+$ can be narrow and a large amount of state-action pairs are not contained. These unseen state-action pairs will be erroneously estimated to have unrealistic values, called a phenomenon *extrapolation error* (Fujimoto et al., 2019). To overcome extrapolation error in offline RL, related works (Fujimoto et al., 2019) introduce batch-constrained RL, which restricts the action space in order to force policy selection of an agent with respect to a given dataset. Define a policy $\pi^+$ to be batch-constrained by $\mathcal{D}^+$ if $\pi^+\left(a \left| s^+\right.\right) = 0$ whenever a tuple $(s^+, a)$ is not contained in $\mathcal{D}^+$. Offline RL (Liu et al., 2020; Chen & Jiang, 2019) approximates policy evaluation for a batch-constrained policy $\pi^+$ by sampling from an offline dataset $\mathcal{D}^+$, which is denoted by $\mathcal{J}_{\mathcal{D}^+}(\pi^+)$ and called *Approximate Dynamic Programming* (ADP; Bertsekas & Tsitsiklis, 1995). During meta-testing, RL agents perform online adaptation using a meta-policy $\pi^+$ in new tasks drawn from meta-RL task distribution. The reward and transition distribution of data collection with $\pi^+$ in $M^+$ during adaptation is defined by

$$\mathbb{P}_{M^+}\left(r_t, s_{t+1} \left| s_t^+, a_t\right.\right) = R^+\left(r_t \left| s_t^+, a_t\right.\right) \cdot P^+\left(s_{t+1} \left| s_t^+, a_t\right.\right), \quad (42)$$

where $P^+\left(s_{t+1} \left| s_t^+, a_t\right.\right)$ is the marginal transition functions of $M^+$, i.e.,

$$P^+\left(s_{t+1} \left| s_t^+, a_t\right.\right) = \mathbb{E}_{\kappa_i \sim b_t^\kappa}\left[P^{\kappa_i}\left(s_{t+1} \left| s_t, a_t\right.\right)\right]. \quad (43)$$

A.2   MAIN RESULTS IN SECTION 3.1

**Definition 1** (Data Distribution Mismatch). *In a BAMDP $M^+$, for each task-dependent behavior policies $[\mu]$ and batch-constrained meta-policy $\pi^+$, the data distribution mismatch between $\pi^+$ and $[\mu]$ is defined by that $\exists s_t^+, a_t$, s.t.,*

$$p_{M^+}^{\pi^+}\left(s_t^+, a_t\right) > 0 \quad \text{and} \quad \mathbb{P}_{M^+}\left(r_t, s_{t+1} \left| s_t^+, a_t\right.\right) \neq \mathbb{P}_{M^+,[\mu]}\left(r_t, s_{t+1} \left| s_t^+, a_t\right.\right), \quad (4)$$

*where $p_{M^+}^{\pi^+}\left(s_t^+, a_t\right)$ is the probability of reaching a tuple $\left(s_t^+, a_t\right)$ while executing $\pi^+$ in $M^+$ (formal definition deferred to Appendix A.2), and $\mathbb{P}_{M^+}, \mathbb{P}_{M^+,[\mu]}$ are the reward and transition distribution of data collection with $\pi^+$ and $[\mu]$ defined in Eq. (1) and (3), respectively.*

More specifically,

$$p_{M^+}^{\pi^+}\left(s_t^+, a_t\right) = p_{M^+}^{\pi^+}(s_t^+) \cdot \pi^+\left(a_t \left| s_t^+\right.\right), \quad (44)$$

where $p_{M^+}^{\pi^+}(s_t^+)$ is defined in Eq. (9).

**Theorem 1.** *There exists a BAMDP $M^+$ with task-dependent behavior policies $[\mu]$ such that, for any batch-constrained meta-policy $\pi^+$, the data distribution induced by $\pi^+$ and $[\mu]$ does not match.*

*Proof.* To serve a concrete example, we construct an offline meta-RL setting shown in Figure 4. In this example, there are $v$ meta-RL tasks $\mathcal{K} = \{\kappa_1, \ldots, \kappa_v\}$ and $v$ behavior policies $\{\mu_1, \ldots, \mu_v\}$,

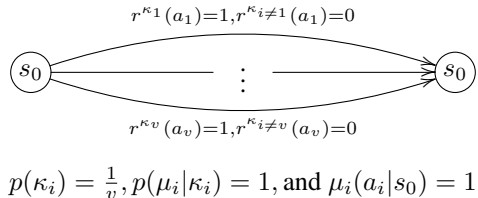

$$p(\kappa_i) = \tfrac{1}{v}, p(\mu_i|\kappa_i) = 1, \text{and } \mu_i(a_i|s_0) = 1$$

Figure 4: A concrete example, which has $v$ meta-RL tasks, one state, $v$ actions, $v$ behavior polices, horizon $H = 1$ in a episode, and $v$ adaptation episodes, where $v \geq 3$.

where $v \geq 3$. Each task $\kappa_i$ has one state $\mathcal{S} = \{s_0\}$, $2v$ actions $\mathcal{A} = \{a_1, \ldots, a_v\}$, and horizon in an episode $H = 1$. For each task $\kappa_i$, RL agents can receive reward 1 performing action $a_i$. During adaptation, the RL agent can interact with the environment within $v$ episodes. The task distribution is uniform, the behavior policy of task $\kappa_i$ is $\mu_i$, and each behavior policy $\mu_i$ will perform $a_i$. When a batch-constrained meta-policy $\pi^+$ selects an action $\tilde{a}$ in the initial state $s_0^+$, we find that

$$\mathbb{P}_{M^+}\left(r = 1 \,\middle|\, s_0^+, \tilde{a}\right) = \frac{1}{v} \neq \mathbb{P}_{M^+,[\mu]}\left(r = 1 \,\middle|\, s_0^+, \tilde{a}\right) = 1, \tag{45}$$

in which there is the probability of $\frac{1}{v}$ to sample a corresponding testing task, whose reward function of $\tilde{a}$ is 1, whereas the reward in the offline dataset collected by $[\mu]$ is all 1. $\qquad\square$

**Fact 1.** *With any probability $1 - \delta \in [0, 1)$, there exists a BAMDP $M^+$ with task-dependent behavior policies $[\mu]$ such that, for any batch-constrained meta-policy $\pi^+$, the agent will visit out-of-distribution belief states during meta-testing due to data distribution mismatch.*

*Proof.* In the example shown in Figure 4, an offline multi-task dataset $D^+$ is drawn from the task-dependent data collection $\mathbb{P}_{M^+,[\mu]}$. Since the reward of $D^+$ is all 1, the belief states in $D^+$ has two types: (i) all task possible $s_0^+$ and (ii) determining task $i$ with receiving reward 1 in action $a_i$. For any batch-constrained meta-policy $\pi^+$ selecting an action $\tilde{a}_j$ on $s_0^+$ during meta-testing, there has probability $1 - \frac{1}{v}$ to receive reward 0 and the belief state will become "excluding task $j$", which is not contained in $D^+$ with $v \geq 3$. For any $\delta \in (0, 1]$, let $v > \frac{1}{\delta}$, with probability $1 - \delta$, the agent will visit out-of-distribution belief states during adaptation. $\qquad\square$

**Proposition 1.** *There exists a BAMDP $M^+$ with task-dependent behavior policies $[\mu]$ such that, for any batch-constrained meta-policy $\pi^+$, the gap of policy evaluation of $\pi^+$ between offline meta-training and online adaptation is at least $\frac{H^+-1}{2}$, where $H^+$ is the planning horizon of $M^+$.*

*Proof.* In the example shown in Figure 2, an offline dataset $D^+$ only contains reward 1, thus for each batch-constrained meta-policy $\pi^+$, the offline evaluation of $\pi^+$ in $D^+$ is $\mathcal{J}_{D^+}(\pi^+) = H^+ = vH$. The optimal meta-policy $\pi^{+,*}$ in this example is to enumerate $a_1, \ldots, a_v$ until the task identification is inferred from an action with a reward of 1. A meta-policy $\pi^{+,*}$ needs to explore in the testing environments and its online policy evaluation is

$$\mathcal{J}_{M^+}\left(\pi^{+,*}\right) = \sum_{k=0}^{N-1} \frac{N-k}{v-k} \prod_{j=0}^{k-1}\left(1 - \frac{1}{v-j}\right) \tag{46}$$

$$= \sum_{k=0}^{N-1} \prod_{j=0}^{k-1} \frac{v-j-1}{v-j} \tag{47}$$

$$= \sum_{k=0}^{N-1} \frac{v-k}{v} = N - \frac{N(N-1)}{2v} \tag{48}$$

$$= \frac{v+1}{2} = \frac{H^+ + 1}{2}, \tag{49}$$

where $N = v$ is the number of adaptation episodes. Thus, the gap of policy evaluation of $\pi^+$ between offline meta-training and online adaptation is

$$\left| \mathcal{J}_{M^+}\left(\pi^+\right) - \mathcal{J}_{D^+}\left(\pi^+\right) \right| \geq \mathcal{J}_{D^+}\left(\pi^+\right) - \mathcal{J}_{M^+}\left(\pi^{+,*}\right) = \frac{H^+ - 1}{2}. \tag{50}$$

$\square$

## A.3 MAIN RESULTS IN SECTION 3.2

**Definition 3** (Transformed BAMDPs (formal version of Definition 2))**.** *A transformed BAMDP is defined as a tuple* $\overline{M}^+ = \left(\overline{\mathcal{S}}^+, \mathcal{A}, \mathcal{R}, H^+, \overline{P}^+, \overline{P}_0^+, \overline{R}^+\right)$*, where* $\overline{\mathcal{S}}^+ = \mathcal{S} \times \overline{\mathcal{B}}$ *is the hyper-state space,* $\overline{\mathcal{B}}$ *is the space of overall beliefs over meta-RL MDPs with behavior policies,* $\mathcal{A}, \mathcal{R}, H^+$ *are the same action space, reward space, and planning horizon as the original BAMDP* $M^+$*, respectively,* $\overline{P}_0^+$ *is the initial hyper-state distribution presenting joint distribution of task and behavior policies* $p(\kappa, \mu) = p(\kappa)p(\mu|\kappa)$*, and* $\overline{P}^+, \overline{R}^+$ *are the transition and reward functions. The goal of meta-RL agents is to find a meta-policy* $\bar{\pi}^+\left(a_t \mid \bar{s}_t^+\right)$ *that maximizes online policy evaluation* $\mathcal{J}_{\overline{M}^+}\left(\bar{\pi}^+\right)$*. Denote the reward and transition distribution of the task-dependent data collection in a transformed BAMDP* $\overline{M}^+$ *by* $\mathbb{P}_{\overline{M}^+, [\mu]}$*, as defined in Eq. (40). Denote the offline multi-task dataset collected by task-dependent data collection* $\mathbb{P}_{\overline{M}^+, [\mu]}$ *by* $\overline{\mathcal{D}}^+$*.*

More specifically, a finite-horizon transformed BAMDP is defined by a tuple $\overline{M}^+ = \left(\overline{\mathcal{S}}^+, \mathcal{A}, \mathcal{R}, H^+, \overline{P}^+, \overline{P}_0^+, \overline{R}^+\right), \overline{\mathcal{S}}^+ = \mathcal{S} \times \overline{\mathcal{B}}$ is the hyper-state space, where $\overline{\mathcal{B}} = \{p(\kappa, \mu|c) \mid c \in \mathcal{C}\}$ is the space of beliefs over meta-RL MDPs with behavior policies, the prior

$$b_0^{\kappa,\mu} = p\left(\kappa, \mu \mid c_{:0}\right) = p(\kappa, \mu) \tag{51}$$

is the distribution of meta-RL MDPs with behavior policies, and $\forall t \in \left[\widehat{H} - 1\right]$, $\forall c_{:(t+1)} \in \mathcal{C}$, denoting $b_t^{\kappa,\mu} = p\left(\kappa, \mu|c_{:t}\right)$ and

$$b_{t+1}^{\kappa,\mu} = p\left(\kappa, \mu \mid c_{:(t+1)}\right) = p\left(p\left(\kappa, \mu|c_{:t}\right) \mid c_{:(t+1)}\right) = p\left(p\left(\kappa, \mu|c_{:t}\right) \mid s_t, c_t\right) = p\left(b_t^{\kappa,\mu} \mid s_t, c_t\right) \tag{52}$$

$$\propto p\left(b_t^{\kappa,\mu}, c_t \mid s_t\right) = p\left(c_t \mid s_t, b_t^{\kappa,\mu}\right) p\left(b_t^{\kappa,\mu} \mid s_t\right) = p\left(c_t \mid s_t, b_t^{\kappa,\mu}\right) b_t^{\kappa,\mu} \tag{53}$$

$$= \mathbb{E}_{(\kappa_i,\mu_i) \sim b_t^{\kappa,\mu}} \left[\mu_i\left(a_t|s_t\right) \cdot R^{\kappa_i}(r_t|s_t, a_t) \cdot P^{\kappa_i}(s_{t+1}|s_t, a_t)\right] \cdot b_t^{\kappa,\mu} \tag{54}$$

is the posterior over the meta-RL MDPs with behavior policies given the context $c_{:(t+1)}$, $\mathcal{A}$, $\mathcal{R}$ and $\widehat{H}$ are the same action space, reward space, and planning horizon as the finite-horizon BAMDP $M^+$, respectively, $\overline{P}^+ : \overline{\mathcal{S}}^+ \times \mathcal{A} \times \mathcal{R} \to \Delta\left(\overline{\mathcal{S}}^+\right)$ is the transition function: $\forall \bar{s}_t^+, \bar{s}_{t+1}^+ \in \overline{\mathcal{S}}^+, a_t \in \mathcal{A}, r_t \in \mathcal{R}$, where denoting $\bar{s}_t^+ = (s_t, b_t^{\kappa,\mu})$ and $\bar{s}_{t+1}^+ = \left(s_{t+1}, \tilde{b}_{t+1}^{\kappa,\mu}\right)$,

$$\overline{P}^+\left(\bar{s}_{t+1}^+ \mid \bar{s}_t^+, a_t, r_t\right) = \overline{P}^+\left(s_{t+1}, \tilde{b}_{t+1}^{\kappa,\mu} \mid s_t, b_t^{\kappa,\mu}, a_t, r_t\right) \tag{55}$$

$$= \overline{P}^+\left(s_{t+1} \mid s_t, b_t^{\kappa,\mu}, a_t, r_t\right) P^+\left(\tilde{b}_{t+1}^{\kappa,\mu} \mid s_t, b_t^{\kappa,\mu}, c_t\right) \tag{56}$$

$$= \mathbb{E}_{(\kappa_i,\mu_i) \sim p\left(b_t^{\kappa,\mu}|s_t, a_t, r_t\right)} \left[P^{\kappa_i}(s_{t+1}|s_t, a_t)\right] \cdot \mathbb{1}\left[\tilde{b}_{t+1}^{\kappa,\mu} = p(b_t^{\kappa,\mu}|s_t, c_t)\right], \tag{57}$$

$\overline{P}_0^+ : \Delta\left(\overline{\mathcal{S}}^+\right)$ is the initial hyper-state distribution, i.e., a deterministic initial hyper-state is

$$\bar{s}_0^+ = (s_0, b_0^{\kappa,\mu}) = (s_0, p(\kappa, \mu)) \in \overline{\mathcal{S}}^+, \tag{58}$$

and $\overline{R}^+ : \overline{\mathcal{S}}^+ \times \mathcal{A} \to \Delta(\mathcal{R})$ is the reward distribution: $\forall \bar{s}^+ = (s, b^{\kappa,\mu}) \in \overline{\mathcal{S}}^+, a \in \mathcal{A}, r \in \mathcal{R}$,

$$\overline{R}^+\left(r|\bar{s}^+, a\right) = \overline{R}^+\left(r \mid s, b^{\kappa,\mu}, a\right) = \mathbb{E}_{(\kappa_i,\mu_i) \sim b^{\kappa,\mu}}\left[R^{\kappa_i}(r|s, a)\right]. \tag{59}$$

In a transformed BAMDP $\overline{M}^+$, the overall belief is about the task-dependent behavior policies, transition function, and reward function, which are constant for a given task. A meta-policy on $\overline{M}^+$ is $\bar{\pi}^+ : \overline{\mathcal{S}}^+ \to \Delta(\mathcal{A})$ prescribes a distribution over actions for each hyper-state. With feasible Bayesian belief updating, the objective of RL agents is now to find a meta-policy on hyper-states $\bar{\pi}^+$ that maximizes the expected return in the transformed BAMDP,

$$\mathcal{J}_{\overline{M}^+}\left(\bar{\pi}^+\right) = \mathbb{E}_{a_t \sim \bar{\pi}^+\left(\cdot \mid \bar{s}_t^+\right), r_t \sim \overline{R}^+\left(\cdot \mid \bar{s}_t^+, a_t\right), s_{t+1}^+ \sim \overline{P}^+\left(\cdot \mid \bar{s}_t^+, a_t\right)} \left[\sum_{t=0}^{\widehat{H}-1} r_t\right] \tag{60}$$

$$= \mathbb{E}_{(\kappa_i, \mu_i) \sim p(\kappa, \mu)} \left[\sum_{j=0}^{N-1} \mathbb{E}_{a_t \sim \bar{\pi}^+\left(\cdot \mid \bar{s}_{jH+t}^+\right), r_t \sim R^{\kappa_i}(\cdot \mid s_t, a_t), s_{t+1} \sim P^{\kappa_i}(\cdot \mid s_t, a_t)} \left[\sum_{t=0}^{H-1} r_t\right]\right] \tag{61}$$

$$= \mathbb{E}_{\kappa_i \sim p(\kappa)} \left[\sum_{j=0}^{N-1} \mathbb{E}_{a_t \sim \bar{\pi}^+\left(\cdot \mid \bar{s}_{jH+t}^+\right), r_t \sim R^{\kappa_i}(\cdot \mid s_t, a_t), s_{t+1} \sim P^{\kappa_i}(\cdot \mid s_t, a_t)} \left[\sum_{t=0}^{H-1} r_t\right]\right]. \tag{62}$$

For any meta-policy on hyper-states $\bar{\pi}^+$, denote the corresponding meta-policy on history contexts $\hat{f}_{\bar{\pi}^+} : \mathcal{C} \to \Delta(\mathcal{A})$, i.e., $\forall t \in \left[\widehat{H}-1\right], \forall c_{:t} \in \mathcal{C}_t$, s.t., $\hat{f}_{\bar{\pi}^+}\left(\cdot \mid c_{:t}\right) = \bar{\pi}^+\left(\cdot \mid \bar{s}_t^+\right)$, where $\bar{s}_t^+ = (s_t, b_t^{\kappa, \mu}) = (s_t, p(\kappa, \mu \mid c_{:t}))$, and we have

$$\mathcal{J}_{\widehat{M}}\left(\hat{f}_{\bar{\pi}^+}\right) = \mathbb{E}_{\kappa_i \sim p(\kappa)} \left[\sum_{j=0}^{N-1} \mathbb{E}_{a_t \sim \hat{f}_{\bar{\pi}^+}\left(\cdot \mid c_{:(jH+t)}\right), r_t \sim R^{\kappa_i}(\cdot \mid s_t, a_t), s_{t+1} \sim P^{\kappa_i}(\cdot \mid s_t, a_t)} \left[\sum_{t=0}^{H-1} r_t\right]\right] \tag{63}$$

$$= \mathcal{J}_{\overline{M}^+}\left(\bar{\pi}^+\right). \tag{64}$$

**Fact 2.** *For feasible Bayesian belief updating, transformed BAMDPs confine the agent in the in-distribution belief states during meta-testing.*

*Proof.* During online adaptation, RL agents construct a hyper-state $\bar{s}_t^+ = \left(s_t, \bar{b}_t\right)$ from the context history and perform a meta-policy $\bar{\pi}^+\left(a_t \mid \bar{s}_t^+\right)$. The new belief $\bar{b}_t$ accounts for the uncertainty of task MDPs and task-dependent behavior policies. In contrast with Fact 1, for feasible Bayesian belief updating, transformed BAMDPs do not allow the agent to visit out-of-distribution belief states. Otherwise, the context history will conflict with the belief about behavior policies, i.e., RL agents cannot update their beliefs $\bar{b}_t$ when they have observed an event that they believe to have probability zero. $\square$

**Lemma 1.** *In an MDP $M$, for each behavior policy $\mu$ and batch-constrained policy $\pi$, collect a dataset $\mathcal{D}$ and the gap between approximate offline policy evaluation $\mathcal{J}_{\mathcal{D}}(\pi)$ and accurate policy evaluation $\mathcal{J}_M(\pi)$ will asymptotically approach to 0, as the offline dataset $\mathcal{D}$ grows.*

From a given dataset $\mathcal{D}$, an abstract MDP $M_{\mathcal{D}}$ can be estimated (Fujimoto et al., 2019; Yin & Wang, 2021; Szepesvári, 2022). According to concentration bounds, the estimated transition and reward function will asymptotically approach $M$ (Yin & Wang, 2021) during the support of $\mathcal{D}$. Then, using the simulation lemma (Alekh Agarwal, 2017; Szepesvári, 2022), the gap between $\mathcal{J}_{\mathcal{D}}(\pi)$ and $\mathcal{J}_M(\pi)$ will asymptotically approach to 0, as the offline dataset $\mathcal{D}$ grows. Formal proofs are deferred in Appendix A.4.

**Theorem 2.** *In a transformed BAMDP $\overline{M}^+$, for each task-dependent behavior policy $[\mu]$ and batch-constrained meta-policy $\bar{\pi}^+$, the data distribution induced by $\bar{\pi}^+$ and $[\mu]$ matches after filtering out out-of-distribution episodes in online adaptation. Besides, the policy evaluation of $\bar{\pi}^+$ in offline meta-training and online adaptation will be asymptotically consistent, as the offline dataset grows.*

*Proof.* We assume feasible Bayesian belief updating in this proof. At first, $\forall s_t^+, a_t$, s.t. $p_{\overline{M}^+}^{\bar{\pi}^+}\left(\bar{s}_t^+, a_t\right) > 0$, we aim to prove

$$\mathbb{P}_{\overline{M}^+}\left(r_t, s_{t+1} \mid \bar{s}_t^+, a_t\right) = \mathbb{P}_{\overline{M}^+, [\mu]}\left(r_t, s_{t+1} \mid \bar{s}_t^+, a_t\right). \tag{65}$$

Since $\bar{\pi}^+$ is batch-constrained policy by $[\mu]$, if $p_{\overline{M}^+}^{\bar{\pi}^+}\left(\bar{s}_t^+, a_t\right) > 0$, we have $p_{\overline{M}^+}^{[\mu]}\left(\bar{s}_t^+, a_t\right) > 0$. Then,

$$\mathbb{P}_{\overline{M}^+}\left(r_t, s_{t+1} \mid \bar{s}_t^+, a_t\right) = \overline{R}^+\left(r_t \mid \bar{s}_t^+, a_t\right) \cdot \overline{P}^+\left(s_{t+1} \mid \bar{s}_t^+, a_t\right) \tag{66}$$

$$= \mathbb{E}_{\kappa_i \sim b_t^{\kappa,\mu}}\left[\mathbb{P}_{\kappa_i}\left(r_t, s_{t+1} \mid s_t, a_t\right)\right] \tag{67}$$

$$= \mathbb{E}_{(\kappa_i, \mu_i) \sim b_t^{\kappa,\mu}}\left[\mathbb{P}_{\kappa_i}\left(r_t, s_{t+1} \mid s_t, a_t\right)\right] \tag{68}$$

where $b_t^{\kappa,\mu}$ is defined in Eq. (51) and

$$p\left(\kappa_i, \mu_i \mid b_t^{\kappa,\mu}\right) = \mathbb{E}_{\kappa_i \sim p(\kappa), \mu_i \sim p(\mu|\kappa_i)}\left[\mathbb{E}_{c_{:t} \sim p_{\overline{M}^+}(c_{:t}|\kappa_i,\mu_i)}\left[\mathbb{1}\left[b_t^{\kappa,\mu} = p(\kappa, \mu|c_{:t})\right]\right]\right], \tag{69}$$

where $p_{\overline{M}^+}\left(c_{:t}|\kappa_i, \mu_i\right)$ is defined in Eq. (9). According to Eq. (40) and (41),

$$\mathbb{P}_{\overline{M}^+, [\mu]}\left(r_t, s_{t+1} \mid \bar{s}_t^+, a_t\right) \tag{70}$$

$$\propto \mathbb{E}_{\kappa_i \sim p(\kappa), \mu_i \sim p(\mu|\kappa_i)}\left[\mathbb{P}_{\kappa_i}\left(r_t, s_{t+1} \mid s_t, a_t\right) \cdot p_{\overline{M}^+}\left(\bar{s}_t^+ \mid \kappa_i, \mu_i\right)\right] \tag{71}$$

$$= \mathbb{E}_{\kappa_i \sim p(\kappa), \mu_i \sim p(\mu|\kappa_i)}\left[\mathbb{E}_{c_{:t} \sim p_{\overline{M}^+}(c_{:t}|\kappa_i,\mu_i)}\left[\mathbb{P}_{\kappa_i}\left(r_t, s_{t+1} \mid s_t, a_t\right) \cdot \mathbb{1}\left[b_t^{\kappa,\mu} = p(\kappa,\mu|c_{:t})\right]\right]\right] \tag{72}$$

$$= \mathbb{E}_{(\kappa_i, \mu_i) \sim b_t^{\kappa,\mu}}\left[\mathbb{P}_{\kappa_i}\left(r_t, s_{t+1} \mid s_t, a_t\right)\right] \tag{73}$$

$$= \mathbb{P}_{\overline{M}^+}\left(r_t, s_{t+1} \mid \bar{s}_t^+, a_t\right). \tag{74}$$

Thus, the data distribution induced by $\bar{\pi}^+$ and $[\mu]$ matches. Directly use Lemma 1 in a transformed BAMDP $\overline{M}^+$, in which $\overline{M}^+$ is a belief MDP, a type of MDP. Therefore, the policy evaluation of $\bar{\pi}^+$ in offline meta-training and online adaptation will be asymptotically consistent, as the offline dataset grows. $\qquad\square$

### A.4 OMITTED PROOF OF LEMMA 1

**Definition 4** (Dataset Induced Finite-Horizon MDPs). *In a finite-horizon MDP $M$ with an offline dataset $\mathcal{D}$, a dataset induced finite-horizon MDP is defined by $M^{\mathcal{D}} = \left(\mathcal{S}, \mathcal{A}, \mathcal{R}, H, P^{M^{\mathcal{D}}}, R^{M^{\mathcal{D}}}\right)$, with the same state space, action space, reward space, and horizon as $M$. The transition function is defined as follows: $\forall s, s' \in \mathcal{S}, a \in \mathcal{A}$,*

$$P^{M^{\mathcal{D}}}(s'|s, a) = \begin{cases} \dfrac{N(s, a, s')}{N(s, a)}, & \text{if } N(s, a) > 0, \\ 0, & \text{otherwise}, \end{cases} \tag{75}$$

*where $N(s, a, s')$ and $N(s, a)$ are the number of times the tuples $(s, a, s')$ and $(s, a)$ are observed in $\mathcal{D}$, respectively. The reward function is defined by $\forall s \in \mathcal{S}, a \in \mathcal{A}, r \in \mathcal{R}$,*

$$R^{M^{\mathcal{D}}}(r|s, a) = \begin{cases} \dfrac{N(s, a, r)}{N(s, a)}, & \text{if } N(s, a) > 0, \\ 0, & \text{otherwise}, \end{cases} \tag{76}$$

*where $N(s, a, r)$ is the number of times the tuple $(s, a, r)$ are observed in $\mathcal{D}$. The offline policy evaluation in $\mathcal{D}$ is equal to the policy evaluation in $M_{\mathcal{D}}$, i.e., for any batch-constrained policy $\pi$,*

$$\mathcal{J}_{\mathcal{D}}(\pi) = \mathcal{J}_{M_{\mathcal{D}}}(\pi). \tag{77}$$

Note that dataset induced finite-horizon MDPs $M^{\mathcal{D}}$ are not defined on supports outside of dataset $\mathcal{D}$. For simplicity, We set all undefined numbers to 0 in the transition and reward function.

**Lemma 2** (Simulation Lemma for Offline Finite-Horizon MDPs). *In an MDP $M$ with an offline dataset $\mathcal{D}$, for any batch-constrained policy $\pi \in \Pi^{\mathcal{D}}$, if*

$$\max_{s \in \mathcal{S}, a \in \mathcal{A} \text{ with } \rho_\pi^{M_{\mathcal{D}}}(s,a)>0} \left\| P^{M_{\mathcal{D}}}(\cdot|s, a) - P^M(\cdot|s, a) \right\|_1 \le \epsilon_P, \tag{78}$$

$$\max_{s \in \mathcal{S}, a \in \mathcal{A} \text{ with } \rho_\pi^{M_{\mathcal{D}}}(s,a)>0} \left| r^{M_{\mathcal{D}}}(s, a) - r^M(s, a) \right| \le \epsilon_r, \tag{79}$$

$$\max_{s \in \mathcal{S}, a \in \mathcal{A}} \max\left(r^{M_{\mathcal{D}}}(s,a), r^M(s,a)\right) \le r_{max}, \tag{80}$$

where $r^M(s,a) = \mathbb{E}_{\tilde{r} \sim R^M(s,a)}[\tilde{r}]$ and $r^{M_{\mathcal{D}}}(s,a) = \mathbb{E}_{\tilde{r} \sim R^{M_{\mathcal{D}}}(s,a)}[\tilde{r}]$, we have

$$|\mathcal{J}_{M_{\mathcal{D}}}(\pi) - \mathcal{J}_M(\pi)| \le H\epsilon_r + \frac{H(H-1)r_{max}}{2}\epsilon_P. \tag{81}$$

*Proof.* Similar with famous Simulation Lemma in finite-horizon MDPs (Alekh Agarwal, 2017), the proof is as follows. Recall value function $\forall h \in [H-1], \forall s \in \mathcal{S}_h$ (see Eq. (10)),

$$V_\pi^M(s) = \sum_{a \in \mathcal{A}} \pi(a|s)\left(r^M(s,a) + \sum_{s' \in \bar{\mathcal{S}}_{h+1}} P^M(s'|s,a)V_\pi^M(s')\right), \tag{82}$$

$\mathcal{J}_M(\pi) = V_\pi^M(s_0)$, and $\forall h \in [H]$,

$$\max_{s \in \bar{\mathcal{S}}_h}\left|V_\pi^{\bar{\kappa}}(s)\right| \le (H-h)r_{max}. \tag{83}$$

We will prove $\forall h \in [H], \forall s \in \mathcal{S}_h$ with $\rho_\pi^{M_{\mathcal{D}}}(s) > 0$,

$$\left|V_\pi^{M_{\mathcal{D}}}(s) - V_\pi^M(s)\right| \le (H-h)\epsilon_r + \frac{(H-h)(H-h-1)r_{max}}{2}\epsilon_P \tag{84}$$

by induction. When $h = H-1$, we have $\forall s \in \mathcal{S}_h$ with $\rho_\pi^{M_{\mathcal{D}}}(s) > 0$,

$$\left|V_\pi^{M_{\mathcal{D}}}(s) - V_\pi^M(s)\right| = \left|\sum_{a \in \mathcal{A}} \pi(a|s)r^{M_{\mathcal{D}}}(s,a) - \sum_{a \in \mathcal{A}} \pi(a|s)r^M(s,a)\right| \le \epsilon_r \tag{85}$$

holds. And $\forall h \in [H-1], \forall s \in \mathcal{S}_h$ with $\rho_\pi^{M_{\mathcal{D}}}(s) > 0$,

$$\left|V_\pi^{M_{\mathcal{D}}}(s) - V_\pi^M(s)\right| \tag{86}$$

$$= \left|\sum_{a \in \mathcal{A}} \pi(a|s)\left(r^{M_{\mathcal{D}}}(s,a) + \sum_{s' \in \mathcal{S}_{h+1}} P^{M_{\mathcal{D}}}(s'|s,a)V_\pi^{M_{\mathcal{D}}}(s')\right) - \right. \tag{87}$$

$$\left.\sum_{a \in \mathcal{A}} \pi(a|s)\left(r^M(s,a) + \sum_{s' \in \mathcal{S}_{h+1}} P^M(s'|s,a)V_\pi^M(s')\right)\right| \tag{88}$$

$$\le \sum_{a \in \mathcal{A}} \pi(a|s)\left|r^{M_{\mathcal{D}}}(s,a) - r^M(s,a)\right| + \tag{89}$$

$$\sum_{a \in \mathcal{A}} \pi(a|s) \sum_{s' \in \mathcal{S}_{h+1}} \left|P^{M_{\mathcal{D}}}(s'|s,a)V_\pi^{M_{\mathcal{D}}}(s') - P^M(s'|s,a)V_\pi^M(s')\right| \tag{90}$$

$$\le \epsilon_r + \sum_{a \in \mathcal{A}} \pi(a|s) \sum_{s' \in \mathcal{S}_{h+1}} \left|P^{M_{\mathcal{D}}}(s'|s,a) - P^M(s'|s,a)\right| V_\pi^M(s') + \tag{91}$$

$$\sum_{a \in \mathcal{A}} \pi(a|s) \sum_{s' \in \mathcal{S}_{h+1}} P^{M_{\mathcal{D}}}(s'|s,a)\left|V_\pi^{M_{\mathcal{D}}}(s') - V_\pi^M(s')\right| \tag{92}$$

$$\le \epsilon_r + (H-(h+1))r_{max}\epsilon_P + \tag{93}$$

$$\left((H-(h+1))\epsilon_r + \frac{(H-(h+1))(H-(h+1)-1)r_{max}}{2}\epsilon_P\right) \tag{94}$$

$$= (H-h)\epsilon_r + \frac{(H-h)(H-h-1)r_{max}}{2}\epsilon_P. \tag{95}$$

Thus,

$$|\mathcal{J}_{M_{\mathcal{D}}}(\pi) - \mathcal{J}_M(\pi)| = \left|V_\pi^{M_{\mathcal{D}}}(s_0) - V_\pi^M(s_0)\right| \le H\epsilon_r + \frac{H(H-1)r_{max}}{2}\epsilon_P. \tag{96}$$

$\square$

**Lemma 3.** *In an MDP $M$ with an offline dataset $\mathcal{D}$ collected by a behavior policy $\mu$, for any batch-constrained policy $\pi \in \Pi^{\mathcal{D}}$, $\forall \delta \in (0,1]$, with probability $1 - \delta$,*

$$|\mathcal{J}_{M_{\mathcal{D}}}(\pi) - \mathcal{J}_M(\pi)| \leq H^2 |\mathcal{S}| \sqrt{\frac{\log\left(\frac{1}{\delta}\right) + \log\left(2 |\mathcal{S}|^2 |\mathcal{A}|\right)}{K d_\mu^{M_{\mathcal{D}}}}}, \tag{97}$$

*where $K$ is the number of trajectories in the dataset $\mathcal{D}$ and $d_\mu^{M_{\mathcal{D}}}$ is the minimal visitation state-action distribution induced by the behavior policy $\mu$ in $M_{\mathcal{D}}$ (see Eq. (19)).*

*Proof.* $\forall s, s' \in \mathcal{S}, a \in \mathcal{A}$ with $\rho_\pi^{M_{\mathcal{D}}}(s,a) > 0$, note that $\rho_\pi^{M_{\mathcal{D}}}(s,a) \geq d_\mu^{M_{\mathcal{D}}}$ and according to Binomial theorem and Hoeffding's inequality, $\forall \epsilon \in [0,1]$,

$$\mathbb{P}\left(\left|P^{M_{\mathcal{D}}}(s'|s,a) - P^M(s'|s,a)\right| \geq \epsilon\right) \leq 2\left(1 - d_\mu^{M_{\mathcal{D}}} + d_\mu^{M_{\mathcal{D}}} \exp\left(-2\epsilon^2\right)\right)^K \tag{98}$$

$$\leq 2\left(1 - d_\mu^{M_{\mathcal{D}}} \epsilon^2\right)^K \tag{99}$$

and

$$\mathbb{P}\left(\left|r^{M_{\mathcal{D}}}(s,a) - r^M(s,a)\right| \geq \epsilon\right) \leq 2\left(1 - d_\mu^{M_{\mathcal{D}}} \epsilon^2\right)^K. \tag{100}$$

Thus, using union bound, $\forall \delta \in (0,1]$, with probability $1 - \delta$, denote

$$\epsilon_P = \max_{s \in \mathcal{S}, a \in \mathcal{A} \text{ with } \rho_\pi^{M_{\mathcal{D}}}(s,a) > 0} \left\|P^{M_{\mathcal{D}}}(\cdot|s,a) - P^M(\cdot|s,a)\right\|_1 \tag{101}$$

$$\leq \max_{s \in \mathcal{S}, a \in \mathcal{A} \text{ with } \rho_\pi^{M_{\mathcal{D}}}(s,a) > 0} |\mathcal{S}| \left\|P^{M_{\mathcal{D}}}(\cdot|s,a) - P^M(\cdot|s,a)\right\|_\infty \tag{102}$$

$$\leq |\mathcal{S}| \sqrt{\frac{\log\left(\frac{1}{\delta}\right) + \log\left(2 |\mathcal{S}|^2 |\mathcal{A}|\right)}{K d_\mu^{M_{\mathcal{D}}}}}, \tag{103}$$

$$\epsilon_r = \max_{s \in \mathcal{S}, a \in \mathcal{A} \text{ with } \rho_\pi^{M_{\mathcal{D}}}(s,a) > 0} \left|r^{M_{\mathcal{D}}}(s,a) - r^M(s,a)\right| \leq \sqrt{\frac{\log\left(\frac{1}{\delta}\right) + \log\left(2 |\mathcal{S}| |\mathcal{A}|\right)}{K d_\mu^{M_{\mathcal{D}}}}}, \tag{104}$$

and $r_{max} = 1$, thus, according to Lemma 2 (a varianted simulation lemma in offline RL), we have

$$|\mathcal{J}_{M_{\mathcal{D}}}(\pi) - \mathcal{J}_M(\pi)| \leq \left(H + \frac{H(H-1)}{2} |\mathcal{S}|\right) \sqrt{\frac{\log\left(\frac{1}{\delta}\right) + \log\left(2 |\mathcal{S}|^2 |\mathcal{A}|\right)}{K d_\mu^{M_{\mathcal{D}}}}} \tag{105}$$

$$\leq H^2 |\mathcal{S}| \sqrt{\frac{\log\left(\frac{1}{\delta}\right) + \log\left(2 |\mathcal{S}|^2 |\mathcal{A}|\right)}{K d_\mu^{M_{\mathcal{D}}}}} \tag{106}$$

$\square$

**Lemma 1.** *In an MDP $M$, for each behavior policy $\mu$ and batch-constrained policy $\pi$, collect a dataset $\mathcal{D}$ and the gap between approximate offline policy evaluation $\mathcal{J}_{\mathcal{D}}(\pi)$ and accurate policy evaluation $\mathcal{J}_M(\pi)$ will asymptotically approach to 0, as the offline dataset $\mathcal{D}$ grows.*

*Proof.* From Lemma 3, as the size of an offline dataset $|\mathcal{D}| = KH$ grows, the gap between approximate offline policy evaluation $\mathcal{J}_{\mathcal{D}}(\pi)$ and accurate policy evaluation $\mathcal{J}_M(\pi)$ will asymptotically approach to 0. $\square$

### A.5 HOW TO FILTER OUT OUT-OF-DISTRIBUTION EPISODES IN TRANSFORMED BAMDPS DURING META-TESTING?

**Definition 5** (Sub-Datasets Collected by Single Task Data Collection). *In a transformed BAMDP $\overline{M}^+$, an offline multi-task dataset $\overline{\mathcal{D}}^+$ is drawn from the task-dependent data collection $\mathbb{P}_{\overline{M}^+, [\mu]}$. A sub-dataset collected by a behavior policy $\mu_i$ in a task $\kappa_i$ is defined by $\mathcal{D}_{\kappa_i, \mu_i}$. Note an offline multi-task dataset $\overline{\mathcal{D}}^+$ is the union of sub-datasets $\mathcal{D}_{\kappa_i, \mu_i}$, i.e.,*

$$\overline{\mathcal{D}}^+ = \bigcup_{\kappa_i, \mu_i} \mathcal{D}_{\kappa_i, \mu_i}. \tag{107}$$

For each sub-dataset $\mathcal{D}_{\kappa_i,\mu_i}$, we can define a batch-constrained policy set in a single-task $(\kappa_i, \mu_i)$ as $\Pi^{\mathcal{D}_{\kappa_i,\mu_i}}$ (see the definition in Eq. (15)).

**Definition 6** (Meta-Policy with Thompson Sampling). *For each transformed BAMDP $\overline{M}^+$, a meta-policy set with Thompson sampling on $\overline{M}^+$ is defined by $\bar{\pi}^{+,T} : \mathcal{S} \times \overline{\mathcal{B}} \times \mathcal{K} \times \Pi_{[\mu]} \to \Delta(\mathcal{A})$, where $\overline{\mathcal{B}}$ is the space of beliefs over meta-RL MDPs with behavior policies, $\mathcal{K}$ is the space of task $\kappa$, and $\Pi_{[\mu]}$ is the space of task-dependent behavior policies. In each episode, $\bar{\pi}^{+,T}$ samples a task hypothesis $(\kappa_i, \mu_i)$ from the current belief $b_{t'}^{\kappa,\mu}$, where $t'$ is the starting step in this episode. During this episode, $\bar{\pi}^{+,T}(\cdot\,|s_t, b_{t'}^{\kappa,\mu}, \kappa_i, \mu_i)$ prescribes a distribution over actions for each state $s_t$, belief $b_{t'}^{\kappa,\mu}$, and task hypothesis $(\kappa_i, \mu_i)$. Beliefs $b_{t'}^{\kappa,\mu}$ and task hypotheses $(\kappa_i, \mu_i)$ will periodically update after each episode.*

In the deep-learning-based implementation, a context-based meta-RL algorithm, PEARL (Rakelly et al., 2019), utilizes a meta-policy with Thompson sampling (Strens, 2000) to iteratively update task belief by interacting with the environment and improve the meta-policy based on the "task hypothesis" sampled from the current beliefs. We can adopt such adaptation protocol to design practical offline meta-RL algorithms for transformed BAMDPs.

**Definition 7** (Batch-Constrained Meta-Policy Set with Thompson Sampling). *For each transformed BAMDP $\overline{M}^+$ with an offline multi-task dataset $\overline{\mathcal{D}}^+$, a batch-constrained meta-policy set with Thompson sampling is defined by*

$$\Pi^{\overline{\mathcal{D}}^+,T} = \left\{ \bar{\pi}^{+,T} \,\middle|\, \bar{\pi}^{+,T}(a_t\,|s_t, b_{t'}^{\kappa,\mu}, \kappa_i, \mu_i) = 0 \text{ whenever } (s_t, a_t) \notin \mathcal{D}_{\kappa_i,\mu_i}, \forall b_{t'}^{\kappa,\mu} \right\}, \quad (108)$$

*where denoting $(s_t, a_t) \in \mathcal{D}_{\kappa_i,\mu_i}$ if there exists a trajectory containing $(s_t, a_t)$ in the dataset $\mathcal{D}_{\kappa_i,\mu_i}$.*

The batch-constrained meta-policy set with Thompson sampling $\Pi^{\overline{\mathcal{D}}^+,T}$ consists of the meta-policies that for any state $s_t$ observed in the hypothesis dataset $\mathcal{D}_{\kappa_i,\mu_i}$, the agent will not select an action outside of the dataset. Note that in each episode with a task hypothesis $(\kappa_i, \mu_i)$, a batch-constrained meta-policy with Thompson sampling $\bar{\pi}^{+,T}$ is batch-constrained within a sub-dataset $\mathcal{D}_{\kappa_i,\mu_i}$, i.e., $\forall b_{t'}^{\kappa,\mu}$, we have $\bar{\pi}^{+,T}(\cdot\,|s_t, b_{t'}^{\kappa,\mu}, \kappa_i, \mu_i) \in \Pi^{\mathcal{D}_{\kappa_i,\mu_i}}$.

**Definition 8** (Probability that a Policy Leaves the Dataset). *In an MDP $M$ and an arbitrary offline dataset $\mathcal{D}$, for each policy $\pi : \mathcal{S} \to \Delta(\mathcal{A})$, the probability that executing $\pi$ in $M$ leaves the dataset $\mathcal{D}$ for an episode is defined by*

$$p_{out}^{M,\mathcal{D}}(\pi) = \sum_{\tau_H} p_\pi^M(\tau_H)\, \mathbb{1}\,[\tau_H \text{ leaves } \mathcal{D}] \quad (109)$$

$$= \sum_{\tau_H} p_\pi^M(\tau_H)\, \mathbb{1}\,[\exists t \in [H] \text{ s.t. } s_t \notin \mathcal{D} \text{ or } (s_t, a_t, r_t) \notin \mathcal{D}], \quad (110)$$

*where $p_\pi^M(\tau_H)$ is the probability of executing $\pi$ in $M$ to generate an $H$-horizon trajectory $\tau_H$ (see the definition in Eq. (9)), denoting $s_t \in \mathcal{D}$ if there exists a trajectory containing $s_t$ in the dataset $\mathcal{D}$, and similarly for $(s_t, a_t, r_t) \in \mathcal{D}$.*

When we aim to confine the agent in the in-distribution states with high probability as the offline dataset $\mathcal{D}$ grows, it is equivalent to bound the probability that executing a policy $\pi$ in $M$ leaves the dataset $\mathcal{D}$ for an episode, i.e., $p_{out}^{M,\mathcal{D}}(\pi)$.

**Theorem 3.** *In a transformed BAMDP $\overline{M}^+$ with an offline multi-task dataset $\overline{\mathcal{D}}^+$ collected by task-dependent behavior policies $[\mu]$, consider each batch-constrained meta-policy with Thompson sampling $\bar{\pi}^{+,T} \in \Pi^{\overline{\mathcal{D}}^+,T}$ (see Definition 6 and 7). For each adaptation episode in a meta-testing task $\kappa_{test} \sim p(\kappa)$, denote the current belief by $b_{t'}^{\kappa,\mu}$, there exists a task hypothesis $(\kappa_i, \mu_i)$ from $b_{t'}^{\kappa,\mu}$, executing $\bar{\pi}^{+,T}$ with $(b_{t'}^{\kappa,\mu}, \kappa_i, \mu_i)$ in $\kappa_{test}$ will confine the agent in the in-distribution belief states with high probability, as the offline dataset $\overline{\mathcal{D}}^+$ grows.*

From Theorem 3, for each adaptation episode in $\kappa_{test}$ with the current belief $b_{t'}^{\kappa,\mu}$, we can sample task task hypothesis $(\kappa_i, \mu_i) \sim b_{t'}^{\kappa,\mu}$ and execute $\bar{\pi}^{+,T}$ with $(b_{t'}^{\kappa,\mu}, \kappa_i, \mu_i)$ to interact with the environment until finding an in-distribution episode. Thus, we prove that meta-policies with Thompson sampling

can filter out out-of-distribution episodes in $\overline{M}^+$ with high probability, as the offline dataset $\overline{\mathcal{D}}^+$ grows. Note that Theorem 3 considers arbitrary task distribution $p(\kappa)$, since the distance between the closest meta-training task $\kappa_{i*}$ and $\kappa_{test}$ will asymptotically approach zero with high probability, as the i.i.d. offline meta-training tasks $\mathcal{K}_{train}$ sampled from $p(\kappa)$ in $\overline{\mathcal{D}}^+$ grows. The detailed proofs are as follows.

*Proof.* For each batch-constrained meta-policy with Thompson sampling $\bar{\pi}^{+,T} \in \Pi^{\overline{\mathcal{D}}^+,T}$ with $(b_{t'}^{\kappa,\mu}, \kappa_i, \mu_i)$, similar to Definition 8, define the probability that executing $\bar{\pi}^{+,T}\left(\cdot|s_t, b_{t'}^{\kappa,\mu}, \kappa_i, \mu_i\right)$ leaves the dataset $\overline{\mathcal{D}}^+$ in an adaptation episode of a meta-testing task $\kappa_{test} \sim p(\kappa)$:

$$p_{out}^{\kappa_{test},\overline{\mathcal{D}}^+}\left(\bar{\pi}^{+,T}, b_{t'}^{\kappa,\mu}, \kappa_i, \mu_i\right) = \sum_{\bar{\tau}_H^+} p_{\bar{\pi}^{+,T}}^{\kappa_{test}}\left(\bar{\tau}_H^+ \big| b_{t'}^{\kappa,\mu}, \kappa_i, \mu_i\right) \mathbb{1}\left[\bar{\tau}_H^+ \text{ leaves } \overline{\mathcal{D}}^+\right], \qquad (111)$$

where $p_{\bar{\pi}^{+,T}}^{\kappa_{test}}\left(\bar{\tau}_H^+ \big| b_{t'}^{\kappa,\mu}, \kappa_i, \mu_i\right)$ is the probability of executing $\bar{\pi}^{+,T}\left(\cdot|s_t, b_{t'}^{\kappa,\mu}, \kappa_i, \mu_i\right)$ in $\kappa_{test}$ to generate an $H$-horizon trajectory $\bar{\tau}_H^+$ in an adaptation episode, i.e.,

$$p_{\bar{\pi}^{+,T}}^{\kappa_{test}}\left(\bar{\tau}_H^+ \big| b_{t'}^{\kappa,\mu}, \kappa_i, \mu_i\right) \qquad (112)$$

$$= \left(\prod_{t=t'}^{t'+H-1} \bar{\pi}^{+,T}\left(a_t \big| s_t, b_{t'}^{\kappa,\mu}, \kappa_i, \mu_i\right) \cdot R^{\kappa_{test}}\left(r_t | s_t, a_t\right)\right) \prod_{t=t'}^{t'+H-2} P^{\kappa_{test}}\left(s_{t+1} | s_t, a_t, r_t\right) \quad (113)$$

$$= p_{\bar{\pi}^{+,T}}^{\kappa_{test}}\left(\tau_H \big| b_{t'}^{\kappa,\mu}, \kappa_i, \mu_i\right), \qquad (114)$$

where we can transform $\bar{\tau}_H^+$ to $\tau_H$ with the same probability since the belief $b_{t'}^{\kappa,\mu}$ and task hypothsis $(\kappa_i, \mu_i)$ will periodically update after each episode. Therefore,

$$p_{out}^{\kappa_{test},\overline{\mathcal{D}}^+}\left(\bar{\pi}^{+,T}, b_{t'}^{\kappa,\mu}, \kappa_i, \mu_i\right) = \sum_{\bar{\tau}_H^+} p_{\bar{\pi}^{+,T}}^{\kappa_{test}}\left(\tau_H \big| b_{t'}^{\kappa,\mu}, \kappa_i, \mu_i\right) \mathbb{1}\left[\bar{\tau}_H^+ \text{ leaves } \overline{\mathcal{D}}^+\right] \qquad (115)$$

$$\leq \sum_{\tau_H} p_{\bar{\pi}^{+,T}}^{\kappa_{test}}\left(\tau_H \big| b_{t'}^{\kappa,\mu}, \kappa_i, \mu_i\right) \mathbb{1}\left[\tau_H \text{ leaves } \mathcal{D}_{\kappa_{i*},\mu_{i*}}\right] \qquad (116)$$

$$= p_{out}^{\kappa_{test},\mathcal{D}_{\kappa_{i*},\mu_{i*}}}\left(\bar{\pi}^{+,T} \big| b_{t'}^{\kappa,\mu}, \kappa_i, \mu_i\right), \qquad (117)$$

where $p_{out}^{\kappa_{test},\mathcal{D}_{\kappa_{i*},\mu_{i*}}}\left(\bar{\pi}^{+,T} \big| b_{t'}^{\kappa,\mu}, \kappa_i, \mu_i\right)$ is the probability that executing $\bar{\pi}^{+,T}\left(\cdot|s_t, b_{t'}^{\kappa,\mu}, \kappa_i, \mu_i\right)$ in $\kappa_{test}$ leaves the sub-dataset $\mathcal{D}_{\kappa_{i*},\mu_{i*}}$ for an episode (see Definition 8), $\mathcal{D}_{\kappa_{i*},\mu_{i*}}$ is a sub-dataset collected in $\overline{\mathcal{D}}^+$ (see Definition 5) and $\kappa_{i*}$ is the closest offline meta-training task to $\kappa_{test}$, i.e.,

$$\kappa_{i*} = \underset{\kappa_i \in \mathcal{K}_{train}}{\arg\min} \|\kappa_i - \kappa_{test}\|_\infty \qquad (118)$$

$$= \underset{\kappa_i \in \mathcal{K}_{train}}{\arg\min} \max\left(\|P^{\kappa_i}(s,a,s') - P^{\kappa_{test}}(s,a,s')\|_\infty, \|R^{\kappa_i}(s,a,r) - R^{\kappa_{test}}(s,a,r)\|_\infty\right), \qquad (119)$$

in which denoting the i.i.d. offline meta-training tasks sampled from $p(\kappa)$ in $\overline{\mathcal{D}}^+$ by $\mathcal{K}_{train}$. From Lemma 5, as the offline dataset $\overline{\mathcal{D}}^+$ grows, $\mathcal{D}_{\kappa_{i*},\mu_{i*}}$ and $\mathcal{K}_{train}$ grow monotonically, for any batch-constrained policy $\pi$ in $\mathcal{D}_{\kappa_{i*},\mu_{i*}}$, i.e., $\forall \pi \in \Pi^{\mathcal{D}_{\kappa_{i*},\mu_{i*}}}$, when executing $\pi$ in an episode of $\kappa_{test} \sim p(\kappa)$, the probability leaving the dataset $\overline{\mathcal{D}}^+$ is $p_{out}^{\kappa_{test},\mathcal{D}_{\kappa_{i*},\mu_{i*}}}(\pi)$, which asymptotically approaches zero.

For the first adaptation episode in a meta-testing task $\kappa_{test} \sim p(\kappa)$ with the prior belief $p(\kappa,\mu) = p(\kappa)p(\mu|\kappa)$, there exists a task hypothesis $(\kappa_{i*}, \mu_{i*})$ in the prior $p(\kappa,\mu)$, then due to $\bar{\pi}^{+,T}\left(\cdot|s_t, p(\kappa,\mu), \kappa_{i*}, \mu_{i*}\right) \in \Pi^{\mathcal{D}_{\kappa_{i*},\mu_{i*}}}$ from Definition 7 and

$$p_{out}^{\kappa_{test},\overline{\mathcal{D}}^+}\left(\bar{\pi}^{+,T}, p(\kappa,\mu), \kappa_{i*}, \mu_{i*}\right) \leq p_{out}^{\kappa_{test},\mathcal{D}_{\kappa_{i*},\mu_{i*}}}\left(\bar{\pi}^{+,T} \big| p(\kappa,\mu), \kappa_{i*}, \mu_{i*}\right), \qquad (120)$$

as the offline dataset $\overline{\mathcal{D}}^+$ grows, executing $\bar{\pi}^{+,T}$ with $(p(\kappa,\mu), \kappa_{i*}, \mu_{i*})$ for the first episode in $\kappa_{test}$ will confine the agent in in-distribution belief states with high probability. In the subsequent

adaptation episodes with current belief $b_{t'}^{\kappa,\mu}$ in $\kappa_{test}$, the task hypothesis $(\kappa_{i*}, \mu_{i*})$ is also in the belief $b_{t'}^{\kappa,\mu}$ by induction.

Therefore, for each adaptation episode with current belief $b_{t'}^{\kappa,\mu}$ in $\kappa_{test}$, there exists a task hypothesis $(\kappa_i, \mu_i)$ from $b_{t'}^{\kappa,\mu}$, e.g., $(\kappa_{i*}, \mu_{i*})$, executing $\bar{\pi}^{+,T}$ with $(b_{t'}^{\kappa,\mu}, \kappa_i, \mu_i)$ in $\kappa_{test}$ will confine the agent in in-distribution belief states with high probability, as the offline dataset $\overline{\mathcal{D}}^+$ grows. $\qquad\square$

### A.6 Omitted Lemmas for Theorem 3

**Lemma 4.** *In an MDP $M$ and an arbitrary offline dataset $\mathcal{D}$, for each policy $\pi$,*

$$p_{out}^{M,\mathcal{D}}(\pi) \le H \left( \left\| \rho_\pi^M(s) - \rho_\pi^{M_\mathcal{D}}(s) \right\|_1 + \left\| \rho_\pi^M(s,a,r) - \rho_\pi^{M_\mathcal{D}}(s,a,r) \right\|_1 \right), \qquad (121)$$

*where $\rho_\pi^M(s,a,r)$ is the visitation distribution of $(s,a,r)$ in $M$, as defined in Eq. (12).*

*Proof.* For each $\tau_H$ leaving $\mathcal{D}$, we use the first outlier data $(s_t \notin \mathcal{D}$ or $(s_t, a_t, r_t) \notin \mathcal{D})$ to represent $\tau_H$. Thus,

$$p_{out}^{M,\mathcal{D}}(\pi) = \sum_{\tau_H} p_\pi^M(\tau_H) \, \mathbb{1}\left[\exists t \in [H] \text{ s.t. } s_t \notin \mathcal{D} \text{ or } (s_t, a_t, r_t) \notin \mathcal{D}\right] \qquad (122)$$

$$\le H \left( \sum_{s \in \mathcal{S} \text{ with } \rho_\pi^{M_\mathcal{D}}(s)=0} \rho_\pi^M(s) + \sum_{s \in \mathcal{S}, a \in \mathcal{A}, r \in \mathcal{R} \text{ with } \rho_\pi^{M_\mathcal{D}}(s,a,r)=0} \rho_\pi^M(s,a,r) \right) \qquad (123)$$

$$\le H \left( \left\| \rho_\pi^M(s) - \rho_\pi^{M_\mathcal{D}}(s) \right\|_1 + \left\| \rho_\pi^M(s,a,r) - \rho_\pi^{M_\mathcal{D}}(s,a,r) \right\|_1 \right). \qquad (124)$$

$$\square$$

**Lemma 5.** *In a transformed BAMDP $\overline{M}^+$ with an offline multi-task dataset $\overline{\mathcal{D}}^+$ collected by task-dependent behavior policies $[\mu]$, denoting the i.i.d. offline meta-training tasks sampled from $p(\kappa)$ in $\overline{\mathcal{D}}^+$ by $\mathcal{K}_{train}$, for each meta-testing task $\kappa_{test} \sim p(\kappa)$, and denoting the closest offline meta-training task to $\kappa_{test}$ by $\kappa_{i*}$, i.e.,*

$$\kappa_{i*} = \underset{\kappa_i \in \mathcal{K}_{train}}{\arg\min} \, \|\kappa_i - \kappa_{test}\|_\infty \qquad (125)$$

$$= \underset{\kappa_i \in \mathcal{K}_{train}}{\arg\min} \, \max\left( \|P^{\kappa_i}(s,a,s') - P^{\kappa_{test}}(s,a,s')\|_\infty, \|R^{\kappa_i}(s,a,r) - R^{\kappa_{test}}(s,a,r)\|_\infty \right), \qquad (126)$$

*then for any batch-constrained policy $\pi$ in $\mathcal{D}_{\kappa_{i*},\mu_{i*}}$, where $\mathcal{D}_{\kappa_{i*},\mu_{i*}}$ is a sub-dataset collected in $\overline{\mathcal{D}}^+$ (see Definition 5), i.e., $\forall \pi \in \Pi^{\mathcal{D}_{\kappa_{i*},\mu_{i*}}}$,*

$$p_{out}^{\kappa_{test}, \mathcal{D}_{\kappa_{i*},\mu_{i*}}}(\pi) \qquad (127)$$

$$\le 2H^2 |\mathcal{S}|^2 |\mathcal{A}| |\mathcal{R}| \left( \underbrace{\sqrt{\frac{\log\left(\frac{1}{\delta}\right) + \log\left(2|\mathcal{S}|^2|\mathcal{A}|\right)}{K_{\kappa_{i*},\mu_{i*}} \cdot d_\mu^{M_{\mathcal{D}_{\kappa_{i*},\mu_{i*}}}}}}_{\substack{\textit{Asymptotically approaches zero,} \\ \textit{when } \mathcal{D}_{\kappa_{i*},\mu_{i*}} \textit{ is sufficiently large}}} + \underbrace{2\left(\frac{\log\left(\frac{1}{\delta}\right)}{|\mathcal{K}_{train}|}\right)^{\frac{1}{|\mathcal{S}||\mathcal{A}|(|\mathcal{S}|+|\mathcal{R}|)}}}_{\substack{\textit{Asymptotically approaches zero,} \\ \textit{when } \mathcal{K}_{train} \textit{ is sufficiently large}}} \right), \qquad (128)$$

*where $K_{\kappa_{i*},\mu_{i*}}$ is the number of trajectories in the sub-dataset $\mathcal{D}_{\kappa_{i*},\mu_{i*}}$, $d_\mu^{M_{\mathcal{D}_{\kappa_{i*},\mu_{i*}}}}$ is the minimal visitation state-action distribution induced by the behavior policy $\mu$ in $M_{\mathcal{D}_{\kappa_{i*},\mu_{i*}}}$ (see Eq. (19)), and $|\mathcal{K}_{train}|$ is the number of i.i.d. offline meta-training tasks sampled from $p(\kappa)$ in $\overline{\mathcal{D}}^+$.*

*Proof.* According to Lemma 4,

$$p_{out}^{\kappa_{test}, \mathcal{D}_{\kappa_{i*},\mu_{i*}}}(\pi) \qquad (129)$$

$$\leq H\left(\left\|\rho_\pi^{M_{\mathcal{D}_{\kappa_{i*},\mu_{i*}}}}(s) - \rho_\pi^{\kappa_{test}}(s)\right\|_1 + \left\|\rho_\pi^{M_{\mathcal{D}_{\kappa_{i*},\mu_{i*}}}}(s,a,r) - \rho_\pi^{\kappa_{test}}(s,a,r)\right\|_1\right) \tag{130}$$

$$\leq H\left(\left\|\rho_\pi^{M_{\mathcal{D}_{\kappa_{i*},\mu_{i*}}}}(s) - \rho_\pi^{\kappa_{i*}}(s)\right\|_1 + \left\|\rho_\pi^{M_{\mathcal{D}_{\kappa_{i*},\mu_{i*}}}}(s,a,r) - \rho_\pi^{\kappa_{i*}}(s,a,r)\right\|_1\right) + \tag{131}$$

$$H\left(\left\|\rho_\pi^{\kappa_{i*}}(s) - \rho_\pi^{\kappa_{test}}(s)\right\|_1 + \left\|\rho_\pi^{\kappa_{i*}}(s,a,r) - \rho_\pi^{\kappa_{test}}(s,a,r)\right\|_1\right). \tag{132}$$

**Part I** Similar with Lemma 3, $\forall \delta \in (0,1]$, with probability $1-\delta$, denote

$$\epsilon_P = \max_{s\in\mathcal{S}, a\in\mathcal{A} \text{ with } \rho_\pi^{M_{\mathcal{D}_{\kappa_{i*},\mu_{i*}}}}(s,a)>0} \left\|P^{M_{\mathcal{D}_{\kappa_{i*},\mu_{i*}}}}(\cdot|s,a) - P^{\kappa_{i*}}(\cdot|s,a)\right\|_1 \tag{133}$$

$$\leq |\mathcal{S}|\sqrt{\frac{\log\left(\frac{1}{\delta}\right) + \log\left(2|\mathcal{S}|^2|\mathcal{A}|\right)}{K_{\kappa_{i*},\mu_{i*}} \cdot d_\mu^{M_{\mathcal{D}_{\kappa_{i*},\mu_{i*}}}}}}, \tag{134}$$

$$\epsilon_r = \max_{s\in\mathcal{S}, a\in\mathcal{A} \text{ with } \rho_\pi^{M_{\mathcal{D}_{\kappa_{i*},\mu_{i*}}}}(s,a)>0} \left|r^{M_{\mathcal{D}_{\kappa_{i*},\mu_{i*}}}}(s,a) - r^{\kappa_{i*}}(s,a)\right| \tag{135}$$

$$\leq \sqrt{\frac{\log\left(\frac{1}{\delta}\right) + \log\left(2|\mathcal{S}||\mathcal{A}|\right)}{K_{\kappa_{i*},\mu_{i*}} \cdot d_\mu^{M_{\mathcal{D}_{\kappa_{i*},\mu_{i*}}}}}}, \tag{136}$$

thus, from Lemma 6, we have

$$H\left(\left\|\rho_\pi^{M_{\mathcal{D}_{\kappa_{i*},\mu_{i*}}}}(s) - \rho_\pi^{\kappa_{i*}}(s)\right\|_1 + \left\|\rho_\pi^{M_{\mathcal{D}_{\kappa_{i*},\mu_{i*}}}}(s,a,r) - \rho_\pi^{\kappa_{i*}}(s,a,r)\right\|_1\right) \tag{137}$$

$$\leq 2H^2|\mathcal{S}|^2|\mathcal{A}||\mathcal{R}|\sqrt{\frac{\log\left(\frac{1}{\delta}\right) + \log\left(2|\mathcal{S}|^2|\mathcal{A}|\right)}{K_{\kappa_{i*},\mu_{i*}} \cdot d_\mu^{M_{\mathcal{D}_{\kappa_{i*},\mu_{i*}}}}}}. \tag{138}$$

**Part II** From Lemma 10, $\forall \delta \in (0,1]$, with probability $1-\delta$, denote

$$\tilde{\epsilon}_P = \max_{s\in\mathcal{S},a\in\mathcal{A}} \left\|P^{\kappa_{i*}}(\cdot|s,a) - P^{\kappa_{test}}(\cdot|s,a)\right\|_1 \tag{139}$$

$$\leq |\mathcal{S}|\left\|\kappa_{i*} - \kappa_{test}\right\|_\infty \tag{140}$$

$$\leq 2|\mathcal{S}|\left(\frac{\log\left(\frac{1}{\delta}\right)}{|\mathcal{K}_{train}|}\right)^{\frac{1}{|\mathcal{S}||\mathcal{A}|(|\mathcal{S}|+|\mathcal{R}|)}}, \tag{141}$$

$$\tilde{\epsilon}_R = \max_{s\in\mathcal{S},a\in\mathcal{A},r\in\mathcal{R}} \left|R^{\kappa_{i*}}(r|s,a) - R^{\kappa_{test}}(r|s,a)\right| \tag{142}$$

$$\leq \left\|\kappa_{i*} - \kappa_{test}\right\|_\infty \tag{143}$$

$$\leq 2\left(\frac{\log\left(\frac{1}{\delta}\right)}{|\mathcal{K}_{train}|}\right)^{\frac{1}{|\mathcal{S}||\mathcal{A}|(|\mathcal{S}|+|\mathcal{R}|)}}, \tag{144}$$

then from Lemma 8, we have

$$H\left(\left\|\rho_\pi^{\kappa_{i*}}(s) - \rho_\pi^{\kappa_{test}}(s)\right\|_1 + \left\|\rho_\pi^{\kappa_{i*}}(s,a,r) - \rho_\pi^{\kappa_{test}}(s,a,r)\right\|_1\right) \tag{145}$$

$$\leq 4H^2|\mathcal{S}|^2|\mathcal{A}||\mathcal{R}|\left(\frac{\log\left(\frac{1}{\delta}\right)}{|\mathcal{K}_{train}|}\right)^{\frac{1}{|\mathcal{S}||\mathcal{A}|(|\mathcal{S}|+|\mathcal{R}|)}}. \tag{146}$$

**Overall** Combining Part I and Part II, we have

$$p_{out}^{\kappa_{test},\mathcal{D}_{\kappa_{i*},\mu_{i*}}}(\pi) \tag{147}$$

$$\leq H\left(\left\|\rho_\pi^{M_{\mathcal{D}_{\kappa_{i*},\mu_{i*}}}}(s) - \rho_\pi^{\kappa_{i*}}(s)\right\|_1 + \left\|\rho_\pi^{M_{\mathcal{D}_{\kappa_{i*},\mu_{i*}}}}(s,a,r) - \rho_\pi^{\kappa_{i*}}(s,a,r)\right\|_1\right) + \tag{148}$$

$$H\left(\left\|\rho_\pi^{\kappa_{i*}}(s) - \rho_\pi^{\kappa_{test}}(s)\right\|_1 + \left\|\rho_\pi^{\kappa_{i*}}(s,a,r) - \rho_\pi^{\kappa_{test}}(s,a,r)\right\|_1\right) \tag{149}$$

$$\leq 2H^2 |\mathcal{S}|^2 |\mathcal{A}| |\mathcal{R}| \left( \sqrt{\frac{\log\left(\frac{1}{\delta}\right) + \log\left(2|\mathcal{S}|^2|\mathcal{A}|\right)}{K_{\kappa_{i^*},\mu_{i^*}} \cdot d_\mu^{M_{\mathcal{D}_{\kappa_{i^*},\mu_{i^*}}}}}} + 2\left(\frac{\log\left(\frac{1}{\delta}\right)}{|\mathcal{K}_{train}|}\right)^{\frac{1}{|\mathcal{S}||\mathcal{A}|(|\mathcal{S}|+|\mathcal{R}|)}} \right). \tag{150}$$

$$\square$$

### A.6.1 Detailed Lemmas (Part I)

**Lemma 6.** *In an MDP $M$ with an offline dataset $\mathcal{D}$, for any batch-constrained policy $\pi \in \Pi^{\mathcal{D}}$, if*

$$\max_{s\in\mathcal{S},a\in\mathcal{A} \text{ with } \rho_\pi^{M_{\mathcal{D}}}(s,a)>0} \left\| P^{M_{\mathcal{D}}}(\cdot|s,a) - P^M(\cdot|s,a) \right\|_1 \leq \epsilon_P, \tag{151}$$

$$\max_{s\in\mathcal{S},a\in\mathcal{A},r\in\mathcal{R} \text{ with } \rho_\pi^{M_{\mathcal{D}}}(s,a)>0} \left| R^{M_{\mathcal{D}}}(r|s,a) - R^M(r|s,a) \right| \leq \epsilon_R, \tag{152}$$

*we have*

$$\left\| \rho_\pi^M(s) - \rho_\pi^{M_{\mathcal{D}}}(s) \right\|_1 + \left\| \rho_\pi^M(s,a,r) - \rho_\pi^{M_{\mathcal{D}}}(s,a,r) \right\|_1 \tag{153}$$

$$\leq |\mathcal{S}| (|\mathcal{A}||\mathcal{R}| + 1) \frac{H-1}{2}\epsilon_P + |\mathcal{S}||\mathcal{A}||\mathcal{R}|\epsilon_R. \tag{154}$$

*Proof.* For each $\hat{s} \in \mathcal{S}, \hat{a} \in \mathcal{A}$, create an auxiliary reward function $\tilde{r}(s,a) : \mathcal{S} \times \mathcal{A} \to [0,1]$: $\forall s \in \mathcal{S}, a \in \mathcal{A}$,

$$\tilde{r}(s,a) = \begin{cases} \dfrac{1}{H}, & \text{if } s = \hat{s} \text{ and } a = \hat{a}, \\ 0, & \text{otherwise.} \end{cases} \tag{155}$$

Denote $\widehat{M} = \left(\mathcal{S}, \mathcal{A}, \mathcal{R}, H, P^M, \tilde{r}\right)$ and $\widehat{M}_{\mathcal{D}} = \left(\mathcal{S}, \mathcal{A}, \mathcal{R}, H, P^{M_{\mathcal{D}}}, \tilde{r}\right)$. Using the offline Simulation Lemma shown in Lemma 2, for any batch-constrained policy $\pi \in \Pi^{\mathcal{D}}$,

$$\rho_\pi^M(\hat{s}, \hat{a}) = \mathcal{J}_{\widehat{M}}(\pi) \quad \text{and} \quad \rho_\pi^{M_{\mathcal{D}}}(\hat{s}, \hat{a}) = \mathcal{J}_{\widehat{M}_{\mathcal{D}}}(\pi). \tag{156}$$

Thus, $\epsilon_r = 0, r_{max} = \frac{1}{H}$, and

$$\left| \rho_\pi^M(\hat{s}, \hat{a}) - \rho_\pi^{M_{\mathcal{D}}}(\hat{s}, \hat{a}) \right| \leq \left| \mathcal{J}_{\widehat{M}}(\pi) - \mathcal{J}_{\widehat{M}_{\mathcal{D}}}(\pi) \right| \tag{157}$$

$$\leq H\epsilon_r + \frac{H(H-1)r_{max}}{2}\epsilon_P \tag{158}$$

$$= \frac{H-1}{2}\epsilon_P. \tag{159}$$

Similarly, we have $\forall s \in \mathcal{S}$,

$$\left| \rho_\pi^M(s) - \rho_\pi^{M_{\mathcal{D}}}(s) \right| \leq \frac{H-1}{2}\epsilon_P. \tag{160}$$

For any $s \in \mathcal{S}, a \in \mathcal{A}, r \in \mathcal{R}$,

$$\left| \rho_\pi^M(s,a,r) - \rho_\pi^{M_{\mathcal{D}}}(s,a,r) \right| \tag{161}$$

$$= \left| \rho_\pi^M(s,a)R^M(r|s,a) - \rho_\pi^{M_{\mathcal{D}}}(s,a)R^{M_{\mathcal{D}}}(r|s,a) \right| \tag{162}$$

$$\leq \left| \rho_\pi^M(s,a) - \rho_\pi^{M_{\mathcal{D}}}(s,a) \right| R^M(r|s,a) + \rho_\pi^{M_{\mathcal{D}}}(s,a) \left| R^M(r|s,a) - R^{M_{\mathcal{D}}}(r|s,a) \right| \tag{163}$$

$$\leq \left| \rho_\pi^M(s,a) - \rho_\pi^{M_{\mathcal{D}}}(s,a) \right| + \left| R^M(r|s,a) - R^{M_{\mathcal{D}}}(r|s,a) \right| \tag{164}$$

$$\leq \frac{H-1}{2}\epsilon_P + \epsilon_R. \tag{165}$$

Therefore,

$$\left\| \rho_\pi^M(s) - \rho_\pi^{M_{\mathcal{D}}}(s) \right\|_1 + \left\| \rho_\pi^M(s,a,r) - \rho_\pi^{M_{\mathcal{D}}}(s,a,r) \right\|_1 \tag{166}$$

$$\leq |\mathcal{S}| \frac{H-1}{2}\epsilon_P + |\mathcal{S}||\mathcal{A}||\mathcal{R}| \left( \frac{H-1}{2}\epsilon_P + \epsilon_R \right) \tag{167}$$

$$= |\mathcal{S}| (|\mathcal{A}||\mathcal{R}| + 1) \frac{H-1}{2}\epsilon_P + |\mathcal{S}||\mathcal{A}||\mathcal{R}|\epsilon_R. \tag{168}$$

$$\square$$

A.6.2 DETAILED LEMMAS (PART II)

**Lemma 7** (Simulation Lemma for Finite-Horizon MDPs). *Given a pair of finite-horizon MDPs $M_1$ and $M_2$ with the same state space $\mathcal{S}$, same action space $\mathcal{A}$, same reward function $\mathcal{R}$, and same horizon $H$. If*

$$\max_{s \in \mathcal{S}, a \in \mathcal{A}} \left\| P^{M_1}(\cdot|s,a) - P^{M_2}(\cdot|s,a) \right\|_1 \leq \tilde{\epsilon}_P, \tag{169}$$

$$\max_{s \in \mathcal{S}, a \in \mathcal{A}} \left| r^{M_1}(s,a) - r^{M_2}(s,a) \right| \leq \tilde{\epsilon}_r, \tag{170}$$

$$\max_{s \in \mathcal{S}, a \in \mathcal{A}} \max \left( r^{M_1}(s,a), r^{M_2}(s,a) \right) \leq r_{max}, \tag{171}$$

*where $r^{M_1}(s,a) = \mathbb{E}_{\tilde{r} \sim R^{M_1}(s,a)}[\tilde{r}]$ and $r^{M_2}(s,a) = \mathbb{E}_{\tilde{r} \sim R^{M_2}(s,a)}[\tilde{r}]$, we have*

$$\left| \mathcal{J}_{M_1}(\pi) - \mathcal{J}_{M_2}(\pi) \right| \leq H\tilde{\epsilon}_r + \frac{H(H-1)r_{max}}{2}\tilde{\epsilon}_P. \tag{172}$$

*Proof.* Similar with famous Simulation Lemma in finite-horizon MDPs (Alekh Agarwal, 2017) and the offline variant shown in Lemma 2, we will prove $\forall h \in [H], \forall s \in \mathcal{S}_h$,

$$\left| V_\pi^{M_1}(s) - V_\pi^{M_2}(s) \right| \leq (H-h)\tilde{\epsilon}_r + \frac{(H-h)(H-h-1)r_{max}}{2}\tilde{\epsilon}_P \tag{173}$$

by induction. When $h = H - 1$, we have $\forall s \in \mathcal{S}_h$, $\left| V_\pi^{M_1}(s) - V_\pi^{M_2}(s) \right| \leq \tilde{\epsilon}_r$ holds. And $\forall h \in [H-1], \forall s \in \mathcal{S}_h$,

$$\left| V_\pi^{M_1}(s) - V_\pi^{M_2}(s) \right| \tag{174}$$

$$\leq \sum_{a \in \mathcal{A}} \pi(a|s) \left| r^{M_1}(s,a) - r^{M_2}(s,a) \right| + \tag{175}$$

$$\sum_{a \in \mathcal{A}} \pi(a|s) \sum_{s' \in \mathcal{S}_{h+1}} \left| P^{M_1}(s'|s,a)V_\pi^{M_1}(s') - P^{M_2}(s'|s,a)V_\pi^{M_2}(s') \right| \tag{176}$$

$$\leq (H-h)\tilde{\epsilon}_r + \frac{(H-h)(H-h-1)r_{max}}{2}\tilde{\epsilon}_P. \tag{177}$$

Thus,

$$\left| \mathcal{J}_{M_1}(\pi) - \mathcal{J}_{M_2}(\pi) \right| = \left| V_\pi^{M_1}(s_0) - V_\pi^{M_2}(s_0) \right| \tag{178}$$

$$\leq H\tilde{\epsilon}_r + \frac{H(H-1)r_{max}}{2}\tilde{\epsilon}_P. \tag{179}$$

$\square$

**Lemma 8.** *Given a pair of finite-horizon MDPs $M_1$ and $M_2$ with the same state space, same action space, same reward function, and same horizon. If*

$$\max_{s \in \mathcal{S}, a \in \mathcal{A}} \left\| P^{M_1}(\cdot|s,a) - P^{M_2}(\cdot|s,a) \right\|_1 \leq \tilde{\epsilon}_P, \tag{180}$$

$$\max_{s \in \mathcal{S}, a \in \mathcal{A}, r \in \mathcal{R}} \left| R^{M_1}(r|s,a) - R^{M_2}(r|s,a) \right| \leq \tilde{\epsilon}_R, \tag{181}$$

*we have*

$$\left\| \rho_\pi^{M_1}(s) - \rho_\pi^{M_2}(s) \right\|_1 + \left\| \rho_\pi^{M_1}(s,a,r) - \rho_\pi^{M_2}(s,a,r) \right\|_1 \tag{182}$$

$$\leq |\mathcal{S}| \left( |\mathcal{A}| |\mathcal{R}| + 1 \right) \frac{H-1}{2}\tilde{\epsilon}_P + |\mathcal{S}| |\mathcal{A}| |\mathcal{R}| \tilde{\epsilon}_R. \tag{183}$$

*Proof.* Similar with Lemma 6, $\forall s \in \mathcal{S}, a \in \mathcal{A}, r \in \mathcal{R}$, using Simulation Lemma 7, we have

$$\left| \rho_\pi^{M_1}(s) - \rho_\pi^{M_2}(s) \right| \leq \frac{H-1}{2}\tilde{\epsilon}_P \quad \text{and} \quad \left| \rho_\pi^{M_1}(s,a) - \rho_\pi^{M_2}(s,a) \right| \leq \frac{H-1}{2}\tilde{\epsilon}_P, \tag{184}$$

and

$$\left| \rho_\pi^{M}(s,a,r) - \rho_\pi^{M_\mathcal{D}}(s,a,r) \right| \leq \frac{H-1}{2}\tilde{\epsilon}_P + \tilde{\epsilon}_R. \tag{185}$$

Therefore,

$$\left\| \rho_\pi^{M_1}(s) - \rho_\pi^{M_2}(s) \right\|_1 + \left\| \rho_\pi^{M_1}(s,a,r) - \rho_\pi^{M_2}(s,a,r) \right\|_1 \tag{186}$$

$$\leq |\mathcal{S}| \left( |\mathcal{A}| |\mathcal{R}| + 1 \right) \frac{H-1}{2} \tilde{\epsilon}_P + |\mathcal{S}| |\mathcal{A}| |\mathcal{R}| \tilde{\epsilon}_R. \tag{187}$$

$\square$

**Lemma 9.** *Let $X, Y$ be two i.i.d. random vectors that take values in $[0,1]^n$, $n \in \mathbb{N}^+$. For any $\epsilon \in (0,1]$, we have*

$$\mathbb{P} \left[ \max_{i \in [n]} |X_i - Y_i| \geq \epsilon \right] \leq 1 - \left( \frac{\epsilon}{2} \right)^n. \tag{188}$$

*Proof.* Denote an auxiliary set

$$V = \left\{ x \in \mathbb{R}^n \left| \max_{i \in [n]} |x_i| < \frac{\epsilon}{2} \right. \right\}, \tag{189}$$

then if $X, Y \in V$, we must have

$$\max_{i \in [n]} |X_i - Y_i| < \epsilon. \tag{190}$$

For any $c \in \mathbb{N}^n$, denote

$$V^c = V + v^c, \quad \text{where} \quad v_i^c = \left( c_i + \frac{1}{2} \right) \epsilon, \quad \forall i \in [n]. \tag{191}$$

We may construct a set of such cosets of $V$ as follows:

$$S = \{ V^c | c \in C \}, \quad \text{where} \quad C = \left\{ c \in \mathbb{N}^n \left| c_i \in \left[ \left\lceil \frac{1}{\epsilon} \right\rceil \right] \right. \right\}. \tag{192}$$

There are several properties related to these constructions:

- For any $c \in \mathbb{N}^n$, if $X, Y \in V^c$, $\max_{i \in [n]} |X_i - Y_i| < \epsilon$.

- The union of sets in $S$ contains $[0,1]^n$

- Any two different sets in $S$ are disjoint.

The only loophole is that we have not considered points in boundaries $\partial V^c$ ($V^c \in S$). These boundaries can be decomposed into disjoint union of hyperplanes in $\mathbb{R}^n$. For each one of the hyperplanes, arbitrarily designate it to an adjacent $V^c \in S$. New $V^c$s are the union of the original one and the hyperplanes designated to it. Note that

$$\sum_{c \in C} [X \in V^c] = 1. \tag{193}$$

Therefore,

$$\mathbb{P} \left[ \max_{i \in [n]} |X_i - Y_i| \geq \epsilon \right] \leq 1 - \sum_{c \in C} \mathbb{P} [X \in V^c] \, \mathbb{P} [Y \in V^c] \tag{194}$$

$$= 1 - \sum_{c \in C} \mathbb{P} [X \in V^c]^2 \tag{195}$$

$$\leq 1 - \frac{1}{|C|} \left( \sum_{c \in C} \mathbb{P} [X \in V^c] \right)^2 \tag{196}$$

$$= 1 - \frac{1}{|C|}. \tag{197}$$

Since $\frac{1}{\epsilon} \geq 1$, we have

$$|C| = \left\lceil \frac{1}{\epsilon} \right\rceil^n < \left( 1 + \frac{1}{\epsilon} \right)^n \leq \left( \frac{2}{\epsilon} \right)^n \quad \text{and} \quad \mathbb{P} \left[ \max_{i \in [n]} |X_i - Y_i| \geq \epsilon \right] \leq 1 - \left( \frac{\epsilon}{2} \right)^n. \tag{198}$$

$\square$

**Lemma 10.** *In a transformed BAMDP $\overline{M}^+$ with an offline multi-task dataset $\overline{\mathcal{D}}^+$, for any meta-testing task $\kappa_{test} \sim p(\kappa)$, $\forall \delta \in (0, 1]$, with probability $1 - \delta$, we have*

$$\|\kappa_{i*} - \kappa_{test}\|_\infty \tag{199}$$

$$= \max\left(\|P^{\kappa_{i*}}(s, a, s') - P^{\kappa_{test}}(s, a, s')\|_\infty, \|R^{\kappa_{i*}}(s, a, r) - R^{\kappa_{test}}(s, a, r)\|_\infty\right) \tag{200}$$

$$\leq 2\left(\frac{\log\left(\frac{1}{\delta}\right)}{|\mathcal{K}_{train}|}\right)^{\frac{1}{|\mathcal{S}||\mathcal{A}|(|\mathcal{S}|+|\mathcal{R}|)}}, \tag{201}$$

*where $\kappa_{i*} \in \mathcal{K}_{train}$ is the closest offline meta-training task to $\kappa_{test}$ (see Eq. (125)), $\|\kappa_{i*} - \kappa_{test}\|_\infty$ is the distance between $\kappa_{i*}$ and $\kappa_{test}$ (see Eq. (126)), and $\mathcal{K}_{train}$ is the i.i.d. offline meta-training tasks sampled from $p(\kappa)$ in $\overline{\mathcal{D}}^+$.*

*Proof.* From Lemma 9, we set $n = |\mathcal{S}||\mathcal{A}|(|\mathcal{S}| + |\mathcal{R}|)$, then $\forall \epsilon \in (0, 1], \forall \kappa_i \in \mathcal{K}_{train}$,

$$\mathbb{P}\left[\|\kappa_i - \kappa_{test}\|_\infty \geq \epsilon\right] \leq 1 - \left(\frac{\epsilon}{2}\right)^n. \tag{202}$$

Hence

$$\mathbb{P}\left[\arg\min_{\kappa_i \in \mathcal{K}_{train}} \|\kappa_i - \kappa_{test}\|_\infty \geq \epsilon\right] = \prod_{\kappa_i \in \mathcal{K}_{train}} \mathbb{P}\left[\|\kappa_i - \kappa_{test}\|_\infty \geq \epsilon\right] \tag{203}$$

$$\leq \left(1 - \left(\frac{\epsilon}{2}\right)^n\right)^{|\mathcal{K}_{train}|}. \tag{204}$$

Therefore, $\forall \delta \in (0, 1]$, with probability $1 - \delta$,

$$\arg\min_{\kappa_i \in \mathcal{K}_{train}} \|\kappa_i - \kappa_{test}\|_\infty \leq 2\left(\frac{\log\left(\frac{1}{\delta}\right)}{|\mathcal{K}_{train}|}\right)^{\frac{1}{|\mathcal{S}||\mathcal{A}|(|\mathcal{S}|+|\mathcal{R}|)}}. \tag{205}$$

$\square$

# B  HYPER-PARAMETER SETTINGS

**Environment Settings.** Table 3 shows hyper-parameter settings for the task sets used in our experiments. Most hyper-parameters are adopted from previous works (Li et al., 2020; Mitchell et al., 2021). For all task sets, 80% of the tasks are meta-training tasks, and the remaining 20% tasks are meta-testing tasks.

Table 3: Environment parameter settings.

| Environment | Episode Length | # of Adaptation Episodes | # of Tasks | # of Trajectories per Task |
|---|---|---|---|---|
| All Meta-World Envs | 500 | 10 | 50 | 45 |
| Cheetah-Vel | 200 | 10 | 100 | 45 |
| Point-Robot | 20 | 20 | 100 | 45 |
| Point-Robot-Sparse | 20 | 20 | 100 | 45 |

**GCC hyper-parameter settings.** Table 4 shows GCC's hyper-parameter settings. Most hyper-parameters are adopted from FOCAL (Li et al., 2020). We set $n_e$ to 1 as the evaluation environments are all nearly deterministic.

Table 4: Detailed hyper-parameter settings for GCC.

| Hyper-Parameter | Hyper-Parameter Values |
|---|---|
| batch size | 256 |
| meta batch size | 16 |
| learning rates for dual critic | 1e-4 |
| learning rates for all other components | 3e-4 |
| network structure for all components | three fully connected layers with 200 units |
| optimizer | adam |
| discount | 0.99 |
| latent size | 20 |
| reward scale | 100 for point envs, 1 for all other envs |
| $n_{it}$ | 1/2 of total adaptation episodes |
| $n_0^z$ | 1/2 of total adaptation episodes |
| $n_{\bar{z}}$ | 5 |
| $n_e$ | 1 |

## C  Analysis and Visualization on Uncertainty Estimation

To further investigate why the uncertainty estimation method (Kidambi et al., 2020) fails to identify in-distribution trajectories, we demonstrate the ensemble's prediction errors and uncertainty estimation on the first 10 trajectories. As shown in Figure 5, uncertainty estimation methods cannot accurately estimate the distance to the offline dataset, and fail to identify in-distribution data. On the other hand, GCC can successfully select in-distribution data with its greedy selection mechanism.

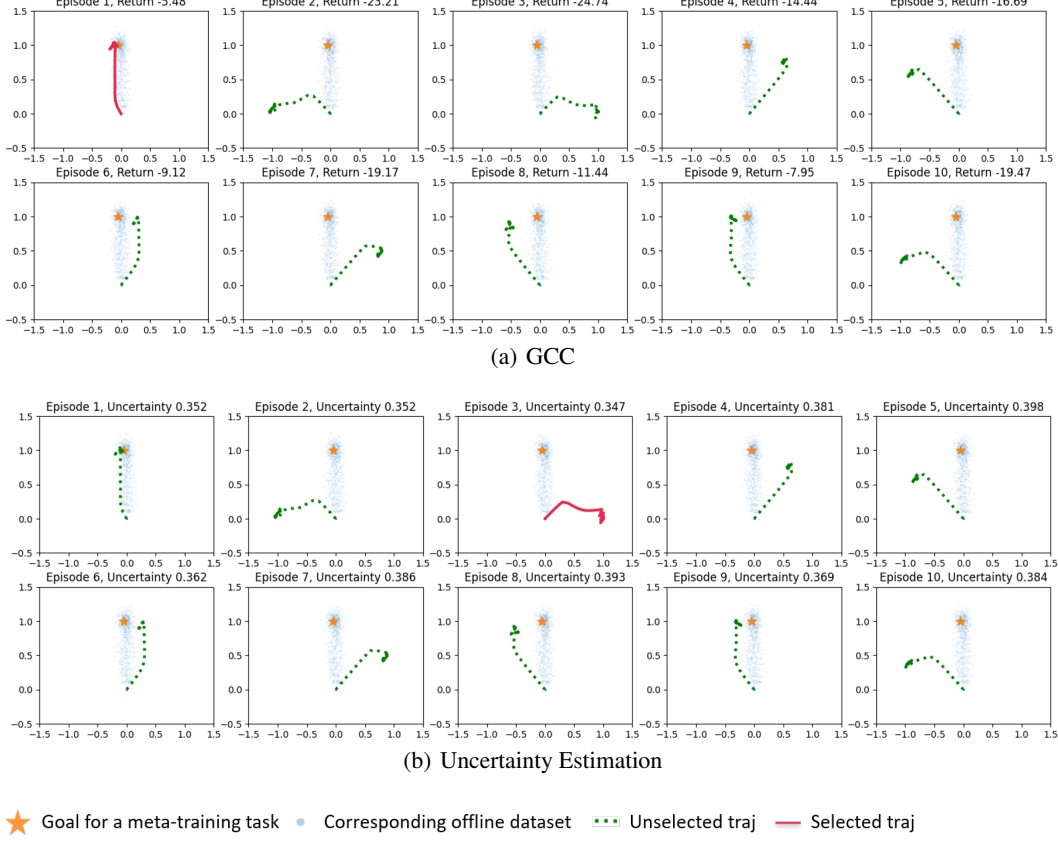

Figure 5: Visualization of GCC and uncertainty estimation method's trajectory selection. (a) GCC successfully selects the in-distribution trajectory. (b) Uncertainty estimation method cannot identify in-distribution data, as its uncertainty estimation is not accurate.

## D  Additional Visualization Results

Figure 6 shows GCC, FOCAL, and uncertainty estimation method (Kidambi et al., 2020) 's adaptation visualization (episode 11-20) after the initial exploration phase in adaptation (episode 1-10). Results demonstrate that while GCC is able to identify in-distribution data and achieve superior adaptation performance, FOCAL utilizes all the 10 trajectories for adaptation, and cannot correctly update task belief due to the out-of-distribution issue. The uncertainty estimation method, as discussed in Appendix C, fails to correctly select the in-distribution trajectory, and thus cannot successfully reach the goal.

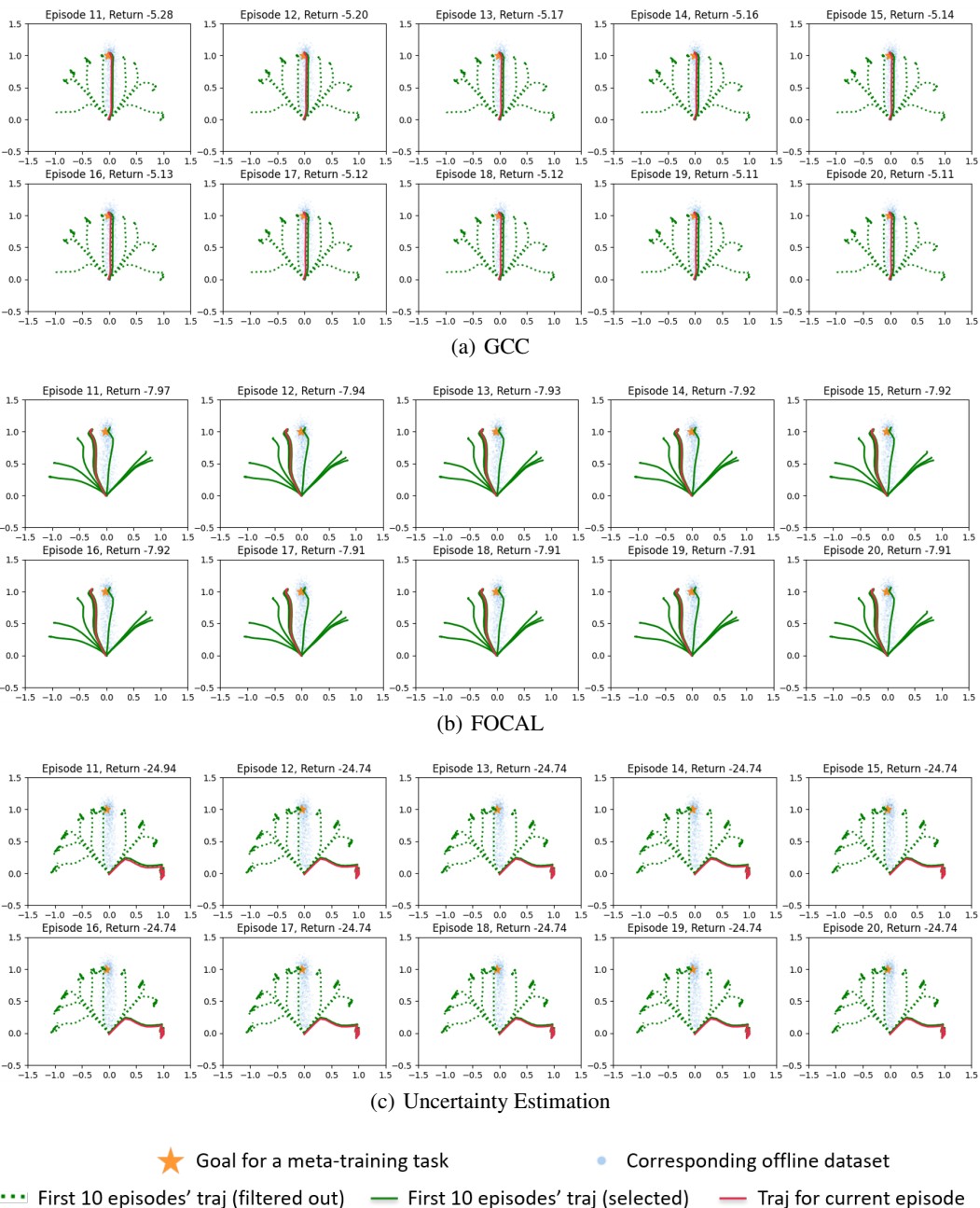

Figure 6: Visualization of GCC, FOCAL and uncertainty estimation method's adaptation in Episode 11-20 on the Point-Robot environment. (a) GCC successfully selects the in-distribution trajectory and keeps improving in adaptation. (b) FOCAL suffers from the out-of-distribution issue, and cannot correctly update posterior belief, leading to poor performance. (c) Uncertainty estimation method cannot identify in-distribution data, and also suffers from the out-of-distribution issue.

# E  ADDITIONAL EXPERIMENT RESULTS

## E.1  UNCERTAINTY ESTIMATION METHOD ON REPRESENTATIVE TASKS

Table 5 shows the uncertainty estimation method's online adaptation performance on several representative tasks. Results demonstrate that the uncertainty estimation method achieves similar performance to FOCAL, and underperforms GCC. This is because the uncertainty estimation method cannot correctly estimate in-distribution trajectories, as discussed in Section 5.1, Appendix C and D.

Table 5: Comparison between GCC, FOCAL, and the uncertainty estimation method's performance on several representative tasks. For Meta-World tasks we show normalized scores.

| Env | GCC | Uncertainty Estimation | FOCAL |
|---|---|---|---|
| Plate-Slide-Back | **0.96** $\pm$ 0.02 | 0.64 $\pm$ 0.21 | 0.58 $\pm$ 0.06 |
| Hammer | **0.84** $\pm$ 0.06 | 0.61 $\pm$ 0.33 | 0.59 $\pm$ 0.07 |
| Coffee-Push | **1.26** $\pm$ 0.13 | 0.02 $\pm$ 0.01 | 0.66 $\pm$ 0.07 |
| Push-Wall | **0.71** $\pm$ 0.15 | 0.48 $\pm$ 0.19 | 0.43 $\pm$ 0.06 |
| Point-Robot | **-5.10** $\pm$ 0.26 | -19.62 $\pm$ 0.77 | -15.38 $\pm$ 0.95 |

## E.2  DETAILED ALGORITHM PERFORMANCE ON ALL TASKS

Table 6, Table 7, and Table 8 shows baselines' online adaptation and offline performance on all 50 Meta-World ML1 task sets, respectively. GCC significantly outperforms baselines with online adaptation, and achieves better or comparable performance to baselines with offline adaptation.

Table 6: Comparison between FOCAL and MACAW with online adaptation and GCC. For Meta-World tasks, "-V2" is omitted for brevity.

| Environment | GCC | FOCAL | MACAW |
|---|---|---|---|
| Coffee-Push | **1.26** ± 0.13 | 0.66 ± 0.07 | 0.01 ± 0.01 |
| Faucet-Close | **1.12** ± 0.01 | 1.06 ± 0.02 | 0.07 ± 0.01 |
| Door-Unlock | **1.11** ± 0.02 | 0.97 ± 0.03 | 0.11 ± 0.01 |
| Plate-Slide-Side | **1.07** ± 0.08 | 0.70 ± 0.14 | 0.00 ± 0.00 |
| Faucet-Open | **1.05** ± 0.02 | 1.01 ± 0.02 | 0.08 ± 0.04 |
| Button-Press-Wall | **1.04** ± 0.04 | 0.99 ± 0.06 | 0.02 ± 0.00 |
| Plate-Slide | **1.01** ± 0.03 | 0.83 ± 0.09 | 0.01 ± 0.00 |
| Door-Close | **0.99** ± 0.00 | 0.97 ± 0.01 | 0.00 ± 0.00 |
| Drawer-Close | **0.99** ± 0.02 | **0.96** ± 0.04 | 0.53 ± 0.50 |
| Plate-Slide-Back-Side | **0.97** ± 0.02 | 0.77 ± 0.20 | 0.02 ± 0.01 |
| Door-Lock | **0.97** ± 0.01 | 0.90 ± 0.02 | 0.25 ± 0.11 |
| Window-Open | **0.96** ± 0.02 | 0.81 ± 0.07 | 0.15 ± 0.11 |
| Door-Open | **0.96** ± 0.02 | 0.78 ± 0.13 | 0.06 ± 0.01 |
| Plate-Slide-Back | **0.96** ± 0.02 | 0.58 ± 0.06 | 0.21 ± 0.17 |
| Window-Close | **0.94** ± 0.01 | 0.79 ± 0.01 | 0.54 ± 0.44 |
| Reach-Wall | **0.93** ± 0.05 | 0.53 ± 0.18 | 0.82 ± 0.02 |
| Dial-Turn | **0.91** ± 0.05 | 0.84 ± 0.09 | 0.00 ± 0.00 |
| Handle-Press-Side | **0.91** ± 0.02 | 0.79 ± 0.10 | 0.51 ± 0.41 |
| Handle-Pull | **0.90** ± 0.02 | 0.67 ± 0.03 | 0.00 ± 0.00 |
| Handle-Press | **0.88** ± 0.05 | **0.87** ± 0.02 | 0.28 ± 0.10 |
| Reach | **0.85** ± 0.03 | 0.62 ± 0.05 | 0.63 ± 0.04 |
| Lever-Pull | **0.85** ± 0.02 | 0.73 ± 0.07 | 0.20 ± 0.16 |
| Hammer | **0.84** ± 0.06 | 0.59 ± 0.07 | 0.10 ± 0.01 |
| Drawer-Open | **0.82** ± 0.06 | 0.64 ± 0.10 | 0.11 ± 0.02 |
| Sweep | **0.77** ± 0.04 | 0.32 ± 0.08 | 0.20 ± 0.20 |
| Button-Press | **0.74** ± 0.08 | **0.68** ± 0.14 | 0.02 ± 0.01 |
| Stick-Push | **0.73** ± 0.09 | 0.46 ± 0.15 | 0.17 ± 0.17 |
| Coffee-Button | **0.73** ± 0.14 | **0.66** ± 0.16 | 0.15 ± 0.13 |
| Push-Wall | **0.71** ± 0.15 | 0.43 ± 0.06 | 0.23 ± 0.18 |
| Shelf-Place | **0.70** ± 0.18 | 0.32 ± 0.11 | 0.01 ± 0.01 |
| Basketball | **0.64** ± 0.15 | 0.41 ± 0.24 | 0.00 ± 0.00 |
| Hand-Insert | **0.63** ± 0.04 | 0.29 ± 0.07 | 0.02 ± 0.01 |
| Sweep-Into | **0.61** ± 0.06 | 0.33 ± 0.05 | 0.00 ± 0.00 |
| Button-Press-Topdown | **0.57** ± 0.11 | 0.45 ± 0.06 | 0.38 ± 0.36 |
| Peg-Unplug-Side | **0.56** ± 0.07 | 0.19 ± 0.09 | 0.00 ± 0.00 |
| Assembly | **0.55** ± 0.13 | 0.28 ± 0.05 | 0.33 ± 0.01 |
| Push | **0.55** ± 0.10 | 0.34 ± 0.14 | 0.28 ± 0.19 |
| Bin-Picking | 0.53 ± 0.16 | 0.31 ± 0.21 | **0.66** ± 0.11 |
| Push-Back | **0.52** ± 0.05 | 0.16 ± 0.04 | 0.00 ± 0.00 |
| Box-Close | **0.51** ± 0.11 | 0.15 ± 0.09 | 0.36 ± 0.11 |
| Soccer | **0.44** ± 0.04 | 0.11 ± 0.03 | **0.38** ± 0.31 |
| Button-Press-Topdown-Wall | **0.43** ± 0.03 | **0.40** ± 0.07 | 0.05 ± 0.02 |
| Disassemble | **0.42** ± 0.14 | 0.26 ± 0.04 | 0.05 ± 0.00 |
| Coffee-Pull | **0.40** ± 0.05 | 0.23 ± 0.04 | 0.19 ± 0.12 |
| Stick-Pull | **0.32** ± 0.06 | 0.17 ± 0.07 | 0.00 ± 0.00 |
| Peg-Insert-Side | **0.30** ± 0.04 | 0.08 ± 0.03 | 0.00 ± 0.00 |
| Pick-Place-Wall | 0.28 ± 0.12 | 0.09 ± 0.04 | **0.39** ± 0.25 |
| Pick-Out-Of-Hole | 0.26 ± 0.25 | 0.16 ± 0.16 | **0.59** ± 0.06 |
| Pick-Place | **0.20** ± 0.03 | 0.07 ± 0.02 | 0.05 ± 0.05 |
| Handle-Pull-Side | **0.14** ± 0.04 | **0.13** ± 0.09 | 0.00 ± 0.00 |
| Average | **0.73** ± 0.07 | 0.53 ± 0.08 | 0.18 ± 0.09 |
| Cheetah-Vel | **-171.52** ± 21.96 | -287.70 ± 30.62 | -233.97 ± 23.46 |
| Point-Robot | **-5.10** ± 0.26 | -15.38 ± 0.95 | -14.61 ± 0.98 |
| Point-Robot-Sparse | **7.78** ± 0.64 | 0.83 ± 0.37 | 0.00 ± 0.00 |

Table 7: Comparison between BOReL and GCC. For Meta-World tasks, "-V2" is omitted for brevity.

| Environment | GCC | BOReL |
|---|---|---|
| Coffee-Push | 1.26 ± 0.13 | 0.00 ± 0.00 |
| Faucet-Close | 1.12 ± 0.01 | 0.13 ± 0.03 |
| Door-Unlock | 1.11 ± 0.02 | 0.13 ± 0.03 |
| Plate-Slide-Side | 1.07 ± 0.08 | 0.00 ± 0.00 |
| Faucet-Open | 1.05 ± 0.02 | 0.12 ± 0.05 |
| Button-Press-Wall | 1.04 ± 0.04 | 0.01 ± 0.00 |
| Plate-Slide | 1.01 ± 0.03 | 0.01 ± 0.00 |
| Door-Close | 0.99 ± 0.00 | 0.37 ± 0.19 |
| Drawer-Close | 0.99 ± 0.02 | 0.00 ± 0.00 |
| Plate-Slide-Back-Side | 0.97 ± 0.02 | 0.01 ± 0.00 |
| Door-Lock | 0.97 ± 0.01 | 0.14 ± 0.00 |
| Window-Open | 0.96 ± 0.02 | 0.03 ± 0.00 |
| Door-Open | 0.96 ± 0.02 | 0.12 ± 0.01 |
| Plate-Slide-Back | 0.96 ± 0.02 | 0.01 ± 0.00 |
| Window-Close | 0.94 ± 0.01 | 0.03 ± 0.00 |
| Reach-Wall | 0.93 ± 0.05 | 0.06 ± 0.00 |
| Dial-Turn | 0.91 ± 0.05 | 0.00 ± 0.00 |
| Handle-Press-Side | 0.91 ± 0.02 | 0.02 ± 0.01 |
| Handle-Pull | 0.90 ± 0.02 | 0.00 ± 0.00 |
| Handle-Press | 0.88 ± 0.05 | 0.01 ± 0.00 |
| Reach | 0.85 ± 0.03 | 0.04 ± 0.01 |
| Lever-Pull | 0.85 ± 0.02 | 0.05 ± 0.00 |
| Hammer | 0.84 ± 0.06 | 0.09 ± 0.01 |
| Drawer-Open | 0.82 ± 0.06 | 0.10 ± 0.00 |
| Sweep | 0.77 ± 0.04 | 0.00 ± 0.00 |
| Button-Press | 0.74 ± 0.08 | 0.01 ± 0.01 |
| Stick-Push | 0.73 ± 0.09 | 0.00 ± 0.00 |
| Coffee-Button | 0.73 ± 0.14 | 0.02 ± 0.00 |
| Push-Wall | 0.71 ± 0.15 | 0.00 ± 0.00 |
| Shelf-Place | 0.70 ± 0.18 | 0.00 ± 0.00 |
| Basketball | 0.64 ± 0.15 | 0.00 ± 0.00 |
| Hand-Insert | 0.63 ± 0.04 | 0.00 ± 0.00 |
| Sweep-Into | 0.61 ± 0.06 | 0.01 ± 0.00 |
| Button-Press-Topdown | 0.57 ± 0.11 | 0.02 ± 0.02 |
| Peg-Unplug-Side | 0.56 ± 0.07 | 0.00 ± 0.00 |
| Assembly | 0.55 ± 0.13 | 0.04 ± 0.00 |
| Push | 0.55 ± 0.10 | 0.00 ± 0.00 |
| Bin-Picking | 0.53 ± 0.16 | 0.00 ± 0.00 |
| Push-Back | 0.52 ± 0.05 | 0.00 ± 0.00 |
| Box-Close | 0.51 ± 0.11 | 0.05 ± 0.01 |
| Soccer | 0.44 ± 0.04 | 0.04 ± 0.02 |
| Button-Press-Topdown-Wall | 0.43 ± 0.03 | 0.05 ± 0.01 |
| Disassemble | 0.42 ± 0.14 | 0.04 ± 0.00 |
| Coffee-Pull | 0.40 ± 0.05 | 0.00 ± 0.00 |
| Stick-Pull | 0.32 ± 0.06 | 0.00 ± 0.00 |
| Peg-Insert-Side | 0.30 ± 0.04 | 0.00 ± 0.00 |
| Pick-Place-Wall | 0.28 ± 0.12 | 0.00 ± 0.00 |
| Pick-Out-Of-Hole | 0.26 ± 0.25 | 0.00 ± 0.00 |
| Pick-Place | 0.20 ± 0.03 | 0.00 ± 0.00 |
| Handle-Pull-Side | 0.14 ± 0.04 | 0.00 ± 0.00 |
| Average | 0.73 ± 0.07 | 0.04 ± 0.01 |
| Cheetah-Vel | -171.52 ± 21.96 | -301.4 ± 36.8 |
| Point-Robot | -5.10 ± 0.26 | -17.28 ± 1.16 |
| Point-Robot-Sparse | 7.78 ± 0.64 | 0.00 ± 0.00 |

Table 8: Comparison between baselines with offline adaptation and GCC. For Meta-World tasks, "-V2" is omitted for brevity.

| Environment | GCC | FOCAL with Expert Context | MACAW with Expert Context |
|---|---|---|---|
| Coffee-Push | **1.26** ± 0.13 | 0.50 ± 0.06 | **1.14** ± 0.27 |
| Faucet-Close | **1.12** ± 0.01 | 1.07 ± 0.02 | 1.01 ± 0.01 |
| Door-Unlock | **1.11** ± 0.02 | 0.96 ± 0.03 | 0.99 ± 0.04 |
| Plate-Slide-Side | **1.07** ± 0.08 | 0.75 ± 0.09 | 0.91 ± 0.09 |
| Faucet-Open | **1.05** ± 0.02 | **1.00** ± 0.02 | 0.99 ± 0.01 |
| Button-Press-Wall | **1.04** ± 0.04 | 0.98 ± 0.05 | **0.99** ± 0.01 |
| Plate-Slide | **1.01** ± 0.03 | **1.00** ± 0.03 | 0.67 ± 0.07 |
| Door-Close | **0.99** ± 0.00 | 0.97 ± 0.01 | 0.92 ± 0.05 |
| Drawer-Close | **0.99** ± 0.02 | 0.96 ± 0.04 | **1.00** ± 0.01 |
| Plate-Slide-Back-Side | **0.97** ± 0.02 | 0.90 ± 0.07 | 0.80 ± 0.05 |
| Door-Lock | **0.97** ± 0.01 | 0.88 ± 0.04 | **0.97** ± 0.03 |
| Window-Open | **0.96** ± 0.02 | **0.93** ± 0.05 | **0.98** ± 0.02 |
| Door-Open | **0.96** ± 0.02 | 0.90 ± 0.02 | **0.99** ± 0.02 |
| Plate-Slide-Back | **0.96** ± 0.02 | **0.93** ± 0.01 | 0.55 ± 0.11 |
| Window-Close | 0.94 ± 0.01 | 0.73 ± 0.02 | **1.00** ± 0.01 |
| Reach-Wall | **0.93** ± 0.05 | **0.91** ± 0.05 | 0.82 ± 0.02 |
| Dial-Turn | 0.91 ± 0.05 | 0.84 ± 0.08 | **0.98** ± 0.01 |
| Handle-Press-Side | **0.91** ± 0.02 | 0.87 ± 0.04 | 0.82 ± 0.10 |
| Handle-Pull | **0.90** ± 0.02 | 0.70 ± 0.05 | **0.95** ± 0.05 |
| Handle-Press | **0.88** ± 0.05 | **0.79** ± 0.08 | 0.56 ± 0.19 |
| Reach | **0.85** ± 0.03 | **0.83** ± 0.05 | 0.64 ± 0.08 |
| Lever-Pull | 0.85 ± 0.02 | 0.76 ± 0.03 | **0.97** ± 0.07 |
| Hammer | **0.84** ± 0.06 | 0.78 ± 0.04 | 0.40 ± 0.18 |
| Drawer-Open | 0.82 ± 0.06 | 0.73 ± 0.11 | **0.98** ± 0.01 |
| Sweep | 0.77 ± 0.04 | 0.74 ± 0.02 | **0.98** ± 0.01 |
| Button-Press | **0.74** ± 0.08 | 0.63 ± 0.09 | 0.71 ± 0.04 |
| Stick-Push | **0.73** ± 0.09 | 0.14 ± 0.09 | 0.67 ± 0.09 |
| Coffee-Button | **0.73** ± 0.14 | **0.61** ± 0.20 | 0.21 ± 0.11 |
| Push-Wall | 0.71 ± 0.15 | **0.90** ± 0.12 | **0.96** ± 0.09 |
| Shelf-Place | **0.70** ± 0.18 | 0.57 ± 0.13 | 0.55 ± 0.03 |
| Basketball | **0.64** ± 0.15 | 0.49 ± 0.17 | 0.47 ± 0.18 |
| Hand-Insert | **0.63** ± 0.04 | **0.64** ± 0.09 | 0.20 ± 0.09 |
| Sweep-Into | **0.61** ± 0.06 | **0.64** ± 0.09 | 0.00 ± 0.00 |
| Button-Press-Topdown | 0.57 ± 0.11 | 0.48 ± 0.11 | **0.92** ± 0.04 |
| Peg-Unplug-Side | **0.56** ± 0.07 | **0.57** ± 0.10 | 0.18 ± 0.10 |
| Assembly | 0.55 ± 0.13 | **0.64** ± 0.03 | 0.36 ± 0.02 |
| Push | 0.55 ± 0.10 | **0.98** ± 0.13 | 0.86 ± 0.02 |
| Bin-Picking | 0.53 ± 0.16 | **0.61** ± 0.12 | **0.63** ± 0.11 |
| Push-Back | **0.52** ± 0.05 | **0.52** ± 0.16 | 0.15 ± 0.09 |
| Box-Close | 0.51 ± 0.11 | **0.56** ± 0.08 | 0.35 ± 0.11 |
| Soccer | 0.44 ± 0.04 | 0.45 ± 0.03 | **0.59** ± 0.11 |
| Button-Press-Topdown-Wall | **0.43** ± 0.03 | 0.40 ± 0.06 | **0.43** ± 0.06 |
| Disassemble | 0.42 ± 0.14 | 0.23 ± 0.05 | **0.50** ± 0.06 |
| Coffee-Pull | 0.40 ± 0.05 | **0.58** ± 0.11 | 0.45 ± 0.11 |
| Stick-Pull | **0.32** ± 0.06 | 0.18 ± 0.06 | 0.27 ± 0.09 |
| Peg-Insert-Side | 0.30 ± 0.04 | **0.52** ± 0.08 | 0.25 ± 0.04 |
| Pick-Place-Wall | **0.28** ± 0.12 | 0.13 ± 0.07 | 0.21 ± 0.16 |
| Pick-Out-Of-Hole | 0.26 ± 0.25 | 0.27 ± 0.27 | **0.59** ± 0.08 |
| Pick-Place | 0.20 ± 0.03 | 0.29 ± 0.11 | **0.72** ± 0.09 |
| Handle-Pull-Side | 0.14 ± 0.04 | 0.09 ± 0.05 | **0.94** ± 0.08 |
| Average | **0.73** ± 0.07 | 0.67 ± 0.07 | 0.68 ± 0.07 |
| Cheetah-Vel | **-171.52** ± 21.96 | **-156.07** ± 23.22 | -292.92 ± 36.66 |
| Point-Robot | **-5.10** ± 0.26 | **-4.68** ± 0.18 | -19.60 ± 1.15 |
| Point-Robot-Sparse | 7.78 ± 0.64 | **8.37** ± 0.67 | 0.00 ± 0.00 |

# F  ABLATION STUDY

In this section, we conduct various ablation studies to investigate the robustness of GCC in dataset quality and hyper-parameters.

**Initial stage length.** Table 9 shows GCC's performance with different initial stage lengths. The total number of adaptation episodes is 20. We find that GCC performs well during 10-15 episodes, which is 50%-75% of the total number of adaptation episodes. A small initial stage length (5) may lead to a possibly unreliable task belief and cause a degrade in performance. The 19-episode does not perform the iterative optimization process, and the task belief updates will not converge.

**Number of latent task variables sampled in the initial phase.** $n_{\bar{z}}$ controls the number of diverse samples used to produce the task embedding candidates $z_t^i$. As shown in Table 10, GCC is robust to changes of $n_{\bar{z}}$, and works in a large range from 5 to 20.

**Dataset Quality.** We test GCC and baselines with several "medium" datasets, which are collected by periodically evaluating policies of SAC. As shown in Table 11, GCC still significantly outperforms baseline algorithms on medium datasets, which implies GCC's ability to learn various datasets. We do not test BOReL, as it already fails on the easier expert-level datasets.

Table 9: GCC's performance with various initial stage lengths.

| Environment | 5 Episodes | 10 Episodes | 15 Episodes | 19 Episodes |
|---|---|---|---|---|
| Point-Robot | -6.04 $\pm$ 0.31 | **-5.11** $\pm$ 0.21 | **-5.10** $\pm$ 0.26 | -5.37 $\pm$ 0.11 |
| Point-Robot-Sparse | 4.04 $\pm$ 0.58 | **7.78** $\pm$ 0.64 | **8.07** $\pm$ 0.62 | 7.29 $\pm$ 0.50 |

Table 10: GCC's performance with various $n_{\bar{z}}$s.

| Environment | $n_{\bar{z}} = 1$ | $n_{\bar{z}} = 5$ | $n_{\bar{z}} = 10$ | $n_{\bar{z}} = 20$ |
|---|---|---|---|---|
| Point-Robot | -5.92 $\pm$ 0.31 | **-5.11** $\pm$ 0.21 | **-4.94** $\pm$ 0.16 | **-4.99** $\pm$ 0.23 |
| Point-Robot-Sparse | 5.66 $\pm$ 0.63 | **7.78** $\pm$ 0.64 | **7.31** $\pm$ 0.74 | **7.78** $\pm$ 0.57 |

Table 11: Algorithms' performance on datasets of various qualities.

| Environment | GCC | FOCAL | MACAW |
|---|---|---|---|
| Sweep | **0.77** $\pm$ 0.04 | 0.32 $\pm$ 0.08 | 0.20 $\pm$ 0.20 |
| Sweep-Medium | **0.59** $\pm$ 0.13 | 0.38 $\pm$ 0.13 | 0.04 $\pm$ 0.03 |
| Peg-Insert-Side | **0.30** $\pm$ 0.04 | 0.08 $\pm$ 0.03 | 0.00 $\pm$ 0.00 |
| Peg-Insert-Side-Medium | **0.30** $\pm$ 0.14 | 0.10 $\pm$ 0.07 | 0.00 $\pm$ 0.00 |

