# OpenReview forum: "Correcting Data Distribution Mismatch in Offline Meta-Reinforcement Learning with Few-Shot Online Adaptation"
_ICLR.cc/2023/Conference — Submitted to ICLR 2023_

### Official Review · Reviewer_HYwC · 2022-10-23

**Confidence:** 4
**Correctness:** 3
**Technical Novelty And Significance:** 3
**Empirical Novelty And Significance:** 3
**Recommendation:** 5

**Clarity, Quality, Novelty And Reproducibility:**

### Clarity

1. Could the authors provide a more detailed discussion on the advantages of the proposed method over SMAC and BoRel? Why aren’t these two methods included as the baselines?
2. The main idea of this paper, i.e., purely relying on online adaptation experience to identify the task instead of relying on the expert-level offline data, should be made much more clear early in the paper. I was confused in terms of the difference between this work and FOCAL/MACAW after reading the introduction.
3. Figure 3(a) is confusing. The offline dataset for the point robot environment should consists of multiple goals (meta-training tasks) but all offline data correspond to the same goal. It exaggerates the distribution shift between offline data and online adaptation trajectory.
4. Similarly, I couldn’t get the idea of Figure 1. Please provide an explanation. Why is the meta-testing task out-of-distribution? Isn’t it identical to task 3?
5. The diverse sampling process described by equation (7) assumes that closer latent task embeddings indicate closer policies. Does this mean that the proposed method may only work with *some* off-the-shelf offline meta RL methods that explicitly force this kind of latent task embedding structure, e.g., FOCAL with contrastive loss?

### Quality

This is a solid paper with extensive theoretical analysis and experimental results. However, I find that a few parts of the paper need to be improved in terms of the clarity.

### Novelty

The greedy context-based meta-testing is novel in the offline meta-RL setting.

### Reproducibility

The authors provided the code.

**Strength And Weaknesses:**

### Strength

1. The proposed method is well-justified by the theoretical analysis and the proposed method performs well compared to baseline methods.

### Weaknesses

1. The clarity needs to be improved (see my detailed comments below).
2. There’s no discussion on the limitations of the proposed method. Specifically, the proposed method still assumes that the meta-testing task is sampled from the same task distribution as the meta-training tasks. Thus, the method is dealing with data distribution shift between offline data and online adaptation data, but not distribution shift between meta-training tasks and the meta-testing task (e.g., in cheetal-vel when all meta-training tasks correspond to velocity targets < 1 m/s while the meta-testing target velocity is 2 m/s). I encourage the authors to make it more clear in the paper.

**Summary Of The Paper:**

This paper introduces a greedy context-based meta-testing procedure to tackle the distribution shift issue when performing online adaptation of offline meta-trained policies. The proposed method extends FOCAL with a diverse latent task embedding sampling strategy and a context selection mechanism based on the highest return in the meta-testing phase. The authors justify such a methods with analysis in the BAMDP framework. The experiment results in several standard offline meta RL benchmark datasets demonstrate the superiority of the proposed method over previous offline meta-RL methods in the online adaptation setting.

**Summary Of The Review:**

This is a borderline paper and I’m willing to adjust my score if my concerns outlined in my above comments (mostly regarding clarity) could be addressed.

---

> ### Author Response · Authors · 2022-11-18
> **Response to Reviewer HYwC (Part I)**
>
> Thank the reviewer for the careful reading and constructive comments. We provide clarification to the reviewer's questions as follows. We appreciate any further questions or comments.
>
> **Q1**: I couldn’t get the idea of Figure 1. Please provide an explanation. Why is the meta-testing task out-of-distribution? Isn’t it identical to task 3?
>
> **A1**: The meta-testing task is in the task distribution and is identical to task 3. In Figure 1, the adaptation episode 1 is out-of-distribution because it has no return for the meta-testing task and is different from the given offline multi-task dataset, although its transitions are the same as the dataset of task 2. Note that when utilizing expert task-dependent behavior policies, the offline dataset consists of successful paths of all tasks, i.e., go right in task 1, go mid in task 2, and go left in task 3. The adaptation episode 1 aims to go mid in task 3, which can not be collected in the offline dataset through expert task-dependent behavior policies. We characterize this remarkable gap of reward and transition distribution (conditioned on the same state-action pair) between offline task-dependent data collection and online adaptation, namely ***data distribution mismatch***, as shown in Eq. (4). Moreover, this out-of-distribution adaptation episode 1 will mislead the agent in inferring task belief since it is not encountered during offline meta-training.
>
> **Q2**: Figure 3(a) is confusing. The offline dataset for the point robot environment should consists of multiple goals (meta-training tasks) but all offline data correspond to the same goal. It exaggerates the distribution shift between offline data and online adaptation episodes.
>
> **A2**: Thanks for spotting the typos. In Figure 3(a), we just show the data of one meta-training task (corresponding to the marked goal). We have modified the legends for clarification in the revision. In this didactic example, the transition function shares across meta-RL tasks, and the distribution shift exists in the reward functions between offline data and online adaptation episodes.
>
> **Q3**: There’s no discussion on the limitations of the proposed method. The method is dealing with data distribution shift between offline data and online adaptation data, but not distribution shift between meta-training tasks and the meta-testing task. I encourage the authors to make it more clear in the paper.
>
> **A3**: Thank the reviewer for the suggestion. Task distributional shift is an important problem for meta-RL [1,2], which requires the agent to address out-of-distribution tasks. We assume the distribution consistency between meta-training and meta-testing tasks in the theoretical analysis and empirical evaluation. One limitation of our proposed method is that GCC is a context-based meta-RL algorithm and cannot provide guarantees on out-of-distribution tasks. One sharing insight with Mendonca et al. [1] is that a context-based task inference module cannot handle out-of-distribution inputs. Thus, we utilize a selective context mechanism in the offline meta-RL with task-dependent data collection to filter out out-of-distribution context during task inference. We will add a detailed discussion of this limitation in the revision.

---

> > ### Author Response · Authors · 2022-11-18
> > **Response to Reviewer HYwC (Part II)**
> >
> > **Q4**: Could the authors provide a more detailed discussion on the advantages of the proposed method over SMAC and BOReL? Why aren’t these two methods included as the baselines?
> >
> > **A4**: BOReL and SMAC require additional information or assumptions for fast adaptation. BOReL assumes that reward functions of tasks are known and augment the data for policy learning by relabeling rewards from datasets of other meta-training tasks. When dynamics functions are shared across task distribution, BOReL leverages reward relabeling to break task-dependent data collection, i.e., the meta-behavior policy is the marginalization of task-dependent behavior policies. SMAC assumes that free interactions with the environment without reward supervision are available. SMAC shares similar ideas with BOReL, which learns reward generative models from offline multi-task datasets and relabels the transitions collected online. Compared with BOReL and SMAC, our method GCC focuses on the setting of offline meta-RL without extra information. The original paper of BOReL also demonstrates a variant that does not utilize extra information, and we compare our method with this variant. The table below shows that BOReL fails in most of the tasks, as BOReL without oracle reward functions will also suffer from the distribution mismatch problem, which is consistent with the results in its original paper. We provide more results of BOReL on the Meta-World ML1 benchmark in Table 1 and 7 of the revised paper.
> >
> > | Task \ Algorithms             | GCC                         | BOReL             |
> > |:------------------------:|:---------------------------:|:-----------------:|
> > | Coffee-Push            | **1.26** $\pm$ 0.13    | 0.00 $\pm$ 0.00   |
> > | Faucet-Close           | **1.12** $\pm$ 0.01    | 0.13 $\pm$ 0.03   |
> > | Faucet-Open            | **1.05** $\pm$ 0.02    | 0.12 $\pm$ 0.05   |
> > | Door-Close             | **0.99** $\pm$ 0.00    | 0.37 $\pm$ 0.19   |
> > | Drawer-Close           | **0.99** $\pm$ 0.02    | 0.00 $\pm$ 0.00   |
> > | Door-Lock              | **0.97** $\pm$ 0.01    | 0.14 $\pm$ 0.00   |
> > | Plate-Slide-Back       | **0.96** $\pm$ 0.02    | 0.01 $\pm$ 0.00   |
> > | Dial-Turn                | **0.91** $\pm$ 0.05    | 0.00 $\pm$ 0.00   |
> > | Handle-Press           | **0.88** $\pm$ 0.05    | 0.01 $\pm$ 0.00   |
> > | Hammer                 | **0.84** $\pm$ 0.06    | 0.09 $\pm$ 0.01   |
> > | Button-Press           | **0.74** $\pm$ 0.08    | 0.01 $\pm$ 0.01   |
> > | Push-Wall                | **0.71** $\pm$ 0.15    | 0.00 $\pm$ 0.00   |
> > | Hand-Insert            | **0.63** $\pm$ 0.04    | 0.00 $\pm$ 0.00   |
> > | Peg-Unplug-Side        | **0.56** $\pm$ 0.07    | 0.00 $\pm$ 0.00   |
> > | Bin-Picking            | **0.53** $\pm$ 0.16             | 0.00 $\pm$ 0.00   |
> > | Soccer                   | **0.44** $\pm$ 0.04    | 0.04 $\pm$ 0.02   |
> > | Coffee-Pull              | **0.40** $\pm$ 0.05    | 0.00 $\pm$ 0.00   |
> > | Pick-Place-Wall        | **0.28** $\pm$ 0.12             | 0.00 $\pm$ 0.00   |
> > | Pick-Out-Of-Hole         |  **0.26** $\pm$ 0.25          | 0.16 $\pm$ 0.16             |
> > | Handle-Pull-Side            | **0.14** $\pm$ 0.04 | 0.13 $\pm$ 0.09    |
> > | Average over all 50 Tasks           | **0.73** $\pm$ 0.07 | 0.04 $\pm$ 0.01    |
> >
> > Table: Comparison of GCC and BOReL on a bunch of Meta-World tasks as well as average performance on all 50 tasks.
> >
> > **Q5**: The diverse sampling process described by equation (7) assumes that closer latent task embeddings indicate closer policies. Does this mean that the proposed method may only work with some off-the-shelf offline meta RL methods that explicitly force this kind of latent task embedding structure, e.g., FOCAL with contrastive loss?
> >
> > **A5**: We assume that closer latent task embeddings indicate closer policies, a popular idea in offline multi-task RL [2] and offline meta-RL [3,4] that distinguishes latent task embeddings for different tasks within limited offline datasets. Contrastive loss is one implementation of this idea and our method can work with this branch of offline meta-RL training methods.
> >
> > **Q6**: The main idea of this paper, i.e., purely relying on online adaptation experience to identify the task instead of relying on the expert-level offline data, should be made much more clear early in the paper.
> >
> > **A6**: Thank the reviewer for the suggestion. We update Section 1 for clarity in the revision.

---

> > > ### Author Response · Authors · 2022-11-18
> > > **Response to Reviewer HYwC (Part III)**
> > >
> > > [1] Mendonca R, Geng X, Finn C, et al. Meta-reinforcement learning robust to distributional shift via model identification and experience relabeling[J]. arXiv preprint arXiv:2006.07178, 2020.
> > >
> > > [2] Li J, Vuong Q, Liu S, et al. Multi-task batch reinforcement learning with metric learning[J]. Advances in Neural Information Processing Systems, 2020, 33: 6197-6210.
> > >
> > > [3] Li L, Yang R, Luo D. Focal: Efficient fully-offline meta-reinforcement learning via distance metric learning and behavior regularization[J]. arXiv preprint arXiv:2010.01112, 2020.
> > >
> > > [4] Yuan H, Lu Z. Robust Task Representations for Offline Meta-Reinforcement Learning via Contrastive Learning[C]//International Conference on Machine Learning. PMLR, 2022: 25747-25759.

---

> > > > ### Author Response · Authors · 2022-12-05
> > > > **Sincerely looking forward to further feedback**
> > > >
> > > > Dear Reviewer,
> > > >
> > > > Thank you for your time and efforts in reviewing our work. We have provided detailed clarification and experimental results to address the issues raised in your comments. If our response has addressed your concerns, we would be grateful if you could re-evaluate our work.
> > > >
> > > > If you have any additional questions or comments, we would be happy to have further discussions.
> > > >
> > > > Thanks,
> > > >
> > > > The authors

---

### Official Review · Reviewer_VSon · 2022-10-24

**Confidence:** 2
**Clarity, Quality, Novelty And Reproducibility:** Mentioned above.
**Correctness:** 4
**Technical Novelty And Significance:** 2
**Empirical Novelty And Significance:** 3
**Recommendation:** 5

**Strength And Weaknesses:**

Strength
- Empirical performance is strong compared to the baselines.
- The proposed method is intuitive.

Weakness
- The authors introduces the term "data distribution mismatch" and describes it as a challenge unique to offline meta-RL. However, I am confused with this term in two ways: 1) "data distribution mismatch" seems too broad a term to indicate a problem specific for offline meta-RL. 2) It is unclear why this problem is unique to offline meta-RL. Distribution shift is a problem that persists for almost any learning models. In what aspect is data distribution match different from distribution shift? In other words, why do we need this new term?

**Summary Of The Paper:**

The paper identifies the distribution shift between the offline dataset and online rollouts as the core problem for offline meta-RL and adopts a Bayesian inference procedure to handle it.

**Summary Of The Review:**

The paper proposes a data distribution correction method to efficiently 'guess' a probable task during policy deployment. While the clarity of the paper can be improved, the proposed method is intuitive and shows better empirical performance compared to the baselines.

---

> ### Author Response · Authors · 2022-11-18
> **Response to Reviewer VSon**
>
> Thank the reviewer for the thoughtful comments. We provide clarification to the reviewer's questions below. We appreciate it if the reviewer has any further questions or comments.
>
> **Q1**: It is unclear why this problem is unique to offline meta-RL. In what aspect is data distribution match different from distribution shift? In other words, why do we need this new term?
>
> **A1**: "Data distribution mismatch" in offline meta-RL differs from the "distribution shift" in offline single-task RL. The latter specifies the **"policy"** distribution shift between offline learning policy and the given dataset collected by behavior policies in the literature [1]. In this paper, we characterize the **"transition and reward"** distribution mismatch between offline task-dependent data collection and online adaptation (see Eq. (4)). Note that in the offline single-task RL [1], the reward and transition function between the given dataset and the environment is consistent. Therefore, the "transition and reward" distribution mismatch is unique for offline meta-RL with online adaptation. Offline meta-RL simultaneously faces "transition and reward" distribution mismatch and "policy" distributional shift, which can exacerbate the gap of visitation distribution between an offline multi-task dataset and online adaptation.
>
> **Q2**: "Data distribution mismatch" seems too broad a term to indicate a problem specific for offline meta-RL.
>
> **A2**: In theory, we define the "data distribution mismatch" to characterize the gap of reward and transition distribution between offline data collection and online adaptation, which is a rigorous definition (see Eq. (4) in Section 3).
>
> [1] Levine S, Kumar A, Tucker G, et al. Offline reinforcement learning: Tutorial, review, and perspectives on open problems[J]. arXiv preprint arXiv:2005.01643, 2020.

---

> > ### Author Response · Authors · 2022-12-05
> > **Sincerely looking forward to further feedback**
> >
> > Dear Reviewer,
> >
> > Thank you for your time and efforts in reviewing our work. We have provided detailed clarification to address the issues raised in your comments. If our response has addressed your concerns, we would be grateful if you could re-evaluate our work.
> >
> > If you have any additional questions or comments, we would be happy to have further discussions.
> >
> > Thanks,
> >
> > The authors

---

### Official Review · Reviewer_7zMW · 2022-10-27

**Confidence:** 4
**Correctness:** 3
**Technical Novelty And Significance:** 3
**Empirical Novelty And Significance:** 3
**Recommendation:** 6

**Clarity, Quality, Novelty And Reproducibility:**

Generally, the clarity of this work is quite high, except in the theoretical sections highlighted above.
The quality and novelty is good: the idea to filter out trajectories is new and quite interesting, though it would benefit from discussion on when such a method would not work, as discussed above.

**Strength And Weaknesses:**

## Strengths

There is much to like about this work. This work clearly and soundly characterizes the problem that it is trying to solve (with some exceptions, described below), and overall, the main idea of filtering out trajectories that the policy doesn't know how to update on from online contexts is a novel and interesting approach to offline meta-RL. Additionally, this approach appears to yield fairly impressive empirical improvements that are relevant to the meta-RL community.

## Weaknesses

The theory could be strengthened in a few areas:
- Specifically, the notion of filtering introduced in Fact 1 is insufficiently precise and requires some work to formalize. One precise notion of filtering is to filter based on visiting belief states under the transformed BAMDP that are also visited by the behavior policies (i.e., are present in the offline data). However, this still requires some work to formalize: how do you deal with multi-step episodes, where the first few timesteps include belief states that are out-of-distribution from the offline data, but later transition into in-distribution beliefs? Intuitively, inclusion of such episodes seems beneficial for the post-adaptation policy, but it requires more precise filtering definitions.
- It should also be noted that the existing notion of filtering based on visiting transformed belief states is generally infeasible. For example, in the case where different behavior policies take distinct actions at the initial state, but the observation after taking these actions does not reveal information about the identity of the current MDP, the transformed belief update will erroneously place all mass on the MDP corresponding to the observed behavior policy, which cannot be corrected without actually knowing the identity of the MDP. At the very least, it would be worthwhile to discuss under what cases such filtering is actually possible.

This work would also benefit from discussion about its own limitations and acknowledgements of which aspects are heuristics or approximations. Specifically, the idea to practically filter in GCC based on the trajectories earning the highest reward is heuristic that does not work in general. In many tasks requiring "challenging" exploration, such as those considered in HyperX [1] or DREAM [2], highly informative exploration trajectories do not necessarily achieve high returns, and would be filtered away in GCC. Other times, it may be impossible or difficult to obtain any high reward trajectories, even when these trajectories are highly informative, which would also cause failure. This work would be strengthened from discussion on this aspect. Additionally, it may be interesting to discuss relationships with the original theoretical motivation of matching behavior policy trajectories or more precisely, testing for in-distribution-ness with the behavior policies.

[1] Exploration in Approximate Hyper-State Space for Meta Reinforcement Learning. Zintgraf et al., '21.

[2] Decoupling Exploration and Exploitation in Meta-Reinforcement Learning without Sacrifices. Liu et al' 21.

## Additional Comments
- There is an error in Proposition 1, where the policy evaluation gap is in fact $\frac{H^+ - 1}{2}$, and not $\frac{H^+}{2}$.
- It's unclear why the example in Figure 2 includes $2v$ actions, where the last $v$ actions yield no reward -- this appears to have no bearing whatsoever to the provided theory, and could probably be omitted for clarity.

**Summary Of The Paper:**

This work considers the problem in offline meta-reinforcement learning, where there is a distribution shift between the contexts seen in the offline data, and the contexts obtained by online roll-outs. This work first offers some formal characterizations of this shift, and then uses these characterizations to motivate a new offline meta-RL algorithm that works by filtering out episodes that appear out-of-distribution in order to obtain contexts that still look in-distribution for online adaptation.

**Summary Of The Review:**

Overall, the ideas in this work are quite interesting and are useful to the community. However, as there are some significant weaknesses to address, I will begin by only weakly recommending acceptance, though I am willing to increase my score based on the authors' response.

---

> ### Author Response · Authors · 2022-11-18
> **Response to Reviewer 7zMW (Part I)**
>
> Thanks for your careful reading and insightful comments. We provide clarification to the reviewer's questions below. We appreciate any further questions or comments.
>
> **Q1**: The notion of filtering introduced in Fact 1 is insufficiently precise and requires some work to formalize. One precise notion of filtering is to filter based on visiting belief states under the transformed BAMDP that are also visited by the behavior policies (i.e., are present in the offline data).
>
> **A1**: Thank the reviewer for this constructive suggestion. To formalize the original Fact 1 (i.e., *transformed BAMDPs implicitly require the agent to filter out out-of-distribution episodes*), we add two facts in the revision, as suggested by the reviewer: new Fact 1 in Section 3.1 and Fact 2 in Section 3.2. In new Fact 1, we prove that there exists a BAMDP that, for any batch-constrained meta-policy, the agent will visit out-of-distribution belief states during meta-testing due to data distribution mismatch. To avoid visiting out-of-distribution belief states, in Fact 2, we prove that for feasible Bayesian belief updating, transformed BAMDPs confine the agent in the in-distribution belief states during meta-testing. Note that new Fact 1 can directly lead to Proposition 1 since RL agents will visit out-of-distribution belief states during meta-testing in BAMDPs and induce unreliable offline policy evaluation. See Section 3 for details.
>
> **Q2**: How do you deal with multi-step episodes, where the first few timesteps include belief states that are out-of-distribution from the offline data, but later transition into in-distribution beliefs? Intuitively, inclusion of such episodes seems beneficial for the post-adaptation policy.
>
> **A2**: In our theoretical setting, such episodes are not available in transformed BAMDPs since the Bayesian belief updating is infeasible on the first few out-of-distribution belief states. We will argue that the probability of such episodes occurring is relatively low because the offline training paradigm cannot provide guarantees for meta-policies on out-of-distribution states [2,3]. In practice, our empirical method GCC can still benefit from these episodes when they have higher returns.
>
> **Q3**: The existing notion of filtering based on visiting transformed belief states is generally infeasible. It would be worthwhile to discuss under what cases such filtering is actually possible.
>
> **A3**: We thank the reviewer for the advice. We add a section (i.e., Appendix A.5) to discuss how to filter out out-of-distribution episodes in transformed BAMDPs during meta-testing. We prove that meta-policies with Thompson sampling [4] can filter out out-of-distribution episodes in transformed BAMDPs with high probability as the offline multi-task dataset grows. Theorem 3 in Appendix A.5 proves that for each adaptation episode in a meta-testing task, there exists a task hypothesis from the current belief, executing a batch-constrained meta-policy with Thompson sampling under this task hypothesis and belief will confine the agent in the in-distribution belief states with high probability. Thus, we can sample several task hypotheses from belief and interact with the environment until finding an in-distribution episode, which is also the basic idea of our context-based algorithm GCC.
>
> **Q4**: For example, in the case where different behavior policies take distinct actions at the initial state, but the observation after taking these actions does not reveal information about the identity of the current MDP, the transformed belief update will erroneously place all mass on the MDP corresponding to the observed behavior policy, which cannot be corrected without actually knowing the identity of the MDP.
>
> **A4**: Our method uses meta-policies with Thompson sampling [4], and can iteratively sample the accurate or nearly accurate task hypothesis to correct the data distribution mismatch in this example. Theorem 3 in Appendix A.5 shows that for each meta-testing task $\kappa_{test}$ in an arbitrary task distribution, the distance between the closest offline meta-training task $\kappa_{i^*}$ (i.e., nearly accurate task hypothesis) and $\kappa_{test}$ will asymptotically approach zero with high probability, as the offline multi-task dataset grows. Therefore, executing a batch-constrained meta-policy with Thompson sampling under the nearly accurate task hypothesis $\kappa_{i^*}$ will confine the agent in the in-distribution belief states with high probability. See Appendix A.5 for details.

---

> > ### Author Response · Authors · 2022-11-18
> > **Response to Reviewer 7zMW (Part II)**
> >
> > **Q5**: In many tasks requiring "challenging" exploration, such as those considered in HyperX [5] or DREAM [6], highly informative exploration episodes do not necessarily achieve high returns, and would be filtered away in GCC. Other times, it may be impossible or difficult to obtain any high reward episodes, even when these episodes are highly informative, which would also cause failure. This work would be strengthened from discussion on this aspect.
> >
> > **A5**: Exploration in BAMDPs is an important problem for meta-RL. Offline meta-RL utilizes a high-quality multi-task dataset to deal with challenging exploration in BAMDPs during meta-training. Compared with fantastic exploration methods in HyperX and DREAM, GCC utilizes a simple exploration method for online adaptation, i.e., Thompson sampling [4], in which we can sample the right "task hypothesis" to obtain high-reward episodes using more adaptation episodes. Since the belief states in the offline dataset may be narrow, how to study a more efficient exploration method in offline meta-RL is an interesting future direction. We will add a detailed discussion in the revision.
> >
> > **Q6**: This work would also benefit from discussion about its own limitations and acknowledgements of which aspects are heuristics or approximations. Specifically, the idea to practically filter in GCC based on the episodes earning the highest reward is heuristic that does not work in general.
> >
> > **A6**: The limitation of this filtering method is that GCC may require more online adaptation episodes to infer accurate task identification. The assumption of filtering out out-of-distribution episodes in GCC is that few-shot out-of-distribution (OOD) episodes generated by an offline-learned meta-policy usually have lower returns. This assumption is sensible because, with the offline training paradigm, its policy is optimized on in-distribution states in a conservative manner but is not well-optimized on OOD states, which tend to have lower returns [2,3]. The contrapositive statement of its assumption is that episodes with higher returns have a higher probability of being in the meta-training distribution of offline datasets. Following this heuristic, GCC may neglect some task information in adaptation episodes with lower returns for conservatism (i.e., for avoiding out-of-distribution episodes with lower confidence). We will add a detailed discussion of this limitation in the revision.
> >
> >
> > **Q7**: It may be interesting to discuss relationships with the original theoretical motivation of matching behavior policy episodes or more precisely, testing for in-distribution-ness with the behavior policies.
> >
> > **A7**: The relationship between our original theoretical motivation and the empirical algorithm is that GCC utilizes an approximate out-of-distribution quantification (discussed in A6 above). We have also tried a popular uncertainty estimation of offline RL by using an ensemble (i.e., MOReL [7]), to distinguish whether an adaptation episode is in the distribution of the given meta-training dataset. MOReL utilizes the maximum distance between ensemble model predictions to quantify the uncertainty of data. However, as shown in Figure 5(b) of Appendix C, this uncertainty estimation method cannot accurately estimate the distance to the offline dataset, resulting in the inability to identify in-distribution data. This phenomenon is consistent with the literature of offline single-task RL [8], in which accurate out-of-distribution quantification is challenging in practice. On the other hand, GCC can successfully identify in-distribution data with the greedy selective context mechanism, as shown in Figure 5(a) of Appendix C. We also evaluate the uncertainty estimation method on several Meta-World ML1 tasks, and as shown in the table below, the uncertainty estimation method achieves similar performance to FOCAL, and underperforms GCC. See details in Appendix E.1.
> >
> > | Task \ Algorithms             | GCC                         | Uncertainty Estimation  Method             |  FOCAL |
> > |:------------------------:|:---------------------------:|:-----------------:|:-----------------:|
> > | Plate-Slide-Back           | **0.96** $\pm$ 0.02    |   0.64 $\pm$  0.21  |  0.58  $\pm$   0.06 |
> > | Hammer            | **0.84** $\pm$ 0.06    |  0.61 $\pm$ 0.33    | 0.59  $\pm$   0.07 |
> > | Coffee-Push            | **1.26** $\pm$ 0.13    |  0.02 $\pm$  0.01   | 0.66  $\pm$   0.07
> > | Push-Wall             | **0.71** $\pm$ 0.15    | 0.48 $\pm$ 0.19   | 0.43   $\pm$  0.06 |
> > | Point-Robot             | **-5.10** $\pm$ 0.26    | -19.62   $\pm$  0.77   | -15.38   $\pm$  0.95 |
> >
> > Table: Comparison of GCC and uncertainty estimation method on several tasks.

---

> > > ### Author Response · Authors · 2022-11-18
> > > **Response to Reviewer 7zMW (Part III)**
> > >
> > > **Q8**: There is an error in Proposition 1, where the policy evaluation gap is in fact $\frac{H^+-1}{2}$, and not $\frac{H^+}{2}$.
> > >
> > > **A8**: Thanks for spotting the typos. We fix it in the revision of Proposition 1 and its proof.
> > >
> > > **Q9**: It's unclear why the example in Figure 2 includes $2v$ actions, which could probably be omitted for clarity.
> > >
> > > **A9**: Actions $\{a_{v+1},\dots,a_{2v}\}$ aim to emphasize the common structure of task distribution, i.e., the reward function for these actions is 0. For clarity, we omit these actions in the revision.
> > >
> > > [1] Fujimoto S, Meger D, Precup D. Off-policy deep reinforcement learning without exploration[C]//International conference on machine learning. PMLR, 2019: 2052-2062.
> > >
> > > [2] Fujimoto, Scott, David Meger, and Doina Precup. "Off-policy deep reinforcement learning without exploration." International conference on machine learning. PMLR, 2019.
> > >
> > > [3] Mendonca R, Geng X, Finn C, et al. Meta-reinforcement learning robust to distributional shift via model identification and experience relabeling[J]. arXiv preprint arXiv:2006.07178, 2020.
> > >
> > > [4] Strens M. A Bayesian framework for reinforcement learning[C]//ICML. 2000, 2000: 943-950.
> > >
> > > [5] Exploration in Approximate Hyper-State Space for Meta Reinforcement Learning. Zintgraf et al., '21.
> > >
> > > [6] Decoupling Exploration and Exploitation in Meta-Reinforcement Learning without Sacrifices. Liu et al' 21.
> > >
> > > [7] Kidambi, Rahul, et al. "Morel: Model-based offline reinforcement learning." Advances in neural information processing systems 33 (2020): 21810-21823.
> > >
> > > [8] Yu, Tianhe, et al. "Combo: Conservative offline model-based policy optimization." Advances in neural information processing systems 34 (2021): 28954-28967.
> > >
> > > EDIT: We updated this part of response to fix typos.

---

> > > > ### Author Response · Authors · 2022-12-05
> > > > **Sincerely looking forward to further feedback**
> > > >
> > > > Dear Reviewer,
> > > >
> > > > Thank you for your time and efforts in reviewing our work. We have provided detailed clarification with additional theorems and experimental results to address the issues raised in your comments. If our response has addressed your concerns, we would be grateful if you could re-evaluate our work.
> > > >
> > > > If you have any additional questions or comments, we would be happy to have further discussions.
> > > >
> > > > Thanks,
> > > >
> > > > The authors

---

> > > > > ### Comment · Reviewer_7zMW · 2022-12-06
> > > > > **Reviewer Response**
> > > > >
> > > > > I sincerely appreciate the thought and effort placed into the author response, which I found to be overall quite encouraging.
> > > > >
> > > > > My overall main concerns were as follows:
> > > > >
> > > > > (1) The heuristic of filtering episodes based on returns is not general. This remains a major concern, though I appreciate the author response. However, I also recognize that part of the main value of this work is the new theory which formalizes that filtering could be a valuable way to deal with distribution shift. Therefore, I think this work can still be useful, even if the implementation of some of the theoretical ideas is not yet perfect.
> > > > >
> > > > > (2) Discussion of weaknesses of the proposed approach and theory. I found the authors revisions regarding these aspects to be quite useful. In particular, discussing when filtering based on the transformed belief states is possible in the new Appendix is helpful, although I would further encourage the authors to also detail the cases where such filtering is not feasible. I would further encourage the authors to also discuss the limitations of the episodic-returns based heuristic above. It is fine that the proposed algorithm is imperfect, but when and where it should be applied should be highlighted.
> > > > >
> > > > > (3) Clarity. Many reviewers, including myself raised concerns about several places of clarity. The author response sufficiently addressed my clarity concerns about the theoretical sections.
> > > > >
> > > > > Overall, I still recommend acceptance of this work. It is clear that major weaknesses still exist: i.e., the proposed algorithm does not handle cases where the exploration behavior recovers a lot of information, but does not necessarily achieve high returns, which often occurs in cases (e.g., experiments in HyperX or DREAM). However, the work also clearly provides valuable insights to the meta-RL community: namely, that filtering can be used to deal with distribution shift in meta-RL. The weaknesses can be alleviated by discussing when the proposed algorithm can work (and when it won't!)

---

> > > > > > ### Author Response · Authors · 2022-12-09
> > > > > > **Further clarification on the questions (Part I)**
> > > > > >
> > > > > > Thanks for your careful reading, insightful feedback, and finding that one of the main contributions in the paper is our theoretical analysis, which formally studies the distributional shift of reward and transition distribution in offline meta-RL. We will provide clarification and experimental results to your questions and concerns as below. If you have any additional questions or comments, please post them and we would be happy to have further discussions.
> > > > > >
> > > > > > **Q1**: It is clear that major weaknesses still exist: i.e., the proposed algorithm does not handle cases where the exploration behavior recovers a lot of information, but does not necessarily achieve high returns, which often occurs in cases (e.g., experiments in HyperX or DREAM).
> > > > > >
> > > > > > **A1**: Our proposed method GCC can solve such tasks (e.g., Sparse HalfCheetahDir and Sparse PointRobot proposed in HyperX) using more adaptation episodes. Different from the meta-training exploration of online meta-RL, the offline dataset collected by task-dependent behavior policies usually contains ***effective exploitation trajectories*** for each meta-training task. Off-the-shelf context-based meta-training algorithms (e.g., FOCAL) will meta-train exploitation policies for each generalizable task embedding (i.e., task hypothesis). These exploitation policies can achieve **high return** and contain sufficient information in the corresponding tasks. During online adaptation, GCC can sample a bunch of task hypotheses and adapt the high-return contexts (potentially generated by exploitation policies with accurate or nearly correct task hypotheses) to meta-testing tasks. GCC will iteratively update the task belief with various episodes, in which a higher-return episode in one iteration may become a lower-return one in later iterations, as shown in Appendix D. Moreover, as the reviewer may realize, one limitation of our method may discard some exploratory trajectories with low returns and require more adaptation episodes.
> > > > > >
> > > > > > We evaluate GCC and FOCAL in Sparse-Cheetah-Dir and Sparse-Point-Robot. For both tasks, we pre-train task-dependent SAC with default entropy-based exploration on each meta-training task to collect offline multi-task datasets.
> > > > > >
> > > > > > | Algorithm \ The number of Adaptation Episodes             | 2                         | 5             |
> > > > > > |:------------------------:|:---------------------------:|:-----------------:|
> > > > > > | GCC            | 115 $\pm$ 73    |  841 $\pm$ 25    |
> > > > > > | FOCAL            | -51.6 $\pm$ 11.7    |  -44.4 $\pm$ 15.2    |
> > > > > >
> > > > > > Table x1: Comparison of GCC and FOCAL on Sparse-Cheetah-Dir. During task-dependent offline data collection, the average performance of pre-trained SAC on each task is 856 $\pm$ 42.
> > > > > >
> > > > > >
> > > > > > | GCC           |FOCAL           | SAC's Optimal Return                         |
> > > > > > |:------------------------:|:---------------------------:|:---------------------------:|
> > > > > > | 7.78 $\pm$ 0.64       | 0.83 $\pm$ 0.37      | 8.27 $\pm$ 0.29   |
> > > > > >
> > > > > > Table x2: Comparison of GCC, FOCAL, and the optimal return achieved by pre-trained SAC on each task individually on Sparse-Point-Robot.
> > > > > >
> > > > > > In Sparse-Cheetah-Dir, the average performance of pre-trained SAC (i.e., containing effective exploitation trajectories) is 856. Results in Table x1 show that GCC with 5 adaptation episodes can infer the accurate task hypothesis and solve this task, i.e., achieving comparable performance with pre-trained task-dependent behavior policies. Comparing GCC with FOCAL, the results show that our filtering method can significantly improve performance using an in-distribution context. The performance on different adaptation episodes indicates that GCC requires more adaptation episodes to find a good task hypothesis. HyperX with online training achieves better performance with the RND exploration technique than pre-trained SAC. In this setting, tasks are uniform over “walk forward” and “walk backward.” The exploration behavior in HyperX can try one direction, receive a low return, and infer the right goal in the other direction. GCC will sample a task hypothesis and try its exploitation policy learned from the offline dataset. When GCC selects a wrong task hypothesis, the return will be low, and GCC continues to sample the subsequent task hypotheses to find the correct task hypothesis.

---

> > > > > > > ### Author Response · Authors · 2022-12-09
> > > > > > > **Further clarification on the questions (Part II)**
> > > > > > >
> > > > > > > In Sparse-Point-Robot, the goals are uniform over the semi-circle, and the agent aims to navigate to the goal. Exploration behavior can explore the goals along the semi-circle in each adaptation episode, which is informative and achieves low return. Our method GCC will sample a bunch of task hypotheses and try their exploitation policy to find informative exploitation trajectories with higher returns (i.e., with nearly accurate task hypotheses). Due to the radius of the goal, GCC can address Sparse PointRobot within 20 adaptation episodes. As shown in Table x2, GCC achieves similar performance to the optimal returns achieved by pre-trained SAC on each task individually. In addition, as shown in Table 1, 2, and 6-8 in the paper, GCC also achieves state-of-the-art performance in the Meta-World ML1 benchmark with 50 tasks, which demonstrates the generality of our greedy filtering method.
> > > > > > >
> > > > > > > Moreover, during task-dependent data collection, the exploration behavior may not be in the distribution of offline datasets, e.g., in Sparse PointRobot, exploring the goals along the semi-circle is challenging to collect using a pre-trained SAC learned from a single-task. It is an interesting and exciting future direction to differentiate in-distribution exploration behavior with lower return during meta-testing, which can utilize fewer adaptation episodes for solving meta-testing tasks. We will add a detailed discussion in the revision.
> > > > > > >
> > > > > > > **Q2**: I would further encourage the authors to also detail the cases where such filtering is not feasible.
> > > > > > >
> > > > > > > **A2**: We thank the reviewer for the advice. When the number of adaptation episodes is very limited (e.g., only 2) our filtering method may not find an in-distribution episode to update the belief and thus may not work well. For the example in Figure 2 of Section 3.1, when there are many meta-RL tasks (i.e., $v\ge 10$) and requires to adapt to the meta-testing task within 2 adaptation episodes, it is challenging to find an in-distribution episode (i.e., with the return 1) in 2 online trials. We will add a detailed discussion in the revision. Moreover, we updated the Part III of the previous response to fix typos and are really grateful to the reviewer for the inspired suggestions on our theory and proposed algorithm.

---

### Official Review · Reviewer_NxHR · 2022-11-03

**Confidence:** 4
**Correctness:** 3
**Technical Novelty And Significance:** 2
**Empirical Novelty And Significance:** 3
**Recommendation:** 5

**Clarity, Quality, Novelty And Reproducibility:**

Clarity - Parts of the paper are clearly written; however Sections 2 and 4 are difficult to parse. I remain unsure if I understand the proposed method correctly.
Quality - theoretical and experimental results are thorough
Novelty - The proposed way to address the offline to online distribution shift in offline meta-RL is novel, though the components exist in prior work. However, unless I am understanding the method wrong, it is very limited in that it cannot acquire information from low-return trajectories in order to adapt to a new task.
Reproducibility - It is not clear if code will be released publicly. The appendix contains a table of hyper parameters which will aid reproduction but will likely not be sufficient given the complexity of meta-RL implementations.


**Strength And Weaknesses:**

Questions / Comments
1) If I understand correctly, the proposed method is to use the returns of few-shot trajectories collected at test time as a proxy for measuring whether the collected data is in distribution. Then adaptation is performed using only the highest-return trajectories as context. In Section 4.2, the motivation for this proxy is stated in one sentence, referencing prior works Fujimoto et al. 2019 and Kostrikov et al. 2021. However, this proxy seems like quite a blunt instrument, conflating areas of the state-action space with low return in a particular task with areas outside the support of the offline data distribution. Consider the didactic example in Section 5.1 - the agent cannot make use of trajectories that don’t reach the goal to update its task belief, because those trajectories are deemed OOD because they have low return? I think I am mis-understanding the method, could the authors please clarify? If I am understanding correctly, then the method is quite limited.

2) It would strengthen the paper to show a baseline that tries to detect OOD data in a more standard fashion, e.g., estimating uncertainty via an ensemble.

3) In the point robot experiment in Figure 3, if the training data is all clustered around a trajectory that goes straight (light blue bubbles), how is it that the exploratory trajectories sampled online navigate coherently outside of this distribution (green dots)? As an aesthetic nitpick, 3a is quite small and hard to parse.
Nitpick in wording: On page 1, “BORel (Dorfman et al., 2021) and SMAC (Pong et al., 2022) employ few-shot online adaptation, but the former assumes known reward functions, and the latter requires free interactions with the environment without reward supervision.” It seems that the advantage of the proposed method over SMAC is that thousands of samples (albeit unsupervised) from the test environment are not required. This sentence doesn’t really say that, and makes it sound as if collecting data without reward supervision is onerous, which is not the case.

4) In Section 6, how is the distribution shift discussed in this paper different from the distribution shift discussed in Pong et al. and Dorfman et al? I’m not as familiar with Dorfman et al, but in the case of Pong et al. it seems that this paper is tackling exactly the same issue as raised in that paper. It’s not a problem to tackle the same problem as other papers, and can lead to confusion not state so clearly.


**Summary Of The Paper:**

This paper tackles offline meta-RL where new tasks are learned via online few-shot adaptation. They observe that there is a distribution shift between the offline data used for meta-training, and the data collected online by the meta-learned policy to use for adaptation. This distribution shift can result in poor adaptation performance. To remedy this, the paper proposes to adapt the policy using only the trajectories with the highest returns - using return as a proxy for out-of-distribution detection. The method is evaluated on the MetaWorld Benchmark where it outperforms prior approaches.

Contributions
an offline meta-RL algorithm that can adapt online to new tasks within a few trajectories, without knowledge of the reward function or offline data for the task.
Define the distribution shift between behavior policy and learned policy and show that OOD context data leads to poor adaptation
Empirical evaluation demonstrating that the proposed algorithm outperforms prior approaches on a standard meta-RL benchmark

**Summary Of The Review:**

The paper tackles a relevant and important problem in the field of meta-RL -- addressing the distribution shift present when meta-training offline and learning new tasks online. The experimental results show significant improvements over prior work. However, my current understanding of the method is that it is not able to distinguish between OOD trajectories and trajectories that achieve low return in a given test task, and thus can only adapt from high-return trajectories, severely limiting the applicability of the method. Therefore I don't think it currently meets the bar for a contribution for this conference.

--------- Update 12/6/22 ----------

Thank you to the authors for your detailed response, which has clarified a misunderstanding I had about the method. There is an assumption that the offline meta-training dataset is collected by expert policies, and thus only contains high-return trajectories. In my first review, I missed this assumption, which is stated in Section 2.2. I would recommend that the authors make this assumption more explicit in other parts of the paper as well. For example, the introduction refers to “task-dependent” policies, but it is not clear that this implies expert policies.

With this clarification, I understand that part of the distribution shift being addressed is between high-return trajectories during meta-training and potentially low-return trajectories collected during meta-testing. Figure 3 is now clear, thank you!

However, the proposed approach of handling this distribution shift by filtering out low-return trajectories has great limitations that in my view diminish the contribution significantly. The method cannot make use of low-return exploration trajectories to update the task belief, so exploration during meta-testing is constrained to sampling the unadapted meta-learned policy. Thus the method effectively does not address the problem of exploration in meta-RL, which is a key challenge. While the experimental results do show improvement over prior methods (the prior methods are not designed for online adaptation and so are severely handicapped, but I am not aware of an existing work that is), I think the methodological limitations are severe enough to outweigh the empirical contribution.

I think clarity of presentation can be further improved; specifically, to make the problem statement very clear. I also find the notation in the theory presentation to be difficult to follow. Limitations of the method should be clearly discussed in the main body of the paper.

Overall, I think that the methodological limitations outweigh the contributions, and that the paper would make a much stronger contribution if this limitation is addressed. Therefore, I maintain my score.

---

> ### Author Response · Authors · 2022-11-18
> **Response to Reviewer NxHR (Part I)**
>
> We thank the reviewer for the constructive comments. We provide detailed clarification to the reviewer's questions as follows. We appreciate any further questions or comments.
>
> **Q1**: Adaptation is performed using only the highest-return trajectories as context.
>
> **A1**: The reviewer may misunderstand our method. GCC will iteratively update the task belief using the context of various episodes. As shown in Algorithm 1, the task belief update in GCC has two phases: (i) in an initial stage, it samples "task hypotheses" from prior task distribution and chooses the highest-return episode to derive the initial posterior task belief, and (ii) in the following iterative optimization process, GCC will further incorporate episodes with higher returns into the context and improve the meta-policy iteratively. Figure 3(a) and Figure 5(a) show the initial stage of online adaptation, and Appendix D illustrates the iterative optimization process. As shown in Figure 6(a), GCC iteratively improves the meta-policy and results in a nearly optimal policy.
>
>
> **Q2**: In Section 4.2, the motivation for this proxy is stated in one sentence, referencing prior works. However, this proxy seems like quite a blunt instrument, conflating areas of the state-action space with low return in a particular task with areas outside the support of the offline data distribution.
>
> **A2**: We respectfully disagree that the proxy of out-of-distribution quantification is not a blunt instrument. Instead, it is logically derived. The only assumption of filtering out out-of-distribution episodes in GCC is that few-shot out-of-distribution (OOD) episodes generated by an offline-learned meta-policy usually have lower returns. This assumption is sensible because, with the offline training paradigm, its policy is optimized on in-distribution states in a conservative manner but is not well-optimized on OOD states, which tend to have lower returns [1,2]. The **contrapositive statement** of its assumption is that episodes with higher returns have a higher probability of being in the meta-training distribution of offline datasets. In offline meta-RL with few-shot online adaptation, we adopt this assumption since we choose to trust the offline dataset due to the remarkable gap of reward and transition distribution between offline datasets and online adaptation (see detailed discussion in Section 1). We add a discussion in the revision of Section 4.2.
>
> The limitation of this filtering method is that GCC may require more online adaptation episodes to infer accurate task identification. Following the above heuristic, GCC may neglect some task information in adaptation episodes with lower returns in order for conservatism. We will add a detailed discussion of this limitation in the revision.
>
> **Q3**: In the point robot experiment in Figure 3, if the training data is all clustered around an episode that goes straight (light blue bubbles), how is it that the exploratory episodes sampled online navigate coherently outside of this distribution (green dots)? As an aesthetic nitpick, 3a is quite small and hard to parse.
>
> **A3**: Thank the reviewer for spotting the typo and for advice. We have updated Figure 3 in the revision for a clearer presentation. The light blue bubbles in Figure 3 only illustrate the dataset of one meta-training task (corresponding to the marked goal). For meta-learning, as the tasks are uniformly sampled from the purple semi-circle, the multi-task dataset well covers the half-round. Therefore, for the first ten episodes in the initial stage, the agent explores by sampling task hypotheses from the prior task belief and executing corresponding policies so that the exploratory episodes will reach the goals of other tasks.

---

> > ### Author Response · Authors · 2022-11-18
> > **Response to Reviewer NxHR (Part II)**
> >
> > **Q4**: Consider the didactic example in Section 5.1 - the agent cannot make use of episodes that don’t reach the goal to update its task belief, because those episodes are deemed OOD because they have low return? I think I am mis-understanding the method, could the authors please clarify? If I am understanding correctly, then the method is quite limited.
> >
> > **A4**: The reviewer may misunderstand our method. These episodes are not **deemed** OOD, but are OOD **in fact** because they are distinct from the offline multi-task dataset, e.g., a sub-dataset for one meta-training task is visualized in Figure 3(a) with light blue bubbles. Even if the transitions of these episodes may occur in the datasets collected by other tasks, the rewards of these episodes are still OOD for the given task because its goals are different from other tasks. As shown in Figure 3(c) and Appendix D, these OOD data prevent the agent from performing accurate task inference and perform poorly during meta-testing.
> >
> > To filter out these OOD data, we first tried uncertainty estimation methods (discussed in A5 below) but it does not work well, as they cannot give accurate estimations of uncertainty. As discussed in A2 above, we take the assumption that episodes with higher returns have a higher probability of being in the meta-training distribution of offline datasets. Our method GCC achieves state-of-the-art performance on a variety of tasks (shown in Section 5 and Appendix E). It is worth noting that the high-and-low comparison of true returns in an adaptation episode is relative. GCC will update the task belief in an iterative way with various episodes, in which a higher-return episode in one iteration may become a lower-return one in later iterations, as shown in Appendix D.
> >
> > **Q5**: It would strengthen the paper to show a baseline that tries to detect OOD data in a more standard fashion, e.g., estimating uncertainty via an ensemble.
> >
> > **A5**: We appreciate the reviewer for this suggestion. We adopt a popular uncertainty estimation approach, which uses the maximum discrepancy between ensemble model predictions to quantify the uncertainty of data [3]. As shown in Figure 3(b) and (c) in Section 5.1, this method fails to identify in-distribution episodes and cannot achieve satisfying performance. We provide detailed analyses of this method in Appendix C and D. Visualization results demonstrate that this method cannot estimate well uncertainty of out-of-distribution episodes relative to the offline dataset, erroneously has low uncertainty for out-of-distribution episodes, and result in the inability to identify in-distribution data. This phenomenon is consistent with the literature on offline single-task RL [4], in which accurate out-of-distribution quantification is challenging in practice. On the other hand, GCC can successfully identify in-distribution data with the greedy selective context mechanism, as shown in Figure 5(a) of Appendix C. We also evaluate the uncertainty estimation method on several Meta-World ML1 tasks, and as shown in the table below, the uncertainty estimation method achieves similar performance to FOCAL, and underperforms GCC. See details in Appendix E.1.
> >
> > | Task \ Algorithms             |                      GCC                         |          Uncertainty Estimation  Method             |     FOCAL      |
> > |:------------------------:|:---------------------------:|:-----------------:|:-----------------:|
> > | Plate-Slide-Back           | **0.96** $\pm$ 0.02    |   0.64 $\pm$  0.21  |  0.58  $\pm$   0.06 |
> > | Hammer            | **0.84** $\pm$ 0.06    |  0.61 $\pm$ 0.33    | 0.59  $\pm$   0.07 |
> > | Coffee-Push            | **1.26** $\pm$ 0.13    |  0.02 $\pm$  0.01   | 0.66  $\pm$   0.07
> > | Push-Wall             | **0.71** $\pm$ 0.15    | 0.48 $\pm$ 0.19   | 0.43   $\pm$  0.06 |
> > | Point-Robot             | **-5.10** $\pm$ 0.26    | -19.62   $\pm$  0.77   | -15.38   $\pm$  0.95 |
> >
> > Table: Comparison of GCC and uncertainty estimation method on several tasks.
> >
> >
> > **Q6**: Nitpick in wording: On page 1, “SMAC employs few-shot online adaptation, but requires free interactions with the environment without reward supervision.” This sentence doesn’t really say that, and makes it sound as if collecting data without reward supervision is onerous, which is not the case.
> >
> > **A6**: SMAC is a brilliant hybrid offline meta-RL algorithm focusing on a different setting, i.e., offline meta-training with thousands of unsupervised online samples (without reward labels). The advantage of our method is that these unsupervised online samples are not required. We update Section 1 in the revision.

---

> > > ### Author Response · Authors · 2022-11-18
> > > **Response to Reviewer NxHR (Part III)**
> > >
> > > **Q7**: In Section 6, how is the distribution shift discussed in this paper different from the distribution shift discussed in SMAC?  It seems that this paper is tackling exactly the same issue as raised in that paper.
> > >
> > > **A7**: This paper and SMAC study different distribution shifts in offline meta-RL with online adaptation. SMAC finds that there exists a **"policy"** distribution shift between offline learning policy and the given dataset collected by behavior policies and results in *distribution shift in z-space*. This "policy" distribution shift is similar to that in single-task offline RL [5]. In contrast, in this paper, we characterize the **"transition and reward"** distribution mismatch between offline task-dependent data collection and online adaptation (see Eq. (4)). Note that in the single-task offline RL [5], the reward and transition function between the given dataset and the environment is consistent, and thus the "transition and reward" distribution mismatch is unique for offline meta-RL. Moreover, the "transition and reward" distribution mismatch can exacerbate the distribution shift in the z-space since the input context of z includes states, actions, and rewards. We will add a detailed discussion in the revision.
> > >
> > > **Q8**: In Section 6, how is the distribution shift discussed in this paper different from the distribution shift discussed in BOReL?
> > >
> > > **A8**: BOReL does not study distribution shift in offline meta-RL with online adaptation. Instead, it focuses on *MDP ambiguity* for task inference, in which it may be impossible to learn an inference model that accurately distinguishes between the different MDPs due to narrow task-dependent behavior policies. As discussed in Q6, in our paper, we focus on the "transition and reward" distribution mismatch between offline task-dependent data collection and online adaptation (see Eq. (4)), which is a different research problem. As shown by results in Table 1, 2, and 6 in the revision, BORel does not address this distribution mismatch. In contrast, our method GCC can address the problem of MDP ambiguity using filtering out out-of-distribution adaptation episodes.
> > >
> > > **Q9**: Reproducibility - It is not clear if code will be released publicly.
> > >
> > > **A9**: The code was provided with the original submission in the Supplementary Material.
> > >
> > > [1] Fujimoto, Scott, David Meger, and Doina Precup. "Off-policy deep reinforcement learning without exploration." International conference on machine learning. PMLR, 2019.
> > >
> > > [2] Mendonca R, Geng X, Finn C, et al. Meta-reinforcement learning robust to distributional shift via model identification and experience relabeling[J]. arXiv preprint arXiv:2006.07178, 2020.
> > >
> > > [3] Kidambi, Rahul, et al. "Morel: Model-based offline reinforcement learning." Advances in neural information processing systems 33 (2020): 21810-21823.
> > >
> > > [4] Yu, Tianhe, et al. "Combo: Conservative offline model-based policy optimization." Advances in neural information processing systems 34 (2021): 28954-28967.
> > >
> > > [5] Levine S, Kumar A, Tucker G, et al. Offline reinforcement learning: Tutorial, review, and perspectives on open problems[J]. arXiv preprint arXiv:2005.01643, 2020.
> > >
> > > [6] Strens M. A Bayesian framework for reinforcement learning[C]//ICML. 2000, 2000: 943-950.

---

> > > > ### Author Response · Authors · 2022-12-05
> > > > **Sincerely looking forward to further feedback**
> > > >
> > > > Dear Reviewer,
> > > >
> > > > Thank you for your time and efforts in reviewing our work. We have provided detailed clarification and experimental results to address the issues raised in your comments. If our response has addressed your concerns, we would be grateful if you could re-evaluate our work.
> > > >
> > > > If you have any additional questions or comments, we would be happy to have further discussions.
> > > >
> > > > Thanks,
> > > >
> > > > The authors

---

> > > > > ### Author Response · Authors · 2022-12-10
> > > > > **Further clarification on the questions (Part I)**
> > > > >
> > > > > Thanks for your careful reading and constructive feedback. We will provide clarification and experimental results to your questions and concerns as below. If you have any additional questions or comments, please post them and we would be happy to have further discussions.
> > > > >
> > > > > **Q1:** There is an assumption that the offline meta-training dataset is collected by expert policies, and thus only contains high-return trajectories. In my first review, I missed this assumption, which is stated in Section 2.2.
> > > > >
> > > > > **A1:** The reviewer may misunderstand our offline task-dependent data collection, and we do not make this assumption. As introduced in Section 2.2, we collect offline data for each meta-training task using task-dependent behavior policies $p(\mu|\kappa)$, which is a common formulation for offline meta-RL (see FOCAL [1], MACAW [2], and BOReL [3]). Each behavior policy $\mu_i$ can be an expert policy, a sub-optimal policy, or a mixed policy (e.g., corresponding to the replay buffer of a pre-trained RL agent). To serve intuitions, we adopt expert task-dependent behavior policies to demonstrate the distributional shift of reward and transition distribution in the didactic examples (e.g., Figure 1, 2, and 3). In Appendix F, we conduct ablation studies to investigate our method GCC in Meta-World ML1 tasks with sub-optimal behavior policies (i.e., with medium datasets). We also demonstrate the performance of GCC, FOCAL, and MACAW and the average performance of pre-trained SAC in Table x1 and x2, respectively.
> > > > >
> > > > > | Environment           |GCC           | FOCAL                        | MACAW                        |
> > > > > |:------------------------:|:---------------------------:|:---------------------------:|:---------------------------:|
> > > > > | Sweep      | 3352 $\pm$   174    | 1392 $\pm$ 348   |  870 $\pm$ 870   |
> > > > > | Sweep-Medium      | 2568 $\pm$  565     | 1654 $\pm$ 565   |  174 $\pm$ 131   |
> > > > > | Peg-Insert-Side      | 1148 $\pm$ 153      | 306 $\pm$ 115  |  3.84 $\pm$ 2.47 |
> > > > > | Peg-Insert-Side-Medium      | 1130 $\pm$ 535    | 383 $\pm$ 268  |  2.34 $\pm$ 1.48   |
> > > > >
> > > > >
> > > > > Table x1: Comparison of GCC, FOCAL and MACAW on datasets of different qualities.
> > > > >
> > > > > | Sweep           |Sweep-Medium            | Peg-Insert-Side                         | Peg-Insert-Side-Medium                        |
> > > > > |:------------------------:|:---------------------------:|:---------------------------:|:---------------------------:|
> > > > > | 4354      | 2874   | 3826   |  2342   |
> > > > >
> > > > >
> > > > > Table x2: During task-dependent offline data collection, the average performance of pre-trained SAC on each task.
> > > > >
> > > > > The empirical results demonstrate that GCC achieves state-of-the-art performance in the medium datasets collected by sub-optimal behavior policies.
> > > > >
> > > > > **Q2:** The method cannot make use of low-return exploration trajectories to update the task belief, so exploration during meta-testing is constrained to sampling the unadapted meta-learned policy. Thus the method effectively does not address the problem of exploration in meta-RL, which is a key challenge.
> > > > >
> > > > > **A2:** Context-based task inference module cannot perform accurate belief updates when receiving out-of-distribution contexts (i.e., not contained in the offline datasets) [4]. In contrast, our proposed method GCC can solve sparse-reward tasks that require meta-testing exploration (e.g., Sparse-HalfCheetahDir and Sparse-Point-Robot proposed in HyperX [5]) with more adaptation episodes.
> > > > >
> > > > > For online meta-RL, common informative exploration trajectories with low return need to be collected by a meta-policy. However, in the offline meta-RL, the community [1-3] collects offline multi-task datasets by task-dependent behavior policies for utilizing multi-source data. Different from the meta-training exploration of online meta-RL, the offline dataset collected by task-dependent behavior policies usually includes ***exploitation trajectories*** for each meta-training task. Current off-the-shelf context-based meta-training algorithms (e.g., FOCAL) aim to extract effective exploitation policies for each generalizable task embedding (i.e., task hypothesis) in offline meta-training. Compared with potential out-of-distribution informative exploration trajectories, these effective exploitation policies can achieve high returns and contain sufficient information in the corresponding tasks. Our method GCC will iteratively sample a bunch of **diverse** task hypotheses and adapt the high-return contexts (potentially generated by exploitation policies with accurate or nearly correct task hypotheses) to meta-testing tasks. As the reviewer may realize, one limitation of our method may discard some exploratory trajectories with low returns and use more adaptation episodes.

---

> > > > > > ### Author Response · Authors · 2022-12-10
> > > > > > **Further clarification on the questions (Part II)**
> > > > > >
> > > > > > We evaluate GCC and FOCAL (which may use out-of-distribution episodes to update the task belief) in Sparse-Cheetah-Dir and Sparse-Point-Robot. For both tasks, we pre-train task-dependent SAC with default entropy-based exploration on each meta-training task to collect offline multi-task datasets.
> > > > > >
> > > > > > | Algorithm \ The number of Adaptation Episodes             | 2                         | 5             |
> > > > > > |:------------------------:|:---------------------------:|:-----------------:|
> > > > > > | GCC            | 115 $\pm$ 73    |  841 $\pm$ 25    |
> > > > > > | FOCAL            | -51.6 $\pm$ 11.7    |  -44.4 $\pm$ 15.2    |
> > > > > >
> > > > > > Table x3: Comparison of GCC and FOCAL on Sparse-Cheetah-Dir. During task-dependent offline data collection, the average performance of pre-trained SAC on each task is 856 $\pm$ 42.
> > > > > >
> > > > > >
> > > > > > | GCC           |FOCAL           | SAC's Optimal Return                         |
> > > > > > |:------------------------:|:---------------------------:|:---------------------------:|
> > > > > > | 7.78 $\pm$ 0.64       | 0.83 $\pm$ 0.37      | 8.27 $\pm$ 0.29   |
> > > > > >
> > > > > > Table x4: Comparison of GCC, FOCAL, and the optimal return achieved by pre-trained SAC on each task individually on Sparse-Point-Robot.
> > > > > >
> > > > > > In Sparse-Cheetah-Dir, the average performance of pre-trained SAC (i.e., containing effective exploitation trajectories) is 856. Results in Table x3 show that GCC with 5 adaptation episodes can infer the accurate task hypothesis and solve this task, i.e., achieving comparable performance with pre-trained task-dependent behavior policies. Comparing GCC with FOCAL, the results show that our filtering method can significantly improve performance using an in-distribution context. The performance on different adaptation episodes indicates that GCC will use more adaptation episodes to find a good task hypothesis. In this setting, tasks are uniform over “walk forward” and “walk backward.” In online meta-RL, the exploration behavior in HyperX can try one direction, receive a low return, and infer the right goal in the other direction. However, such trajectories are challenging to collect by a task-dependent pre-trained SAC. The offline dataset includes exploitation trajectories, either “walk forward” or “walk backward,” with higher returns. GCC will sample a task hypothesis and try its exploitation policy learned from the offline dataset. When GCC selects a wrong task hypothesis, the return will be low, and GCC continues to sample the subsequent task hypotheses to find the correct task hypothesis.
> > > > > >
> > > > > > In Sparse-Point-Robot, the goals are uniform over the semi-circle, and the agent aims to navigate to the goal. In online meta-RL, the exploration behavior will explore the goals along the semi-circle in each adaptation episode, which is informative and achieves low return. However, such data is also challenging to collect by the task-dependent SAC, which is pre-trained on each task. The offline dataset includes exploitation trajectories toward the goal. Our method GCC will sample a bunch of diverse task hypotheses and try their exploitation policy to find informative exploitation trajectories with higher returns (i.e., with nearly accurate task hypotheses). Due to the radius of the goal, GCC can address Sparse PointRobot within 20 adaptation episodes. As shown in Table x4, GCC achieves similar performance to the optimal returns achieved by pre-trained SAC. In addition, as shown in Table 1, 2, and 6-8 in the paper, GCC also achieves state-of-the-art performance in the Meta-World ML1 benchmark with 50 tasks, which demonstrates the generality of our greedy filtering method.
> > > > > >
> > > > > > Distinguishing between in-distribution (if it exists) and out-of-distribution exploration trajectories with lower returns during meta-testing is also an interesting and exciting future direction. We can utilize in-distribution exploration trajectories to achieve fewer adaptation episodes in meta-testing tasks. We will add a detailed discussion in Q2 to the revision of paper.
> > > > > >
> > > > > > [1] Li, L., Yang, R., Luo, D. (2020). Focal: Efficient fully-offline meta-reinforcement learning via distance metric learning and behavior regularization. arXiv preprint arXiv:2010.01112.
> > > > > >
> > > > > > [2] Mitchell, Eric, et al. "Offline meta-reinforcement learning with advantage weighting." International Conference on Machine Learning. PMLR, 2021.
> > > > > >
> > > > > > [3] Dorfman R, Shenfeld I, Tamar A. Offline Meta Reinforcement Learning--Identifiability Challenges and Effective Data Collection Strategies[J]. Advances in Neural Information Processing Systems, 2021, 34: 4607-4618.
> > > > > >
> > > > > > [4] Mendonca R, Geng X, Finn C, et al. Meta-reinforcement learning robust to distributional shift via
> > > > > > model identification and experience relabeling[J]. arXiv preprint arXiv:2006.07178, 2020.
> > > > > >
> > > > > > [5] Exploration in Approximate Hyper-State Space for Meta Reinforcement Learning. Zintgraf et al., '21.

---

### Decision · Program_Chairs · 2023-01-20

**Decision:**

Reject

**Justification For Why Not Higher Score:**

The limitation of the algorithm may lead to severe drop of performance but is not well discussed even after rebuttal.

**Justification For Why Not Lower Score:**

N/A

**Metareview: Summary, Strengths And Weaknesses:**

This paper considers an offline meta-RL setting with few-shot online adaptation where in the test task no data is given ahead of time and the agent relies on its own exploration to learn the new task. It provides theoretical analysis on the data distribution shift that is caused by the different behavior policy, that is, task-dependent behavior policies in training tasks versus offline meta-policy in test tasks. It proposes to filter out-of-distribution data during online exploration and use filtered data to update the posterior of test tasks. It then introduces a greedy context-based data distribution correction approach, GCC, and filter data according to returns. Experiments on Meta-World ML1 benchmark shows SOTA performance compared to recent offline meta-RL baselines.

Strengths:
- It studies an important problem in offline meta-RL
- The theoretical analysis of the data distribution shift caused by different behavior policies in the offline meta-RL setting is novel and insightful. This is different from various distribution shift previously studied, e.g., in meta-training/test tasks in meta-RL setting and the offline-online policy shift in single-task RL setting.
- The proposed filtering based approach to mitigate distribution shift is theoretically justified.
- SOTA performance in the offline meta-RL online adaptation setting that none of prior works studied.

Weaknesses:
- The particular instantiation of the proposed filtering approach that uses return to filter data has a severe limitation. As pointed out by multiple reviewers, it will ignore low return trajectories that could be used to update the task posterior distribution. Also, filtering out low return trajectory will cause the agent fail to learn in a task that requires exploration. The method would not apply to offline dataset collected by low quality policies either. This limitation is not sufficiently discussed or demonstrated in the experiments.
- Clarity could be further improved. Multiple reviewers express confusion about the particular OOD problem the paper studied, the behavior policy in meta-training. While the rebuttal provide much better clarification than the submitted version, the paper could be further improved with more thorough explanation.

**Summary Of Ac-Reviewer Meeting:**

During the AC-reviewer meeting, the concern on the limitation of the concrete proposed algorithm based on trajectory return was brought up and discussed. Multiple reviewers (including those from offline discussion) agree that this was a severe limitation and was not properly justified conceptually or analyzed in the experiments.

A related concern raised during the discussion was when the proposed notion of filtering could be possible. Even in the theoretical setting some exploratory behaviors may not reveal information about the MDP. While the new appendix is useful, the limitation could be better discussed explicitly.

Based on the shared concern among reviewers, I consider this paper to be below the bar for acceptance. I would encourage the authors to discuss this limitation more explicitly in a future revision and study the condition under which the proposed method would work.